# Information-Theoretic Generalization Analysis for Expected Calibration Error

**Futoshi Futami**[*]
Osaka University / RIKEN AIP
futami.futoshi.es@osaka-u.ac.jp

**Masahiro Fujisawa**[*†]
RIKEN AIP
masahiro.fujisawa@riken.jp

## Abstract

While the expected calibration error (ECE), which employs *binning*, is widely adopted to evaluate the calibration performance of machine learning models, theoretical understanding of its estimation bias is limited. In this paper, we present the first comprehensive analysis of the estimation bias in the two common binning strategies, *uniform mass* and *uniform width binning*. Our analysis establishes upper bounds on the bias, achieving an improved convergence rate. Moreover, our bounds reveal, for the first time, the optimal number of bins to minimize the estimation bias. We further extend our bias analysis to generalization error analysis based on the information-theoretic approach, deriving upper bounds that enable the numerical evaluation of how small the ECE is for unknown data. Experiments using deep learning models show that our bounds are nonvacuous thanks to this information-theoretic generalization analysis approach.

## 1 Introduction

Ensuring reliable predictions from machine learning models holds paramount importance in risk-sensitive applications such as medical diagnosis [18]. To achieve this, it is essential not only to evaluate the accuracy of the point predictions of models but also to precisely quantify the associated uncertainty. One effective approach to accomplishing this is to leverage the concept of *calibration*. In the classification context, the calibration performance is evaluated by how well predictive probabilities of a model align with the actual frequency of true labels, and a close correspondence between them indicates that the model is well calibrated [7, 42]. Unfortunately, machine learning models are often not well calibrated [11, 24], prompting extensive research on their calibration performance both theoretically and numerically. In this paper, we assume a binary classification problem.

To evaluate the calibration performance, we often use the *calibration error* or *true calibration error* (TCE) [12, 31, 10]. This evaluates the disparity between the predicted probability of a model and the conditional expectation of the label given the model prediction, instead of the true label frequency. However, analytically computing the TCE is challenging, primarily due to the intractability of the conditional expectation. One way to address this issue is by using the *binning* method [48]. This method enables the estimation of conditional expectations by dividing the probability range $[0, 1]$ into $B$ small intervals called *bins* and comparing the empirical mean of predictive probabilities and label frequencies within each bin, utilizing the finite test dataset denoted as $S_{\text{te}}$. The obtained estimator of the TCE is termed the (binned) *expected calibration error* (ECE).

Given that the ECE estimates the TCE, it is crucial to theoretically explore the bias between them, termed *total bias* in this paper, to confirm the accuracy of calibration evaluation using the ECE. Furthermore, it is vital to identify the conditions under which our training algorithm achieves a low ECE or TCE for unknown test datasets. This can be paraphrased as the importance of conducting *generalization error analysis* under the ECE and TCE. Nevertheless, research on these aspects remains

---

[*]Equal contribution.    [†]Corresponding author.

38th Conference on Neural Information Processing Systems (NeurIPS 2024).

scant. Existing studies have only shown that the ECE underestimates the TCE [24] and have only analyzed the bias caused by a finite sample under specific conditions such as using uniform-mass binning (UMB) [13, 12]. Consequently, there remains a significant gap in the comprehensive theoretical understanding of the biases introduced by binning (termed *binning bias*) and the *statistical bias* resulting from the use of finite test data samples. While these studies have concentrated on scenarios utilizing UMB, there has been no corresponding analysis for uniform-width binning (UWB), which is also frequently employed in practice. This limitation could be due to the challenges posed by UWB, where the equal partitioning of probability intervals can lead to bins without any samples, making bias analysis difficult. Unfortunately, to the best of our knowledge, there are also no existing generalization analyses on the basis of the ECE and TCE.

To address the challenges outlined above, in this paper, we comprehensively analyze the ECE for both UWB and UMB. We derive the upper bounds of the total bias of the ECE using a newly derived concentration inequality (Corollary 1). Our bounds improve the order of convergence regarding the bin size. Furthermore, the optimal bin size that minimizes the total bias is successfully derived from these results. With this optimal bin size, the total bias of the ECE exhibits a rate of $\mathcal{O}(1/n_{\text{te}}^{1/3})$ for both UWB and UMB, where $n_{\text{te}}$ is the number of test samples (Eq. (13)).

This bias analysis leads to our second novel contribution, providing the *generalization error analysis* for the ECE and TCE (Theorems 4 and 5) using the *information-theoretic* (IT) analysis [43, 15, 17]. Directly applying the existing IT analysis is, however, challenging because the ECE on the training dataset is no longer represented by the sum of independent and identically distributed (i.i.d.) random variables w.r.t. the trained model. We circumvent this problem by applying a novel exponential moment inequality derived in the process of our bias analysis described above. We further connect our results to classical uniform convergence theory using the metric entropy of function classes, which allows us to discuss the convergence rate of our bounds under a broader range of models (Theorem 6). Using our generalization bounds, we theoretically explore the existing conjecture [11, 24] that recalibration with the reuse of training data leads to severe overfitting. We then show that our analysis successfully characterizes such an additional bias (Theorem 7). Numerical experiments using deep learning models confirm that our bound is nonvacuous and validate our findings.

## 2   Preliminaries

For a random variable denoted in capital letters, we express its realization with corresponding lower-case letters. Let $P_X$ denote the distribution of $X$, and let $P_{Y|X}$ represent the conditional distribution of $Y$ given $X$. We express the expectation of a random variable $X$ as $\mathbb{E}_X$. The symbol $I(X;Y)$ represents the mutual information (MI) between $X$ and $Y$, while $I(X;Y|Z)$ is the conditional MI (CMI) between $X$ and $Y$ given $Z$. We further define $[n] = \{1, \ldots, n\}$ for $n \in \mathbb{N}$.

We consider binary classification in this paper. Let $\mathcal{Z} = \mathcal{X} \times \mathcal{Y}$ be the domain of data, where $\mathcal{X}$ and $\mathcal{Y} = \{0, 1\}$ are input and label spaces, respectively. Suppose $\mathcal{D}$ represents an *unknown* data distribution, and let $S_{\text{tr}} := \{Z_m\}_{m=1}^n$ denote the training dataset consisting of $n$ samples drawn i.i.d. from $\mathcal{D}$. We also define the test dataset comprising $n_{\text{te}}$ samples as $S_{\text{te}} \sim \mathcal{D}^{n_{\text{te}}}$. Let $f_w : \mathcal{X} \to [0, 1]$ be a parametric probabilistic classifier that outputs a prediction of the probability $Y = 1$, and we denote its parameters as $w \in \mathcal{W} \subset \mathbb{R}^d$. We consider a randomized algorithm $\mathcal{A} : \mathcal{Z}^n \times \mathcal{R} \to \mathcal{W}$, where $R \in \mathcal{R}$ is the randomness of an algorithm, independent of all other random variables. For fixed $R = r$ and $S_{\text{tr}} = s$, $\mathcal{A}(s, r)$ is a deterministic function and $f_{\mathcal{A}(s,r)}(x)$ is the prediction at point $x$ given $s$ and $r$. We evaluate the accuracy of the trained predictor $f_w$ using the loss function $l : \mathcal{W} \times \mathcal{Z} \to [0, 1]$, where $l(\mathcal{A}(s, r), z)$ denotes the loss incurred by the prediction $f_w(x)$ for label $y$. For example, the zero-one loss is commonly used to evaluate the accuracy. Then, the training loss is given by $\hat{L}_{S_{\text{tr}}} := \frac{1}{n} \sum_{m=1}^n l(\mathcal{A}(S_{\text{tr}}, R), Z_m)$ and the test loss is given as $L_Z := l(\mathcal{A}(S_{\text{tr}}, R), Z)$ where $Z \sim \mathcal{D}$. We also define the expected version of them as $L_S := \mathbb{E}_{S_{\text{tr}}, R} \hat{L}_{S_{\text{tr}}}$ and $L_{\mathcal{D}} := \mathbb{E}_{S_{\text{tr}}, Z, R} L_Z$.

### 2.1   Calibration error and its estimator

In this section, we introduce a calibration metric and its corresponding estimator. The most widely known metric is the *true calibration error* (TCE) [31, 10, 12] defined as

$$\text{TCE}(f_w) := \mathbb{E}\left[|\mathbb{E}[Y|f_w(X)] - f_w(X)|\right], \tag{1}$$

conditioned on $W = w$. Unfortunately, evaluating the TCE directly is challenging due to the intractable calculation of $\mathbb{E}[Y|f_w(X)]$. To avoid this issue, we often use the *binning* method [11, 48, 49]. This method estimates the TCE by partitioning the prediction probability range $[0, 1]$ into $B$ intervals $\mathcal{I} = \{I_i\}_{i=1}^{B}$ (called *bins*) and averaging within each bin using the evaluation dataset $S_e \coloneqq \{Z_m\}_{m=1}^{n_e} \in \mathcal{Z}^{n_e}$, where we assume $n_e \geq 2B$. For instance, we have $S_e = S_{\text{te}}$ when the test dataset is used for evaluation. The TCE estimator on the basis of $\mathcal{I}$, with $S_e$, is defined as

$$\text{ECE}(f_w, S_e) \coloneqq \sum_{i=1}^{B} p_i |\bar{f}_{i,S_e} - \bar{y}_{i,S_e}|, \tag{2}$$

where $|I_i| \coloneqq \sum_{m=1}^{n_e} \mathbb{1}_{f_w(x_m) \in I_i}$, $p_i \coloneqq \frac{|I_i|}{n_e}$, $\bar{f}_{i,S_e} \coloneqq \frac{1}{|I_i|} \sum_{m=1}^{n_e} \mathbb{1}_{f_w(x_m) \in I_i} f_w(x_m)$, and $\bar{y}_{i,S_e} \coloneqq \frac{1}{|I_i|} \sum_{m=1}^{n_e} \mathbb{1}_{f_w(x_m) \in I_i} y_m$. This estimator is called the *expected calibration error* (ECE) [3].

There are two common methods to construct $\mathcal{I}$. One is *uniform width binning* (UWB) [11], which divides the $[0, 1]$ interval into equal widths as follows: $I_i = ((i - 1)/B, i/B]$ for $i$ in $[B]$. The other approach is *uniform mass binning* (UMB) [48], which sets $\mathcal{I}$ so that each bin contains an equal number of samples. That is, we calculate predicted probabilities as $f_m = f_w(x_m)$ for $x_m \in S_e$, let $f_{(m)}$ be the $m$-th order statistics of $(f_1, \ldots, f_{n_e})$, and then set $I_1 = (0, u_1], I_2 = (u_1, u_2], \ldots, I_B = (u_{B-1}, u_B]$ for $b \in [B-1]$ and $u_b \coloneqq f_{(\lfloor n_e b/B \rfloor)}$ with $u_B = 1$. Here, $\lfloor x \rfloor \coloneqq \max\{m \in \mathbb{Z} : m \leq x\}$.

## 2.2 Biases of ECE and limitation of existing work

Given that the ECE is an estimator of the TCE, it is of practical importance to understand the nature of the bias defined as $|\text{TCE}(f_w) - \text{ECE}(f_w, S_e)|$. We call this bias as the *total bias*. To facilitate the total bias analysis, we adopt the following definition of the binned function of $f_w$ [24]:

$$f_{\mathcal{I}}(x) \coloneqq \sum_{i=1}^{B} \mathbb{E}[f_w(X)|f_w(X) \in I_i] \cdot \mathbb{1}_{f_w(x) \in I_i}, \tag{3}$$

which represents the conditional expectation within each bin. When we evaluate the ECE on $S_e = S_{\text{te}}$, we expect that $\text{ECE}(f_w, S_{\text{te}})$ will converge to $\text{TCE}(f_{\mathcal{I}}) = \mathbb{E}|\mathbb{E}[Y|f_{\mathcal{I}}(x)] - f_{\mathcal{I}}(x)|$ with increasing $n_{\text{te}}$. However, $\text{TCE}(f_{\mathcal{I}})$ underestimates $\text{TCE}(f_w)$ [24, 10], that is,

$$\text{TCE}(f_{\mathcal{I}}) \leq \text{TCE}(f_w), \tag{4}$$

which implies that $\text{ECE}(f_w)$ is a biased estimator of $\text{TCE}(f_w)$. Therefore, comprehending the extent of this bias is crucial to an accurate calibration performance evaluation. Nevertheless, previous studies [24, 13, 12] have exclusively focused on the *statistical bias* in UMB, defined as $|\text{TCE}(f_{\mathcal{I}}) - \text{ECE}(f_w, S_{\text{te}})|$ as discussed in Section 1. This brings us to an analysis of the total bias for both UWB and UMB.

## 2.3 Information-theoretic generalization error analysis

We now briefly outline the IT analysis using the evaluated CMI (eCMI) [34] that we utilize in our study. Consider $\tilde{Z} \in \mathcal{Z}^{n \times 2}$ as an $n \times 2$ matrix, where each entry is drawn i.i.d. from $\mathcal{D}$. We refer to this matrix as a *supersample*. Each column of $\tilde{Z}$ has indexes $\{0, 1\}$ associated with $U = (U_1, \ldots, U_n) \sim \text{Uniform}(\{0, 1\}^n)$ independent of $\tilde{Z}$. We denote the $m$-th row as $\tilde{Z}_m$ with entries $(\tilde{Z}_{m,0}, \tilde{Z}_{m,1})$. In this setting, we consider $\tilde{Z}_U \coloneqq (\tilde{Z}_{m,U_m})_{m=1}^{n}$ as the training dataset and $\tilde{Z}_{\bar{U}} \coloneqq (\tilde{Z}_{m,\bar{U}_m})_{m=1}^{n}$ as the test dataset with $n_{\text{te}} = n$, where $\bar{U}_m = 1 - U_m$. With these notations, we can see that $\hat{L}_{\tilde{Z}} \coloneqq \frac{1}{n} \sum_{m=1}^{n} l(\mathcal{A}(\tilde{Z}_U, R), \tilde{Z}_{m,U_m})$ corresponds to the training error since $L_S = \mathbb{E}_{\tilde{Z}, R, U} \hat{L}_{\tilde{Z}}$. Also, $L_{\tilde{Z}} \coloneqq \frac{1}{n} \sum_{m=1}^{n} l(\mathcal{A}(\tilde{Z}_U, R), \tilde{Z}_{m,\bar{U}_m})$ corresponds to the test error, $L_{\mathcal{D}} = \mathbb{E}_{\tilde{Z}, R, U} L_{\tilde{Z}}$. The described settings, called the *CMI setting* [17], lead to the following theorem.

**Theorem 1** (Theorem 6.7 in Steinke & Zakynthinou [34]). *Under the CMI setting, we have*

$$\mathbb{E}_{\tilde{Z}, R, U} |\hat{L}_{\tilde{Z}} - L_{\tilde{Z}}| \leq \sqrt{\frac{2}{n}(\text{eCMI}(l) + \log 2)}, \tag{5}$$

---

[3] Although some existing studies refer to Eq. (1) as the ECE, in this study, we follow the definitions of the TCE and ECE outlined by Roelofs et al. [31] and Gruber & Buettner [10] to make a clear distinction from the estimator obtained through binning.

where $\text{eCMI}(l) := I(l(\mathcal{A}(\tilde{Z}_U, R), \tilde{Z}); U|\tilde{Z})$ and $l(\mathcal{A}(\tilde{Z}_U, R), \tilde{Z})$ is an $n \times 2$ loss matrix obtained by applying $l(\mathcal{A}(\tilde{Z}_U, R), \cdot)$ elementwise to $\tilde{Z}$.

The reason we focus on IT analysis is that it enables algorithm-dependent analysis. The conventional uniform convergence theory [37] focuses solely on function classes to derive bounds. However, recent findings suggest that models trained by some algorithms are not well calibrated but show high accuracy [11, 24]. Therefore, it seems essential to incorporate information about not only the function class but also the algorithm in the ECE analysis. Hence, in this paper, we adopt the eCMI framework, which is the generalized analysis approach that maximizes the use of algorithmic information. Furthermore, because eCMI-based bounds can be estimated using training and test data, the generalization performance of the model can be evaluated numerically, making it desirable from a practical standpoint.

## 3 Proposed analysis of total bias in binned ECE

Here, we present our first main analyses of the bias analysis of the ECE as the estimator of the TCE. Our analysis primarily focuses on the total bias defined as follows:

$$\text{Bias}_{\text{tot}}(f_w, S_{\text{te}}) := |\text{TCE}(f_w) - \text{ECE}(f_w, S_{\text{te}})|. \tag{6}$$

We can derive the following upper bound of Eq. (6) by using the triangle inequality,

$$\text{Bias}_{\text{tot}}(f_w, S_{\text{te}}) \leq \text{Bias}_{\text{bin}}(f_w, f_{\mathcal{I}}) + \text{Bias}_{\text{stat}}(f_w, S_{\text{te}}), \tag{7}$$

where $\text{Bias}_{\text{bin}}(f_w, f_{\mathcal{I}}) := |\text{TCE}(f_w) - \text{TCE}(f_{\mathcal{I}})|$ and $\text{Bias}_{\text{stat}}(f_w, S_{\text{te}}) := |\text{TCE}(f_{\mathcal{I}}) - \text{ECE}(f_w, S_{\text{te}})|$. We call the former as the *binning bias*, which arises from nonparametric estimation via binning, and the latter as the *statistical bias* caused by estimation on finite data points.

Before showing our results, we introduce the following assumption that is also used by Gupta & Ramdas [12] and Sun et al. [35]:

**Assumption 1.** *Given $W = w$, $f_w(x)$ is absolutely continuous w.r.t. the Lebesgue measure.*

This assumption means that $f_w(x)$ has a probability density, and it is satisfied without loss of generality as elaborated in Appendix C in Gupta & Ramdas [12].

From Eq. (7), we can obtain an upper bound on the total bias by analyzing the binning and statistical biases separately. First, we present the following results of our statistical bias analysis:

**Theorem 2** (Statistical bias analysis)**.** *Given $W = w$, under Assumption 1, we have*

$$\text{TCE}(f_{\mathcal{I}}) \leq \mathbb{E}_{S_{\text{te}}} \text{ECE}(f_w, S_{\text{te}}), \tag{8}$$

$$\mathbb{E}_{S_{\text{te}}} \text{Bias}_{\text{stat}}(f_w, S_{\text{te}}) \leq \begin{cases} \sqrt{\dfrac{2B \log 2}{n_{\text{te}}}} & \text{(for UWB)}, \\ \sqrt{\dfrac{2B \log 2}{n_{\text{te}} - B}} + \dfrac{2B}{n_{\text{te}} - B} & \text{(for UMB)}. \end{cases} \tag{9}$$

*Proof sketch.* First, we reformulate the ECE as $\text{ECE}(f_w, S_{\text{te}}) = \sum_{i=1}^{B} |\mathbb{E}_{(X,Y) \sim \hat{S}_{\text{te}}}(Y - f_w(X)) \cdot \mathbb{1}_{f_w(X) \in I_i}|$ and the TCE as $\text{TCE}(f_{\mathcal{I}}) = \sum_{i=1}^{B} |\mathbb{E}_{(X,Y) \sim \mathcal{D}}(Y - f_w(X)) \cdot \mathbb{1}_{f_w(X) \in I_i}|$, where $\hat{S}_{\text{te}}$ is the empirical distribution of $S_{\text{te}}$. The proof of this reformulation is shown in Appendix C.1. Thanks to these transformations, our analysis does not have the problem that the UWB method can lead to bins without any samples. By evaluating the exponential moment for UWB using McDiarmid's inequality under these reformulation, we have, for any $\lambda \geq 0$,

$$\lambda \mathbb{E}_{S_{\text{te}}} \text{Bias}_{\text{stat}}(f_w, S_{\text{te}}) \leq \log \mathbb{E}_{S_{\text{te}}} e^{\lambda |\text{TCE}(f_{\mathcal{I}}) - \text{ECE}(f_w, S_{\text{te}})|} \leq B \log 2 + \lambda^2/(2n_{\text{te}}). \tag{10}$$

Using this, we can derive both the bias and the high probability bound. We can derive a similar bound for UMB. The complete proof is provided in Appendix D.1. $\qquad\square$

Eq. (8) shows that $\text{ECE}(f_w, S_{\text{te}})$ overestimates $\text{TCE}(f_{\mathcal{I}})$ in expectation. Combined with Eq. (4), we can see that $\text{ECE}(f_W, S_{\text{te}})$ cannot be the upper or lower bound of $\text{TCE}(f_w)$ in expectation. This emphasizes the importance of the rigorous bias analysis of $|\text{TCE}(f_w) - \text{ECE}(f_w, S_{\text{te}})|$.

**Comparison with existing work:** Eq. (9) provides better generality and a tighter bound than prior results. Our bound exhibits $\mathcal{O}(\sqrt{B/n_{\text{te}}})$ in expectation (and $\mathcal{O}_p(\sqrt{B/n_{\text{te}}})$ in high probability w.r.t. $S_{\text{te}}$ proved in Appendix D.3.). In contrast, the existing analysis [13, 12, 24] provided a similar bound focused on UMB scale as $\mathcal{O}(B/\sqrt{n_{\text{te}}})$ in expectation (and $\mathcal{O}_p(\sqrt{B \log B/n_{\text{te}}})$ in high probability). In terms of generality, our derivation techniques can be applied to both UMB and UWB, whereas existing bounds are limited to UMB.

**Pros of our proof technique:** The proof procedure in existing work [13, 12, 24] involves (i) showing that the samples assigned to each bin are i.i.d., (ii) applying the Hoeffding inequality to derive concentration bounds *separately for each bin*, and (iii) summing up these error bounds across all bins. This approach results in slow convergence and only applicable to UMB. On the other hand, our approach simultaneously handles all bins by utilizing the concentration inequality in Eq. (10) and provides the improved upper bound and can be used for both UWB and UMB. We offer a more detailed explanation of this in Appendix D.6.

Next, we show the results of our binning bias analysis under the following common assumption in the nonparametric estimation context [36].

**Assumption 2.** *Given $W = w$, $\mathbb{E}[Y|f_w(x)]$ satisfies L-Lipschitz continuity.*

**Theorem 3** (Binning bias analysis)**.** *Given $W = w$, under Assumptions 1 and 2, we have*

$$\mathbb{E}_{S_{\text{te}}}\text{Bias}_{\text{bin}}(f_w, f_{\mathcal{I}}) \leq \begin{cases} \frac{1+L}{B} & \textit{(for UWB)}, \\ (1+L)(\frac{1}{B} + \sqrt{\frac{2B \log 2}{n_{\text{te}}-B}} + \frac{2B}{n_{\text{te}}-B}) & \textit{(for UMB)}. \end{cases} \tag{11}$$

*Proof sketch.* In Appendix D.4, we show that

$$\text{Bias}_{\text{bin}}(f_w, f_{\mathcal{I}}) \leq \mathbb{E}|\mathbb{E}[Y|f_w(X)] - \mathbb{E}[Y|f_{\mathcal{I}}(X)]| + \mathbb{E}|f_w(X) - f_{\mathcal{I}}(X)|.$$

We then derive the upper bound of the right-hand side from the definitions of bins in Section 2.1. For UWB, the upper bound is $\mathcal{O}(1/B)$ because UWB divides the interval into equal widths. For UMB, we need to evaluate how samples are split by bins. The complete proof is in Appendix D.4. $\qquad\square$

Substituting the results from Theorems 2 and 3 into Eq. (7) yields the following upper bound for the total bias.

**Corollary 1.** *Given $W = w$, under Assumptions 1 and 2, we have*

$$\mathbb{E}_{S_{\text{te}}}\text{Bias}_{\text{tot}}(f_w, S_{\text{te}}) \leq \begin{cases} \frac{1+L}{B} + \sqrt{\frac{2B \log 2}{n_{\text{te}}}} & \textit{(for UWB)}, \\ \frac{1+L}{B} + (2+L)(\sqrt{\frac{2B \log 2}{n_{\text{te}}-B}} + \frac{2B}{n_{\text{te}}-B}) & \textit{(for UMB)}. \end{cases} \tag{12}$$

The above result evidently implies a trade-off concerning $B$. Intuitively, this indicates that while a larger number of bins, $B$, improves the precision of $f_w$ estimation, accurately estimating the conditional expectation requires a greater sample size. We further determine the optimal number of bins by minimizing the upper bound of Eq. (12) w.r.t. $B$, which results in $B = \mathcal{O}(n_{\text{te}}^{1/3})$ and gives

$$\mathbb{E}_{S_{\text{te}}}\text{Bias}_{\text{tot}}(f_W, S_{\text{te}}) = \mathcal{O}(1/n_{\text{te}}^{1/3}). \tag{13}$$

Since the bin size has been tuned heuristically in practice, this result sheds light on how to choose it theoretically for both UMB and UWB rigorously.

**Regarding tightness of Eq.** (12)**:** As we mentioned in Section 2.1, we use binning methods to estimate intractable $\mathbb{E}[Y|f_w(x)]$ in the TCE evaluation. Thus, the TCE evaluation can be viewed as nonparametric estimation of a one-dimensional function on $[0, 1]$. According to Tsybakov [36], the error in such nonparametric regression *cannot be smaller than* $\mathcal{O}(1/n_{\text{te}}^{1/3})$ under Assumption 2. Our bound is convincing because its order aligns with that in Tsybakov [36]. We provide a detailed discussion in Appendix F.6 and F.7.

We finally remark that the total bias of binning using UWB and UMB cannot be improved even assuming the Hölder continuity for $\mathbb{E}[Y|f_w(x)]$ instead of Assumption 2. This is because the binning bias includes the error term $\mathbb{E}|f_w(X) - f_{\mathcal{I}}(X)| = \mathcal{O}(1/B)$ even under the Hölder continuity. Thus, we suffer from the slow converge $\mathbb{E}_{S_{\text{te}}}\text{Bias}_{\text{tot}}(f_w, S_{\text{te}}) = \mathcal{O}(1/n_{\text{te}}^{1/3})$ under the optimal bin size

$B = \mathcal{O}(n_{\text{te}}^{1/3})$ (see Appendix D.5 for this proof). According to Tsybakov [36], the lower bound of the nonparametric estimation is $\mathcal{O}(1/n_{\text{te}}^{\beta/(2\beta+1)})$ under $\beta$-Hölder continuity. This implies that the binning method cannot leverage the underlying smoothness of the data distribution. Thus, the slow convergence is the fundamental limitation of the binning scheme for both UMB and UWB.

## 4 Generalization error analysis in calibration error

Another goal of our study is to identify the conditions under which a training algorithm achieves a low ECE or TCE on unknown data by analyzing the generalization error, which has been overlooked in previous studies. In this section, we present our theoretical analysis regarding these points.

### 4.1 Information-theoretic analysis of generalization error in ECE and TCE

The expected generalization error between the ECE and TCE can be defined through the total bias notion, that is, $\mathbb{E}_{R,S_{\text{tr}}}\text{Bias}_{\text{tot}}(f_W, S_{\text{tr}}) := \mathbb{E}_{R,S_{\text{tr}}}|\text{TCE}(f_W) - \text{ECE}(f_W, S_{\text{tr}})|$. In this section, we derive the upper bound of $\mathbb{E}_{R,S_{\text{tr}}}\text{Bias}_{\text{tot}}(f_W, S_{\text{tr}})$ by analyzing the statistical and binning biases in the same manner as in Section 3. First, we derive the following upper bound of the statistical bias, $\text{Bias}_{\text{stat}}(f_w, S_{\text{tr}}) := |\text{TCE}(f_{\mathcal{I}}) - \text{ECE}(f_w, S_{\text{tr}})|$, using a similar proof technique as in Theorem 2.

**Theorem 4** (Generalization error bound of the ECE). *Under the CMI setting and under Assumption 1, for both UWB and UMB, we have*

$$\mathbb{E}_{R,S_{\text{tr}}}\text{Bias}_{\text{stat}}(f_W, S_{\text{tr}}) \leq \mathbb{E}_{R,S_{\text{tr}},S_{\text{te}}}|\text{ECE}(f_W, S_{\text{te}}) - \text{ECE}(f_W, S_{\text{tr}})| \leq \sqrt{\frac{8(\text{eCMI}(\tilde{l}) + B\log 2)}{n}},$$
(14)

*where* $\text{eCMI}(\tilde{l}) = I(\tilde{l}; U|\tilde{Z})$ *and*

$$\tilde{l}(U, R, \tilde{Z}) := |\text{ECE}(f_{\mathcal{A}(\tilde{Z}_U, R)}, \tilde{Z}_{\bar{U}}) - \text{ECE}(f_{\mathcal{A}(\tilde{Z}_U, R)}, \tilde{Z}_U)|.$$
(15)

*Proof sketch.* We reformulate the ECE similarly to the proof outline in Theorem 2. Errors between the losses evaluated on the training and test data are similar to the left-hand side of Eq. (5); however, directly applying Eq. (5) leads to a suboptimal rate of $\mathcal{O}(B/\sqrt{n})$. Therefore, we derive the extended version of Eq. (5) by correlating $B$ bins according to Eq. (10) and combining this with the Donsker–Varadhan lemma. The complete proof can be found in Appendix E.1. □

Comparing Eq. (14) with Eq. (9) in Theorem 2, we find that eCMI measures the *additional bias* that arises when evaluating the ECE using training data that are dependent on the trained model $f_w$ instead of test data, which are independent of it. In other words, the term $\mathbb{E}_{R,S_{\text{tr}},S_{\text{te}}}|\text{ECE}(f_W, S_{\text{te}}) - \text{ECE}(f_W, S_{\text{tr}})|$ can be regarded as the expected generalization error of the ECE, and eCMI is the dominant term of the generalization gap. Therefore, if the trained model has a sufficiently low eCMI, it achieves good generalization performance in terms of the ECE. The behavior of eCMI clearly affects the convergence rate of this bound, which is discussed in Section 4.2. Moreover, our bound and eCMI are numerically evaluable and we confirm that our bound is numerically nonvacuous (see Section 6). We also show the application of our bound in the setting of recalibration in Section 4.3.

In Appendix E, we derive the binning bias under the training data similar to Theorem 3. By combining this result with Theorem 4, we obtain the following generalization error bound for the TCE.

**Theorem 5** (Generalization error bound of the TCE). *Under the CMI setting and under Assumptions 1 and 2, we have*

$$\mathbb{E}_{R,S_{\text{tr}}}\text{Bias}_{\text{tot}}(f_W, S_{\text{tr}}) \leq \begin{cases} \frac{1+L}{B} + \sqrt{\frac{8(\text{eCMI}(\tilde{l}) + B\log 2)}{n}} & \text{(for UWB)}, \\ \frac{1+L}{B} + \sqrt{\frac{8(\text{eCMI}(\tilde{l}) + B\log 2)}{n}} + (1+L)\sqrt{\frac{2(\text{fCMI} + B\log 2)}{n}} & \text{(for UMB)}. \end{cases}$$
(16)

*In the above,* $\text{eCMI}(\tilde{l})$ *is defined as Eq. (15) and*

$$\text{fCMI} := I(f_{\mathcal{A}(\tilde{Z}_U, R)}(\tilde{X}); U|\tilde{Z}),$$
(17)

*where* $\tilde{x}$ *denotes the* $n \times 2$ *matrix obtained by projecting each element of* $\tilde{z}$, *and* $f_{\mathcal{A}(\tilde{Z}_U, R)}(\tilde{X})$ *is the* $n \times 2$ *matrix calculated by the elementwise application of* $f_{\mathcal{A}(\tilde{Z}_U, R)}(\cdot)$ *to* $\tilde{x}$.

When comparing Eq. (12) with the above results, it is observed that an additional bias, eCMI (including fCMI in UMB), derived from training data arises. This implies that the trained model shows a low TCE when it sufficiently reduces these additional biases and achieves a small ECE. From a practical viewpoint, this implies that our bound can potentially be used as a theoretical guarantee for some recent training algorithms, which directly control the ECE under the training dataset [23, 29, 38]. Our theory might guarantee the ECE under test dataset for them.

Another interesting implication from our bounds is that we can derive the optimal bin size to minimize the upper bound in Theorem 5. If $\text{eCMI}(\tilde{l})$ and fCMI are sufficiently small compared with $n$, for example, $\mathcal{O}(\log n)$ (we discuss this in Section 4.2), then, the optimal bin size can be derived as $B = \mathcal{O}(n^{1/3})$ by minimizing Eq. (16) w.r.t. $B$. Such an optimal $B$ leads to

$$\mathbb{E}_{R,S_{\text{tr}}}\text{Bias}_{\text{tot}}(f_w, S_{\text{tr}}) = \mathcal{O}(\log n / n^{1/3}). \tag{18}$$

According to Eq. (13) and the above result, we can anticipate that $\mathbb{E}_{R,S_{\text{tr}}}\text{Bias}_{\text{tot}}(f_w, S_{\text{tr}})$ is much smaller than $\mathbb{E}_{S_{\text{te}}}\text{Bias}_{\text{tot}}(f_W, S_{\text{te}})$ because the number of training data is often much larger than that of test data ($n \gg n_{\text{te}}$). This implies that if the model generalizes well, evaluating the ECE using the training dataset may better reduce the total bias than that using test dataset. Although proposing such a new TCE estimation method is beyond the scope of this paper, this represents an important direction for future research.

## 4.2 On the behavior of eCMI and the order of total bias on metric entropy

In this section, we analyze how additional biases, i.e., $\text{eCMI}(\tilde{l})$ and fCMI, behave. The initial observation is that the following relation holds [17]: $\text{eCMI}(\tilde{l}) \leq \text{fCMI} \leq I(W; S)$. Furthermore, we can see that $I(W; S) = \mathcal{O}(\log n)$ under certain constrained conditions, such as the conditionally i.i.d. setting when $f_w(x)$ is the underlying probability model for $p(y|x; w)$ with $\mathcal{W}$ being compact and holding appropriate smoothness assumptions [14, 44]. Furthermore, fCMI can be upper bounded when the algorithms satisfy the various notions of stability [34]. For example, differential private algorithms and stochastic gradient Langevin dynamics (SGLD) [41] algorithms are included in this argument. A more detailed discussion can be found in Appendix F.4.

These arguments, however, hold true only for specific models and algorithms. Therefore, we extend Theorem 5 by utilizing the concept of *metric entropy* to overcome this issue.

**Theorem 6** (Metric entropy). *Let $\mathcal{F}$ be the function class $f_w$ belongs to. Suppose that $\mathcal{F}$ with the metric $\|\cdot\|_\infty$ has the metric entropy, $\log \mathcal{N}(\mathcal{F}, \|\cdot\|_\infty, \delta)$, with the parameter $\delta\ (> 0)$. That is, there exists a set of functions $\mathcal{F}_\delta := \{f_1, \ldots, f_{\mathcal{N}(\mathcal{F}, \|\cdot\|_\infty, \delta)}\}$ that consists of $\delta$-cover of $\mathcal{F}$. Then, under the CMI setting and under Assumptions 1 and 2, for any $\delta \in (0, 1/B]$ and for UWB, we have*

$$\mathbb{E}_{R,S_{\text{tr}}}\text{Bias}_{\text{tot}}(f_W, S_{\text{tr}}) \leq \frac{1+L}{B} + (2+L)\delta + \sqrt{\frac{8B \log 2\mathcal{N}(\mathcal{F}, \|\cdot\|_\infty, \delta/B)}{n}}. \tag{19}$$

See Appendix E.3 for the proof. Theorem 6 connects the IT-based bound to the uniform convergence theory. With this result, we can discuss the optimal number of bins across a broad spectrum of models. For example, we can obtain $\mathcal{N}(\mathcal{F}, \|\cdot\|_\infty, \delta) \asymp \left(\frac{L_0}{\delta}\right)^d$ when $f_w$ is a $d$-dimensional parametric function that is $L_0$-Lipschitz continuous ($L_0 > 0$) [37], leading to the following upper bound:

$$\mathbb{E}_{R,S_{\text{tr}}}\text{Bias}_{\text{tot}}(f_W, S_{\text{tr}}) \lesssim \frac{3+2L}{B} + \sqrt{\frac{8dB \log(2L_0 B^2)}{n}}, \tag{20}$$

where we set $\delta = \mathcal{O}(1/B)$. This bound is minimized when $B = \mathcal{O}(n^{1/3})$, resulting in a bias of $\mathcal{O}(\log n / n^{1/3})$, which is consitent with Eq. (18).

The drawback of the above bound is that it depends on the model's dimensionality explicitly, making them unsuitable for large models such as neural networks. To address this issue, we can use the different combinatorial properties, such as the fat-shattering dimension. See Appendix E.3 for the detail.

## 4.3 Generalized error analysis on recalibration and bias due to reuse of training data

As an application of our generalization error bound, we analyze recalibration using a post-hoc recalibration function, which is used when the trained model is not well calibrated. We focus on the

recalibration using UMB with recalibration data [35, 12]. In this setting, we first split overall data into the training, recalibration, and test datasets (see Appendix B for details of this splitting strategy). After training $f_w$ using the training dataset, we construct the recalibrated function $h$ using the recalibration dataset $S_{\mathrm{re}}$ as

$$h_{\mathcal{I},S_{\mathrm{re}}}(x) := \sum_{i=1}^{B} \bar{y}_{i,S_{\mathrm{re}}} \cdot \mathbb{1}_{f_w(x) \in I_i}, \tag{21}$$

where $\bar{y}_{i,S_{\mathrm{re}}}$ is the empirical mean of $\{y_m\}_{m=1}^{n_{\mathrm{re}}} \in S_{\mathrm{re}}$ in the $i$-th bin defined as in Eq. (2) and $n_{\mathrm{re}}$ is the number of the recalibration dataset. In short, Eq. (21) provides an estimator of the conditional expectation of $Y$ given $f_w(x)$ by setting $S_e = S_{\mathrm{re}}$. Gupta & Ramdas [12] clarified that the statistical bias of Eq. (21) is given by $\mathbb{E}_{S_{\mathrm{re}}}\mathrm{TCE}(h_{\mathcal{I},S_{\mathrm{re}}}) = \mathbb{E}[|\mathbb{E}[Y|h_{\mathcal{I},S_{\mathrm{re}}}(X)] - h_{\mathcal{I},S_{\mathrm{re}}}(X)|] = \mathcal{O}_p(\sqrt{B \log B/|S_{\mathrm{re}}|})$. Since we need to split the overall data into three datasets, this approach could be sample-inefficient and could result in a very loose bound. Although reusing the training dataset may solve this problem to some extent, it has been suggested that this method may cause performance degradation due to overfitting [24, 12].

Our contribution here is quantifying the bias caused by overfitting due to the reuse of training data by utilizing our generalization error analysis in Section 4.1 as follows.

**Theorem 7** (Recalibration reusing the training dataset). *Replacing $S_{\mathrm{re}}$ with $S_{\mathrm{tr}}$ in Eq. (21), under the CMI setting and under Assumptions 1 and 2, we have*

$$\mathbb{E}_{R,S_{\mathrm{tr}}}\mathrm{TCE}(h_{\mathcal{I},S_{\mathrm{tr}}}) = \mathbb{E}_{R,S_{\mathrm{tr}}}\mathbb{E}[|\mathbb{E}[Y|h_{\mathcal{I},S_{\mathrm{tr}}}(X)] - h_{\mathcal{I},S_{\mathrm{tr}}}(X)] \leq 2\sqrt{\frac{2(\mathrm{fCMI} + B\log 2)}{n}},$$

*where* fCMI *is defined in Eq. (17).*

The complete proof is provided in Appendix E.4. In the above, fCMI corresponds to the additional bias caused by overfitting due to the reuse of $S_{\mathrm{tr}}$. This indicates that reusing $S_{\mathrm{tr}}$ does not negatively affect the order of the bias if fCMI is smaller than other terms, as discussed in Sections 4.1 and 4.2. Since the size of $S_{\mathrm{tr}}$ is much larger than that of $S_{\mathrm{re}}$, the recalibration function $h_{\mathcal{I},S_{\mathrm{tr}}}$ may exhibit a much smaller bias compared to $h_{\mathcal{I},S_{\mathrm{re}}}$. We investigate this possibility numerically in Section 6 by using the tighter version of Theorem 7 provided in Appendix E.4 (Corollary 4).

## 5  Related work

We have presented the results of our analyses of the total bias in the ECE and the generalization error for both the ECE and the TCE. Existing studies have primarily focused on the statistical bias, with little attention given to the binning bias. Gupta et al. [13] and Gupta & Ramdas [12] examined the statistical bias associated with UMB, but they did not address the binning bias as we did. In contrast, Kumar et al. [24] studied the binning bias but did not specify how this bias depends on $n$ and $B$. Moreover, most analyses have concentrated on UMB and UWB has not been thoroughly analyzed. As outlined in the proof of Theorem 2, our approach allows us to analyze UWB even in cases where some bins do not have any data points by employing our reformulation and concentration inequality. It is important to note that Roelofs et al. [31] studied the numerical behavior of the total bias in some practical models, whereas we focus on the theoretical aspect of the total bias. Recently, Sun et al. [35] have derived the optimal number of bins under the recalibration with UMB. Compared with this, we derived the optimal number of bins for UMB and UWB without recalibration under a similar Lipschitz assumption. This leads to the discussion of estimating the TCE from the nonparametric estimation. An additional discussion is summarized in Appendix F.

We have extended the existing eCMI bound [34, 15, 17, 40], which is used for analyzing generalization performance in terms of prediction accuracy, to calibration analysis. In addition, whereas existing eCMI bounds numerically evaluated eCMI and fCMI for *discrete* random variables such as zero-one loss, our analysis is conducted on *continuous* random variables as shown in Eq. (15). We show in the next section that our bounds are still nonvacuous even for the continuous random variables. The IT analysis was also utilized by Russo & Zou [33] to study the bias caused by data reuse. Our analysis can be seen as an extension of this approach to the ECE and recalibration.

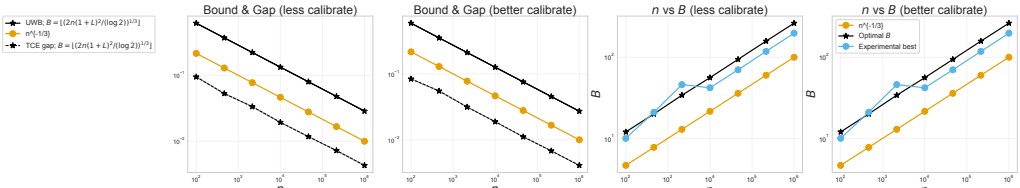

Figure 1: Behavior of the upper bound in Eq. (12) as $n$ increases when UWB is used. The following two terms: *less calibrate* and *better calibrate* refer to $\beta = (0.5, -1.5)$ and $\beta = (0.2, -1.9)$, respectively, where the former setting produces a worse value of the TCE estimator.

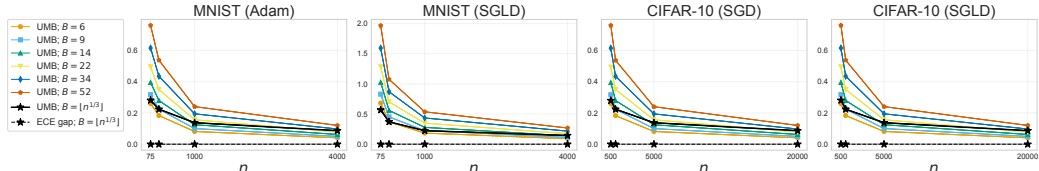

Figure 2: Behavior of the upper bound in Eq. (14) for various $B$ as $n$ increases (mean $\pm$ std.). For clarity, only the results using UMB are shown. The ECE gap is shown for $B = \lfloor n^{1/3} \rfloor$ since the change in $B$ did not result in significant differences. We refer to Figure 5 in Appendix H.3 for a detailed analysis of the relationship between (log-scaled) ECE gap values and bound values across different bin settings.

## 6  Experiments

In this section, we present experimental results validating our bounds (Section 6.1) and the additional bias arising from reusing the training dataset for recalibration (Section 6.2).

### 6.1  Verification of our bounds

In this section, we empirically validate our theoretical findings in Eq. (12), the nonvacuous nature of our bounds in Eq. (14), and confirm the efficiency of the optimal number of bins as discussed in Section 4.1.

**Experiments on synthetic datasets:**  We first conducted simple experiments on synthetic datasets following Zhang et al. [50]. In this experiment, we assume the distribution of $Y$ as $P(Y = 1) = P(Y = 0) = \frac{1}{2}$, and we adopt $f_w(x) = P(Y = 1|X = x) = 1/(1 + \exp(-\beta_0 - \beta_1 x))$ as the prediction model, where $w = \{\beta_0, \beta_1\}$ are parameters. Under these settings, we can calculate the closed-form of $\mathbb{E}[Y|f_w(X)]$ in Eq. (1), which allows us to estimate the TCE through Monte Carlo integration. Next, we empirically evaluated the *TCE gap*, which is the empirical estimator of $\mathbb{E}_{S_{\text{te}}}\text{Bias}_{\text{tot}}(f_w, S_{\text{te}})$, by calculating the difference between the TCE estimator and the ECE using UWB. Here, the optimal $B$ that minimizes the upper bound of Eq. (14) is $B = \lfloor 2n(1 + L)^2/(\log 2)^{1/3} \rfloor$, where $L$ is estimated to be the maximum value of the gradient of the closed-form of $\mathbb{E}[Y|f_w(X)]$. We provide the details of the experimental settings in Appendix G.1.

Figure 1 shows the results. The two leftmost figures show that the TCE gap is $\mathcal{O}(n^{-1/3})$ when our optimal $B$ is used. The other two figures show that the number of bins achieving the smallest upper bound is closest to the optimal $B$ and its order is $O(n^{-1/3})$, where the candidates for $B$ are set as $\{n^{-1/2}, n^{-1/3}, n^{-1/4}, 2n^{-1/2}, 2n^{-1/3}, 2n^{-1/4}, 3n^{-1/2}, 3n^{-1/3}, 3n^{-1/4}\}$. These observations show the validity of our theoretical findings through Corollary 1.

**Experiments on image datasets:**  We further conducted two binary classification tasks on MNIST [25] using a convolutional neural network (CNN) and on CIFAR-10 [21] using ResNet. These models were trained using SGD with momentum for ResNet, Adam for CNN, and SGLD for both, following the strategy of Hellström & Durisi [17]. The details of our experimental settings are summarized in Appendix G.2. We initially evaluated the sum of the right-hand side terms of

Table 1: Comparison of our method with existing recalibration in terms of the ECE gap and its upper bound in Theorem 7 (mean $\pm$ std.). Lower values are better. We adopted $B = \lfloor n^{1/3} \rfloor$. The bound values for the existing recalibration method originate from Corollary 4 in Appendix E.5. Here, we report the ECE gap as **TCE** because $\mathbb{E}_{R,S_{re}}\mathbb{E}[|\mathbb{E}[Y|h_{\mathcal{I},S_{re}}(X)] - h_{\mathcal{I},S_{re}}(X)] = \mathbb{E}_{R,S_{re},S_{te}}\mathrm{ECE}(h_{\mathcal{I},S_{re}}, S_{te})$ (existing recalibration methods) $\mathbb{E}_{R,S_{tr}}\mathbb{E}[|\mathbb{E}[Y|h_{\mathcal{I},S_{tr}}(X)] - h_{\mathcal{I},S_{tr}}(X)] = \mathbb{E}_{R,S_{tr},S_{te}}\mathrm{ECE}(h_{\mathcal{I},S_{tr}}, S_{te})$ (our recalibration method) from the definition of recalibration.

| Dataset | Optimizer | Methods | TCE | Bound value |
|---|---|---|---|---|
| MNIST ($n = 4000$) | Adam | Recalib. | $.0085 \pm .0016$ | $.8475$ |
| | | Our recalib. | $\mathbf{.0058 \pm .0026}$ | $\mathbf{.1444 \pm .0000}$ |
| | SGLD | Recalib. | $.0101 \pm .0025$ | $.8475$ |
| | | Our recalib. | $\mathbf{.0072 \pm .0007}$ | $\mathbf{.1444 \pm .0000}$ |
| CIFAR-10 ($n = 20000$) | SGD | Recalib. | $\mathbf{.0139 \pm .0010}$ | $1.455$ |
| | | Our recalib. | $.0197 \pm .0044$ | $\mathbf{.0865 \pm .0000}$ |
| | SGLD | Recalib. | $.0109 \pm .0012$ | $1.455$ |
| | | Our recalib. | $\mathbf{.0089 \pm .0006}$ | $\mathbf{.0865 \pm .0000}$ |

Eq. (14) and the ECE estimated using the training dataset, aiming to ascertain whether the disparity from the ECE estimated using the test dataset was adequately minimal. We call this disparity as the *ECE gap*, which is the estimator of $\mathbb{E}_{R,S_{tr},S_{te}}[|\mathrm{ECE}(f_W, S_{te}) - \mathrm{ECE}(f_W, S_{tr})|]$. We show the results obtained when using UMB in Figure 2. These results show that our bound value becomes less than 1 with an appropriate setting of $B$. We also observed that the bound values decrease with $n$, whereas these values sometimes become vacuous for small $n$ when $B$ is large. Adjusting $B$ could pose challenges; however, a notable trend towards acquiring relatively stable nonvacuous bounds can be observed when adopting $B = \lfloor n^{1/3} \rfloor$, even though this is the optimal choice only at the upper bound of TCE, as discussed in Theorems 5 and 6 in Sections 4.1 and 4.2. Similar results are obtained when using UWB (see Figure 3 in Appendix H).

### 6.2 Confirming additional bias due to reusing training dataset in recalibration

In this section, we empirically confirm the efficiency of the method when using the complete training dataset for recalibration, referred to here as the *reusing method*. Table 1 illustrates that the reusing method reduces the statistical bias of the ECE more effectively than existing methods using independently created recalibration datasets ($n_{re} = 100$). We also compared the tighter version of our bound in Theorem 7, Corollary 4, with the bound of the existing recalibration methods presented in Corollary 5 of Appendix E.5. Moreover, our bound values are lower than those for the existing recalibration methods on the test dataset. These results suggest that reusing training data could be beneficial if the trained model generalizes well and eCMI is sufficiently small, as discussed in Section 4.3.

## 7 Conclusion and limitations

We provided the first comprehensive analysis of the bias associated with the ECE when using the test and training datasets. This leads to the derivation of the optimal bin size to minimize the total bias. Numerical experiments show that our upper bound of the bias is nonvacuous for deep learning models thanks to the IT generalization error analysis. Despite rigorous analysis, our analysis still has limitations. Firstly, we focus on the binary classification; thus, the extension of our analysis to the multiclass classification setting is an important future direction. However, the application of our analytical techniques to this setting seems not clear. Additionally, our analysis cannot be applied to the higher-order TCE, in which we use the $p$-th norm in Eq. (1). These limitations should be addressed in future work to develop a more principled understanding of uncertainty.

## Acknowledgments and Disclosure of Funding

We sincerely appreciate the anonymous reviewers for their insightful feedback. FF was supported by JSPS KAKENHI Grant Number JP23K16948. FF was supported by JST, PRESTO Grant Number JPMJPR22C8, Japan. MF was supported by RIKEN Special Postdoctoral Researcher Program. MF was supported by JST, ACT-X Grant Number JPMJAX210K, Japan.

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

## A   Remarks about the order expressions

Let $f, g : \mathbb{R} \to \mathbb{R}$. We say $f(x) \asymp g(x)$ when there exist positive constants $a$, $b$ and $n_0$ such that $\forall n > n_0$, we have $ag(n) \leq f(x) \leq bg(x)$ holds. Moreover $f(n) \lesssim g(n)$ means that there exists a positive constant $c$ and $n_0$ such that $\forall n > n_0$, we have $|f(n)| \leq cg(n)$ holds. This is equivalent to $f(n) = \mathcal{O}(g(n))$.

## B   The detailed explanation of how the data is split and used in our bounds

Here we remark how the data is prepared and used in our analysis.

### B.1   The binning ECE and its evaluation

In the standard supervised learning settings assume that $\mathrm{N_{all}}$ data is obtained i.i.d from data generating distribution $\mathcal{D}$. We express this as $S_{\mathrm{all}} = \{(x_i, y_i)\}_{i=1}^{\mathrm{N_{all}}}$. Then the dataset is divided to

$$S_{\mathrm{all}} = S_{\mathrm{tr}} \cup S_{\mathrm{te}}, \quad S_{\mathrm{tr}} \cap S_{\mathrm{te}} = \phi,$$

where $S_{\mathrm{tr}}$ is the training data which is $n$ data points and $S_{\mathrm{te}}$ is the test data points which is $n_{\mathrm{te}}$ data points. Thus $\mathrm{N_{all}} = n + n_{\mathrm{te}}$.

#### B.1.1   Evaluation of the ECE under the test dataset in Section 3

Here we discuss the evaluation of $\mathrm{ECE}(f_w, S_{\mathrm{te}})$, which uses $S_{\mathrm{te}}$ to calculate the ECE in Section 3. This is the most common common approach in practice. We remark that as for the UWB, since we do not use $S_{\mathrm{te}}$ when preparing bins, those $S_{\mathrm{te}}$ are i.i.d. inside each bin.

As for UMB, the situation is a bit complicated. Since we use $S_{\mathrm{te}}$ to construct bins, it seems that $S_{\mathrm{te}}$ are no more i.i.d inside each bin. Surprisingly, Gupta & Ramdas [12] have shown that the samples allocated in each bin are i.i.d. under UMB method. So using the same $S_{\mathrm{te}}$ for constructing bins and evaluation of the binning ECE does not result in a large bias.

In these ways, we can calculate $\mathrm{ECE}(f_w, S_{\mathrm{te}})$. We also remark that the training samples $S_{\mathrm{tr}}$ are only used to learn the parameter $W$.

#### B.1.2   Evaluation of the ECE under the training dataset in Section 4

ere we discuss the evaluation of $\mathrm{ECE}(f_w, S_{\mathrm{tr}})$, which uses $S_{\mathrm{tr}}$ to calculate the ECE in Section 3. Thus, we use the training dataset $S_{\mathrm{tr}}$ for learning $W$ and calculating ECE.

We need to carefully consider how the data is used when considering the result of the UMB in Theorem 4. This theorem provides us the generalization guarantee of the ECE between $\mathrm{ECE}(f_w, S_{\mathrm{tr}})$ and $\mathrm{ECE}(f_w, S_{\mathrm{te}})$. As for $\mathrm{ECE}(f_w, S_{\mathrm{tr}})$ using UMB, we first train $f_W$ with $S_{\mathrm{tr}}$ and construct bins with $S_{\mathrm{tr}}$ and calculate the empirical mean of each bin with $S_{\mathrm{tr}}$. On the other hand, when calculating $\mathrm{ECE}(f_w, S_{\mathrm{te}})$, we calculate the empirical mean of each bin with $S_{\mathrm{te}}$ and we use the same bins with $\mathrm{ECE}(f_w, S_{\mathrm{tr}})$, so bins are constructed using $S_{\mathrm{tr}}$ in Theorem 4. In this sense, Theorem 4 provides us with the generalization gap, where we regard that the bins constructed with the training dataset are regarded as part of our trained model in our theoretical analysis.

### B.2   The recalibration in Section 4.3

When the recalibration is performed, we further split the test data into

$$S_{\mathrm{all}} = S_{\mathrm{tr}} \cup S_{\mathrm{re}} \cup S_{\mathrm{te}}, \quad \text{any common part sets are empty},$$

where $S_{\mathrm{tr}}$ is the training datasets used for learning $W$, and $S_{\mathrm{re}}$ is the dataset used for the recalibration. The most widely used approach is the UMB-based recalibration. First, we construct bins following the UMB approach using $S_{\mathrm{re}}$. Then let us express $S_{\mathrm{re}} = \{(x_i, y_i)\}_{i=1}^{n_{\mathrm{re}}}$. The recalibrated function

$$h_{\mathcal{I}, S_{\mathrm{re}}}(x) = \sum_{i=1}^{B} \hat{\mu}_{i, S_{\mathrm{re}}} \cdot \mathbb{1}_{f_w(x) \in I_i}, \quad \hat{\mu}_{i, S_{\mathrm{re}}} := \frac{\sum_{m=1}^{n_{\mathrm{re}}} y_m \cdot \mathbb{1}_{f_w(x_m) \in I_i}}{\sum_{m=1}^{n_{\mathrm{re}}} \mathbb{1}_{f_w(x_m) \in I_i}}. \tag{22}$$

Then $S_{\text{te}}$ is used for evaluating the ECE or test accuracy.

However, since preparing both $S_{\text{te}}$ and $S_{\text{re}}$ is sample inefficient, our idea is reusing training sample $S_{\text{tr}}$ with size $n$ even for the recalibration. In our setting, we split the data

$$S_{\text{all}} = S_{\text{tr}} \cup S_{\text{te}}, \quad S \cap S_{\text{te}} = \phi,$$

and we construct bins following UMB approach using $S_{\text{tr}}$ and calculate recalibration function by using $S_{\text{tr}}$

$$h_{\mathcal{I}, S_{\text{tr}}}(x) = \sum_{i=1}^{B} \hat{\mu}_{i, S_{\text{tr}}} \cdot \mathbb{1}_{f_w(x) \in I_i}, \quad \hat{\mu}_{i, S_{\text{tr}}} := \frac{\sum_{m=1}^{n} y_m \cdot \mathbb{1}_{f(x_m) \in I_i}}{\sum_{m=1}^{n} \mathbb{1}_{f(x_m) \in I_i}}. \tag{23}$$

We then finally evaluate the ECE using $S_{\text{te}}$. So our approach is more sample-efficient.

## C   Auxiliary lemma and facts

Here we introduce auxiliary lemma and facts, which we will use repeatedly in our proofs. In this section, we express $f_w$ as $f$ to simplify the notation.

### C.1   Binning and bias

First, from the definition of ECE in Eq. (2), we can immediately reformulate it as

$$\text{ECE}(f, S_e) := \sum_{i=1}^{B} |\mathbb{E}_{(X,Y) \sim \hat{S}_e}(Y - f(X)) \cdot \mathbb{1}_{f_w(X) \in I_i}|, \tag{24}$$

where $\hat{S}_e$ is the empirical distribution of $S_e$.

Next we introduce the binned function of $f$ as $f_{\mathcal{I}}$, which is the conditional expectation given bins:

$$f_{\mathcal{I}}(x) := \sum_{i=1}^{B} f_{I_i}(x) \cdot \mathbb{1}_{f(x) \in I_i} = \sum_{i=1}^{B} \mathbb{E}[f(X)|f(X) \in I_i] \cdot \mathbb{1}_{f(x) \in I_i}, \tag{25}$$

$$f_{I_i}(x) = \mathbb{E}[f(X)|f(x) \in I_i]. \tag{26}$$

The following relation holds:

**Lemma 1.** *We define the test-binned risk as*

$$\text{ECE}^{\text{Bin}}(f) := \sum_{i=1}^{B} |\mathbb{E}_{(X,Y) \sim \mathcal{D}}(Y - f(X)) \cdot \mathbb{1}_{f(X) \in I_i}|, \tag{27}$$

*then we have*

$$\text{ECE}^{\text{Bin}}(f) = \text{TCE}(f_{\mathcal{I}}), \tag{28}$$

*where* $\text{TCE}(f_{\mathcal{I}})$ *means the TCE of the function* $f_{\mathcal{I}}$.

*Proof.* By definition, we have

$$\text{ECE}^{\text{Bin}}(f) = \sum_{i=1}^{B} |\mathbb{E}_{(X,Y) \sim \mathcal{D}}[(Y - f(X)) \cdot \mathbb{1}_{f(X) \in I_i}]|$$

$$= \sum_{i=1}^{B} P(f(X) \in I_i)\mathbb{E}|\mathbb{E}[Y|f(X) \in I_i] - \mathbb{E}[f(X)|f(X) \in I_i]|, \tag{29}$$

where we used the definition of the conditional expectation. On the other hand, We have

$$
\begin{aligned}
\mathrm{TCE}(f_{\mathcal{I}}) &= \mathbb{E}|\mathbb{E}[Y|f_{\mathcal{I}}(x)] - f_{\mathcal{I}}(x)| \\
&= \sum_{i=1}^{B} \mathbb{E}\left[|\mathbb{E}[Y|f_{\mathcal{I}}(x)] - f_{\mathcal{I}}(x)| \cdot \mathbb{1}_{f_{\mathcal{I}}(X)\in I_i}\right] \\
&= \sum_{i=1}^{B} P(f_{\mathcal{I}}(x) \in I_i)\mathbb{E}\left[|\mathbb{E}[Y|f_{\mathcal{I}}(x)] - f_{\mathcal{I}}(x)|f_{\mathcal{I}}(X) \in I_i\right] \\
&= \sum_{i=1}^{B} P(f(X) \in I_i)\mathbb{E}|\mathbb{E}[Y|f(X) \in I_i] - \mathbb{E}[f(X) \in I_i]|, \quad (30)
\end{aligned}
$$

where we used the tower property. This concludes the proof. $\qquad\square$

Thus, we can transform the loss and ECEs by Eqs. (24) and (27)

## C.2 Useful inequalities

Here we introduce lemmas, which we will use repeatedly in our proofs.

**Lemma 2** (Corollary 3.2 in Boucheron et al. [3]). *We say that a function $f : \mathcal{X} \to \mathbb{R}$ has the bounded differences property if for some nonnegative constants $c_1, \ldots, c_n$,*

$$
\sup_{x_1,\ldots,x_n,x_i'\in\mathcal{X}} |f(x_1,\ldots,x_n) - f(x_1,\ldots,x_{i-1},x_i',x_{i+1},\ldots,x_n)| \le c_i, \quad 1 \le i \le n. \quad (31)
$$

*If $X_1, \ldots, X_n$ are independent random variables taking values in $\mathcal{X}$, we define the real-valued random variable as*

$$
Z = f(X_1, \ldots, X_n). \quad (32)
$$

*If $f$ has the bounded difference property with constants $c_1, \ldots, c_n$, then we have*

$$
\mathrm{Var}[Z] \le \frac{1}{4}\sum_{i=1}^{n} c_i^2. \quad (33)
$$

Combining this lemma with Holder inequality, we have the following relation,

$$
\mathbb{E}|Z - \mathbb{E}[Z]| \le \sqrt{\frac{1}{4}\sum_{i=1}^{n} c_i^2}. \quad (34)
$$

We often consider the case where $-1 \le f \le 1$. This implies that $c_i = 2$ for all $i$. Then

$$
\mathbb{E}|Z - \mathbb{E}[Z]| \le 1 \quad (35)
$$

Another situation is that given a function $g : \mathcal{X} \to [-1,1]$, consider $f(x_1,\ldots,x_n) :=$ $\frac{1}{n}\sum_{i=1}^{n} f(x_i).$ , where $f$ corresponds to the empirical mean of some function $g$. This implies that $c_i = 2/n$ for all $i$. Then

$$
\mathbb{E}|f - \mathbb{E}[f]| \le \frac{1}{\sqrt{n}}. \quad (36)
$$

**Lemma 3** (Used in the proof of McDiarmid's inequality). *Given a bounded difference function $f$, for any $t \in \mathbb{R}$, we have*

$$
\mathbb{E}\left[e^{t(f(X_1,\ldots,X_n)-\mathbb{E}[f(X_1,\ldots,X_n)])}\right] \le e^{\frac{t^2}{8}\sum_{i=1}^{n} c_i^2}. \quad (37)
$$

# D  Proofs of Section 3

## D.1  Proofs of Theorem 2

*Proof.* Here we express the samples in $S_{\text{te}}$ as $\{Z'_m\}_{m=1}^{n_{\text{te}}} = \{(X'_m, Y'_m)\}$.

As for the first inequality, it is the consequence of the Jensen inequality, as follows;

$$
\begin{aligned}
\text{TCE}(f_{\mathcal{I}}) &= \sum_{i=1}^{B} \left| \mathbb{E}_{Z'=(X',Y')} \left[ (Y' - f_W(X') \cdot \mathbb{1}_{f_W(X') \in I_i}] \right] \right| \\
&= \sum_{i=1}^{B} \left| \mathbb{E}_{\{Z'_m=(X'_m,Y'_m)\}_{m=1}^{n_{\text{te}}}} \frac{1}{n_{\text{te}}} \sum_{m=1}^{n_{\text{te}}} (Y'_m - f_W(X'_m)) \cdot \mathbb{1}_{f_W(X'_m) \in I_i} \right| \\
&\leq \mathbb{E}_{\{Z'_m=(X'_m,Y'_m)\}_{m=1}^{n_{\text{te}}}} \sum_{i=1}^{B} \left| \frac{1}{n_{\text{te}}} \sum_{m=1}^{n_{\text{te}}} (Y'_m - f_W(X'_m)) \cdot \mathbb{1}_{f_W(X'_m) \in I_i} \right| \\
&= \mathbb{E}_{S_{\text{te}}} \text{ECE}(f_W, S_{\text{te}}),
\end{aligned}
\tag{38}
$$

where we used the Jensen inequality.

As for the second inequality, we separately prove UWB and UMB.

### D.1.1  Proof for the UWB

We start from UWB. Conditioned on $W$, using the relation between the loss and ECEs by Eqs. (24) and (27), we have

$$
\begin{aligned}
&|\text{TCE}(f_{\mathcal{I}}) - \text{ECE}(f_W, S_{\text{te}})| \\
&= \left| \sum_{i=1}^{B} \left| \mathbb{E}_{Z''=(X'',Y'')} \left[ (Y'' - f_W(X'')) \cdot \mathbb{1}_{f_W(X'') \in I_i} \right] \right| - \sum_{i=1}^{B} \left| \frac{1}{n_{\text{te}}} \sum_{m=1}^{n_{\text{te}}} (Y'_m - f_W(X'_m)) \cdot \mathbb{1}_{f_W(X'_m) \in I_i} \right| \right| \\
&\leq \sum_{i=1}^{B} \left| \mathbb{E}_{Z''=(X'',Y'')} \left[ (Y'' - f_W(X'')) \cdot \mathbb{1}_{f_W(X'') \in I_i} \right] - \frac{1}{n_{\text{te}}} \sum_{m=1}^{n_{\text{te}}} (Y'_m - f_W(X'_m)) \cdot \mathbb{1}_{f_W(X'_m) \in I_i} \right| \\
&\leq \sum_{i=1}^{B} \left| \mathbb{E}_{Z''} l_i(Z'') - \frac{1}{n_{\text{te}}} \sum_{m=1}^{n_{\text{te}}} l_i(Z'_m) \right|,
\end{aligned}
\tag{39}
$$

where we used the triangle inequality $||a| - |b|| \leq |a - b|$ for the first inequality and set $l_i(z) = (y - f_W(x)) \cdot \mathbb{1}_{f_W(x) \in I_i}$.

We use the following relation: for the one-dimensional real random variable $X$, by Jensen inequality, we have

$$
t\mathbb{E}[|X|] \leq \log \mathbb{E}[e^{t|X|}].
\tag{40}
$$

Then combining with the above, we have

$$
\mathbb{E}_{S_{\text{te}}} |\text{TCE}(f_{\mathcal{I}}) - \text{ECE}(f_W, S_{\text{te}})| \leq \frac{1}{t} \mathbb{E}_{S_{\text{te}}} e^{t \sum_{i=1}^{B} \left| \mathbb{E}_{Z''} l_i(Z'') - \frac{1}{n_{\text{te}}} \sum_{m=1}^{n_{\text{te}}} l_i(Z'_m) \right|}.
\tag{41}
$$

By setting $g(i, S_{\text{te}}) := \mathbb{E}_{Z''} l_i(Z'') - \frac{1}{n_{\text{te}}} \sum_{m=1}^{n_{\text{te}}} l_i(Z'_m)$, we have

$$\mathbb{E}_{S_{\text{te}}} e^{t \sum_{i=1}^{B} |g(i, S_{\text{te}})|} = \mathbb{E}_{S_{\text{te}}} \prod_{i=1}^{B} e^{t |g(i, S_{\text{te}})|}$$

$$\leq \mathbb{E}_{S_{\text{te}}} \prod_{i=1}^{B} \left( e^{t g(i, S_{\text{te}})} + e^{-t g(i, S_{\text{te}})} \right)$$

$$\leq \mathbb{E}_{S_{\text{te}}} \sum_{v_1, \ldots, v_B = 0, 1} e^{t \sum_{i=1}^{B} (-1)^{v_i} g(i, S_{\text{te}})} \tag{42}$$

$$= \sum_{v_1, \ldots, v_B = 0, 1} \mathbb{E}_{S_{\text{te}}} e^{t \sum_{i=1}^{B} (-1)^{v_i} g(i, S_{\text{te}})}$$

$$= \sum_{v_1, \ldots, v_B = 0, 1} \mathbb{E}_{S_{\text{te}}} e^{t \sum_{i=1}^{B} (-1)^{v_i} \left[ \mathbb{E}_{Z''} l_i(Z'') - \frac{1}{n_{\text{te}}} \sum_{m=1}^{n_{\text{te}}} l_i(Z'_m) \right]}, \tag{43}$$

where $\sum_{v_1, \ldots, v_B = 0, 1}$ is all the combinations that will be generated by expanding $\prod_{i=1}^{B}$ in Eq. (42) and it has $2^B$ combinations.

We would like to upper bound $\mathbb{E}_{S_{\text{te}}} e^{t \sum_{i=1}^{B} (-1)^{v_i} \left[ \mathbb{E}_{Z''} l_i(Z'') - \frac{1}{n_{\text{te}}} \sum_{m=1}^{n_{\text{te}}} l_i(Z'_m) \right]}$ using Lemma 3. For that purpose, here we evaluate $c_i$s of Lemma 3. By focusing on the exponent, we can estimate $c_i$s by

$$\sup_{\{z'_m\}_{m=1}^{n_{\text{te}}}, \tilde{z}_{m'} \in \mathcal{Z}} \sum_{i=1}^{B} t(-1)^{v_i} \cdot \left[ \mathbb{E}_{Z''} l_i(Z'') - \frac{1}{n_{\text{te}}} \sum_{m=1}^{n_{\text{te}}} l_i(z'_m) \right]$$

$$- t(-1)^{v_i} \cdot \left[ \mathbb{E}_{Z''} l_i(Z'') - \frac{1}{n_{\text{te}}} \sum_{m \neq m'}^{n_{\text{te}}} l_i(z'_m) - \frac{1}{n_{\text{te}}} l_i(\tilde{z}_{m'}) \right]$$

$$= \sup_{z'_m, \tilde{z}_{m'} \in \mathcal{Z}} \sum_{i=1}^{B} \frac{t(-1)^{v_i}}{n_{\text{te}}} \cdot [-l_i(z'_{m'}) + l_i(\tilde{z}_{m'})]$$

$$= \sup_{z'_m, \tilde{z}_{m'} \in \mathcal{Z}} \frac{t(-1)^{v_1}}{n_{\text{te}}} \cdot \left( -\left( (y'_{m'} - f_W(x'_{m'})) \cdot \mathbb{1}_{f_W(x'_{m'}) \in I_1} \right) + \left( (\tilde{y}'_{m'} - f_W(\tilde{x}'_{m'})) \cdot \mathbb{1}_{f_W(\tilde{x}'_{m'}) \in I_1} \right) \right) +$$

$$\vdots$$

$$+ \frac{t(-1)^{v_B}}{n_{\text{te}}} \cdot \left( -\left( (y'_{m'} - f_W(x'_{m'})) \cdot \mathbb{1}_{f_W(x'_{m'}) \in I_B} \right) + \left( (\tilde{y}_{m'} - f_W(\tilde{x}_{m'})) \cdot \mathbb{1}_{f_W(\tilde{x}_{m'}) \in I_B} \right) \right) \tag{44}$$

$$\leq \frac{2t}{n_{\text{te}}}, \tag{45}$$

where the last inequality is derived as follows; Here by definition of the binning, each data point is allocated to a single bin. This means that for the input $x'_{m'}$, one of $\{\mathbb{1}_{f_W(x'_{m'}) \in I_i}\}_{i=1}^{B}$ is not zero. We refer to such index as $b$. Then $\mathbb{1}_{f_W(x'_{m'}) \in I_b} \neq 0$ at the $b$-th bin and $\mathbb{1}_{f_w(x'_{m'}) \in I_{b' \neq b}} = 0$ holds. A similar argument holds for the input $\tilde{x}'_{m'}$ and we refer to the index that the indicator function is not zero as $\tilde{b}$, which implies $\mathbb{1}_{f_W(\tilde{x}'_{m'}) \in I_{\tilde{b}}} \neq 0$ and $\mathbb{1}_{f_w(x'_{m'}) \in I_{b' \neq \tilde{b}}} = 0$. Note that such $b$ and $\tilde{b}$ can be equal and can be different. Thus, although there are $2B$ indicator functions in Eq. (44), at most only two indicator functions are not zero.

Combined with the fact that $|y'_{m'} - f_w(x'_{m'})| \leq 1$, we obtain Eq. (45). Note that by Assumption 1, $\{f_w(x_m)\}_{m=1}^{n_{\text{te}}}$ in $x_m \in S_{\text{te}}$ takes the distinct values almost surely and in the above discussion, we do not consider the case when $b/B = f_w(x_m)$ for some $b$ holds, which means that the predicted probability is just the value of the boundary of bins.

When we do not assume that Assumption 1, there may be a possibility that $b/B = f_w(x_m)$ for some $b$ holds, which means that the predicted probability is just the value of the boundary of bins. Then,

at most only four indicator functions are not zero. This results in a worse bound

$$
\sup_{\{z_m\}_{m=1}^{n_{\text{te}}}, \tilde{z}_m \in \mathcal{Z}} \sum_{i=1}^{B} t(-1)^{v_i} \cdot \left[ \mathbb{E}_{Z'} l_i(Z') - \frac{1}{n_{\text{te}}} \sum_{m=1}^{n_{\text{te}}} l_i(z'_m) \right]
$$

$$
- t(-1)^{v_i} \cdot \left[ \mathbb{E}_{Z'} l_i(Z') - \frac{1}{n_{\text{te}}} \sum_{m \neq m'}^{n_{\text{te}}} l_i(z'_m) - \frac{1}{n_{\text{te}}} l_i(\tilde{z}_{m'}) \right]
$$

$$
\leq \frac{4t}{n_{\text{te}}}. \tag{46}
$$

Combined with Lemma 3, we have that

$$
\mathbb{E}_{S_{\text{te}}} e^{t \sum_{i=1}^{B} |g(i, Z'_m)|} \leq \sum_{v_1, \ldots, v_B = 0,1} \prod_{m=1}^{n_{\text{te}}} \mathbb{E}_{Z'_m} e^{t \sum_{i=1}^{B} (-1)^{v_i} \left[ \mathbb{E}_{Z''} l_i(Z'') - \frac{1}{n_{\text{te}}} \sum_{m=1}^{n_{\text{te}}} l_i(Z'_m) \right]}
$$

$$
\sum_{v_1, \ldots, v_B = 0,1} \prod_{m=1}^{n_{\text{te}}} \mathbb{E}_{Z'_m} e^{t \sum_{i=1}^{B} (-1)^{v_i} \left[ \mathbb{E}_{Z''} l_i(Z'') - \frac{1}{n_{\text{te}}} \sum_{m=1}^{n_{\text{te}}} l_i(Z'_m) \right]}
$$

$$
\leq \sum_{v_1, \ldots, v_B = 0,1} e^{(t^2/8) n_{\text{te}} \left( \frac{2}{n_{\text{te}}} \right)^2}
$$

$$
= 2^B e^{\frac{2t^2}{n_{\text{te}}}}. \tag{47}
$$

Combining this with Eq. (41), we have

$$
\mathbb{E}_{S_{\text{te}}} |\text{TCE}(f_{\mathcal{I}}) - \text{ECE}(f_W, S_{\text{te}})| \leq \frac{1}{t} \log \mathbb{E}_{S_{\text{te}}} e^{t \sum_{i=1}^{B} \left| \mathbb{E}_{Z''} l_i(Z'') - \frac{1}{n_{\text{te}}} \sum_{m=1}^{n_{\text{te}}} l_i(Z'_m) \right|}
$$

$$
\leq \frac{\log 2^B e^{\frac{t^2}{2n_{\text{te}}}}}{t}
$$

$$
= \frac{B \log 2}{t} + \frac{t}{2n_{\text{te}}}
$$

$$
\leq \sqrt{2 \frac{B \log 2}{n_{\text{te}}}}. \tag{48}
$$

### D.1.2   Proofs for the UMB

The proof goes similarly as in the case of UWB up to Eq.(41). Then we need special care to bound the exponential moment since in the UMB, bins are constructed using $S_{\text{te}}$, and thus the samples are no longer i.i.d. Recall the definitions that $f_{(m)}$ is the $m$-th order statistics of $(f_1, \ldots, f_{n_{\text{te}}})$ and the boundaries of bins are defined by such order statistics $u_b := f_{(\lfloor n_{\text{te}} b/B \rfloor)}$.

We define the set $S_{(B)}$, which is the set of test data points used for defining the boundaries of bins. We then define $\tilde{S}_{\text{te}}$ as $S_{\text{te}} - S_{(B)}$, and thus $\tilde{S}_{\text{te}}$ is the set of test data points, which is not used for the boundaries. We express $f_w(S_{(B)}) = \{f_w(x) | x \in S_{(B)}\}$.

We define $k_b := \lfloor n_{\text{te}} b/B \rfloor$, which is used for defining the $b$-th bin. Let fix $b \in [B]$ and denote $u = k_b$ and $l = k_{b-1}$. Then Gupta & Ramdas [12] showed that $f_{(l+1)}, \ldots, f_{(u-1)}$ are independent and identically distributed given boundary points $\{f_{(\lfloor n_{\text{te}} b/B \rfloor)}\}_{b=0}^{B-1}$ in Lemma 1 [12]. Moreover, Lemma 2 in Gupta & Ramdas [12] showed that let $p$ be the density induced by the distribution $P(f_w(X))$, then

$$
p(f_{(l+1)}, \ldots, f_{(u-1)} | f_w(S_{(B)}))
$$
$$
= p(\tilde{f}_{l+1}, \ldots, \tilde{f}_{u-1} | f_w(S_{(B)}), \text{for every } i \in [l+1, u-1], f_{(l)} < \tilde{f}_i < f_{(u)}), \tag{49}
$$

where each $\tilde{f}_i$ is independent random variables $\tilde{f}_i \sim P(f_w(X))$. This implies that given the boundary points defining bins, the data points inside the boundary are i.i.d.

In order to use this result, we need to eliminate $f_{(u)}$ from the empirical mean of UMB. This is evaluated as follows; using this from the result of UWB, we have

$$|\mathrm{TCE}(f_{\mathcal{I}}) - \mathrm{ECE}(f_W, S_{\mathrm{te}})|$$

$$\leq \sum_{i=1}^{B} \left| \mathbb{E}_{Z''} l_i(Z'') - \frac{1}{n_{\mathrm{te}}} \sum_{m=1}^{n_{\mathrm{te}}} l_i(Z'_m) \right|$$

$$\leq \sum_{i=1}^{B} \left| \mathbb{E}_{Z''} l_i(Z'') - \frac{1}{|\tilde{S}_{\mathrm{te}}|} \sum_{m=1}^{|\tilde{S}_{\mathrm{te}}|} l_i(Z'_m) \right| + \left| \frac{1}{|\tilde{S}_{\mathrm{te}}|} \sum_{m=1}^{|\tilde{S}_{\mathrm{te}}|} l_i(Z'_m) - \frac{1}{n_{\mathrm{te}}} \sum_{m=1}^{n_{\mathrm{te}}} l_i(Z'_m) \right|. \qquad (50)$$

This partition eliminates the boundary point from the empirical estimation. We can upper bound the second term as

$$\left| \frac{1}{|\tilde{S}_{\mathrm{te}}|} \sum_{m=1}^{|\tilde{S}_{\mathrm{te}}|} l_i(Z'_m) - \frac{1}{n_{\mathrm{te}}} \sum_{m=1}^{n_{\mathrm{te}}} l_i(Z'_m) \right| \leq \frac{2B}{n_{\mathrm{te}} - B}, \qquad (51)$$

which follows directly by definition (see also Corollary 1 in Gupta & Ramdas [12].). Then we have

$$\mathbb{E}_{S_{\mathrm{te}}} |\mathrm{TCE}(f_{\mathcal{I}}) - \mathrm{ECE}(f_W, S_{\mathrm{te}})|$$

$$\leq \frac{1}{t} \mathbb{E}_{S_{\mathrm{te}}} e^{t \sum_{i=1}^{B} |\mathbb{E}_{Z''} l_i(Z'') - \frac{1}{n_{\mathrm{te}}} \sum_{m=1}^{n_{\mathrm{te}}} l_i(Z'_m)|}$$

$$\leq \frac{1}{t} \mathbb{E}_{S_{(B)}} \left[ \mathbb{E}_{\tilde{S}_{\mathrm{te}}} e^{t \sum_{i=1}^{B} |\mathbb{E}_{Z''} l_i(Z'') - \frac{1}{|\tilde{S}_{\mathrm{te}}|} \sum_{m \in \tilde{S}_{\mathrm{te}}} l_i(Z'_m)| + \frac{2tB}{n-B}} \right]. \qquad (52)$$

Given the boundary point, the above exponential moment satisfies the condition of Lemma 3, since the $l_i(Z'_m)$ are i.i.d, given in each bin by Lemma 2 in Gupta & Ramdas [12]. This can also be confirmed that for random variables $(f_{(1)}, \ldots, f_{(i)}, \ldots, f_{(n_{\mathrm{te}})})$, given $f_{(i)}$, $(f_{(1)}, \ldots, f_{(i-1)})$ and $f_{(i+1)}, \ldots, f_{(n_{\mathrm{te}})})$ are conditionally independent (this is proved in Gupta & Ramdas [12], especially the proof of Lemma 2). Combined with Eq. (49), given the boundary points, $\tilde{S}_{\mathrm{te}}$ are i.i.d and the size of which is $n_{\mathrm{te}} - B$. Then we only need to upper bound

$$\frac{1}{t} \mathbb{E}_{S_{(B)}} \left[ \mathbb{E}_{\tilde{S}_{\mathrm{te}}} e^{t \sum_{i=1}^{B} |\mathbb{E}_{Z''} l_i(Z'') - \frac{1}{|\tilde{S}_{\mathrm{te}}|} \sum_{m \in \tilde{S}_{\mathrm{te}}} l_i(Z'_m)|} \right]. \qquad (53)$$

We can upper bound this in a similar way as in the case of UWB, replacing $n_{\mathrm{te}}$ with $n_{\mathrm{te}} - B$ under Assumption 1. Thus, we have

$$\mathbb{E}_{S_{\mathrm{te}}} |\mathrm{TCE}(f_{\mathcal{I}}) - \mathrm{ECE}(f_W, S_{\mathrm{te}})| \leq \frac{1}{t} \mathbb{E}_{S_{\mathrm{te}}} e^{t \sum_{i=1}^{B} |\mathbb{E}_{Z''} l_i(Z'') - \frac{1}{n_{\mathrm{te}}} \sum_{m=1} l_i(Z'_m)|}$$

$$\leq \sqrt{2 \frac{B \log 2}{n_{\mathrm{te}} - B}} + \frac{2B}{n_{\mathrm{te}} - B}. \qquad (54)$$

This concludes the proof. $\qquad \square$

### D.2 Comparison with the trivial bound

We remark that for UWB, we can also upper bound in the following way;

$$\mathbb{E}_{S_{\mathrm{te}}} |\mathrm{TCE}(f_{\mathcal{I}}) - \mathrm{ECE}(f_W, S_{\mathrm{te}})| \leq \mathbb{E}_{S_{\mathrm{te}}} \sum_{i=1}^{B} \left| \mathbb{E}_{Z''} l_i(Z'') - \frac{1}{n_{\mathrm{te}}} \sum_{m=1}^{n_{\mathrm{te}}} l_i(Z'_m) \right|$$

$$\leq \mathbb{E}_{S_{\mathrm{te}}} \sum_{i=1}^{B} \sqrt{\mathrm{Var} \left[ \frac{1}{n_{\mathrm{te}}} \sum_{m=1}^{n_{\mathrm{te}}} l_i(Z'_m) \right]}$$

$$\leq \mathbb{E}_{S_{\mathrm{te}}} \sum_{i=1}^{B} \sqrt{\frac{1}{n_{\mathrm{te}}}} = \frac{B}{\sqrt{n_{\mathrm{te}}}}, \qquad (55)$$

where we used the triangle inequality $||a| - |b|| \leq |a - b|$ for the first inequality and set $l_i(z) = (y - f_W(x)) \cdot \mathbb{1}_{f_W(x) \in I_i}$. Note that since $-1 \leq l_i(z) \leq 1$, we can use Eq. (36). However, since we did not use the property of the indicator function, this suffers from the slow convergence of $B$.

## D.3 High probability bound

In the main paper, we present the expectation bound. On the other hand, as shown in the above proof, we evaluated the exponential moment. Thus, we can obtain the high probability bound directly.

**Corollary 2.** *Under the same assumptions in Theorem 2, for any $\delta \in (0, 1)$, we have*

$$P_{S_{\text{te}}} \left( |\text{TCE}(f_{\mathcal{I}}) - \text{ECE}(f_W, S_{\text{te}})| \leq \sqrt{2 \frac{B \log 2 + \log \frac{1}{\delta}}{n_{\text{te}}}} \right) \geq 1 - \delta. \tag{56}$$

This means the statistical bias is $\mathcal{O}_p(\sqrt{B/n_{\text{te}}})$.

*Proof.* Using the proof of Theorem 2, and Chernoff-bounding technique, for any $t > 0$, we have

$$P_{S_{\text{te}}} \left( |\text{TCE}(f_{\mathcal{I}}) - \text{ECE}(f_W, S_{\text{te}})| \geq \varepsilon \right) \leq e^{-t\varepsilon} \mathbb{E}_{S_{\text{te}}} e^{t \sum_{i=1}^{B} \left| \mathbb{E}_{Z''} l_i(Z'') - \frac{1}{n_{\text{te}}} \sum_{m=1}^{n_{\text{te}}} l_i(Z'_m) \right|}$$

$$\leq 2^B e^{-\frac{n\varepsilon^2}{2n_{\text{te}}} + \frac{(t - n\varepsilon)^2}{2n_{\text{te}}}}. \tag{57}$$

By setting $t = n\varepsilon$ then

$$P_{S_{\text{te}}} \left( |\text{TCE}(f_{\mathcal{I}}) - \text{ECE}(f_W, S_{\text{te}})| \geq \varepsilon \right) \leq 2^B e^{-\frac{n_{\text{te}}\varepsilon^2}{2n_{\text{te}}}}, \tag{58}$$

and setting $\delta := 2^B e^{-\frac{n\varepsilon^2}{2n_{\text{te}}}}$, we have that

$$P_{S_{\text{te}}} \left( |\text{TCE}(f_{\mathcal{I}}) - \text{ECE}(f_W, S_{\text{te}})| \geq \sqrt{2 \frac{B \log 2 + \log \frac{1}{\delta}}{n_{\text{te}}}} \right) \leq \delta. \tag{59}$$

$\square$

## D.4 Proofs of Theorem 3

*Proof.* We use the following lemma to study the binning bias.

**Lemma 4.**

$$\text{TCE}(f_{\mathcal{I}}) \leq \text{TCE}(f_w) \leq \text{TCE}(f_{\mathcal{I}}) + \mathbb{E}||\mathbb{E}[Y|f_w(X)] - \mathbb{E}[Y|f_{\mathcal{I}}(X)]| + \mathbb{E}|f_w(X) - f_{\mathcal{I}}(X)|. \tag{60}$$

*This implies that*

$$|\text{TCE}(f_w) - \text{TCE}(f_{\mathcal{I}})| \leq \mathbb{E}||\mathbb{E}[Y|f_w(X)] - \mathbb{E}[Y|f_{\mathcal{I}}(X)]| + \mathbb{E}|f_w(X) - f_{\mathcal{I}}(X)|. \tag{61}$$

*Proof.* The first inequality has been proved in Proposition 3.3 in Kumar et al. [24].

The second inequality is proved as follows;

$$
\begin{aligned}
\mathrm{TCE}(f_w) &= \mathbb{E}\left[|\mathbb{E}[Y|f_w(X)] - f_w(X)|\right] \\
&= \sum_{i=1}^{B} \mathbb{E}[\mathbb{1}_{f_w(X)\in I_i} \cdot |\mathbb{E}[Y|f_w(X)] - f_w(X)|] \\
&= \sum_{i=1}^{B} P(f_w(X) \in I_i)\mathbb{E}[|\mathbb{E}[Y|f_w(X)] - f_w(X)||f_w(X) \in I_i] \\
&= \sum_{i=1}^{B} P(f_w(X) \in I_i)\mathbb{E}[|\mathbb{E}[Y|f_w(X)] - \mathbb{E}[f_w(X)|f_w(X) \in I_i] \\
&\quad + \mathbb{E}[f_w(X)|f_w(X) \in I_i] - f_w(X)||f_w(X) \in I_i] \\
&\leq \sum_{i=1}^{B} P(f_w(X) \in I_i)\mathbb{E}||\mathbb{E}[Y|f_w(X)] - \mathbb{E}[Y|f_w(X) \in I_i]| \\
&\quad + \sum_{i=1}^{B} P(f_w(X) \in I_i)\mathbb{E}|\mathbb{E}[Y|f_w(X) \in I_i] - \mathbb{E}[f_w(X)|f_w(X) \in I_i]| \\
&\quad + \sum_{i=1}^{B} P(f_w(X) \in I_i)\mathbb{E}[|\mathbb{E}[f_w(X)|f_w(X) \in I_i] - f_w(X)||f_w(X) \in I_i]. \quad (62)
\end{aligned}
$$

In the above, the second term corresponds to $\mathrm{TCE}(f_{\mathcal{I}})$. $\qquad\square$

As for UWB, from Lemma 4, we have

$$
|\mathrm{TCE}(f_w) - \mathrm{TCE}(f_{\mathcal{I}})| \leq \mathbb{E}||\mathbb{E}[Y|f_w(X)] - \mathbb{E}[Y|f_{\mathcal{I}}(X)]| + \mathbb{E}|f_w(X) - f_{\mathcal{I}}(X)| \leq \frac{L}{B} + \frac{1}{B} \tag{63}
$$

where we used the fact that with UWB, we split the function with equal width $1/B$ and used the Lipschitz continuity of the function.

Next, we focus on UMB. To analyze the binning bias of this, we focus on Eq. (50). We replace the loss $l_m(z)$ in that equation with

$$
l_m(z) = \frac{\mathbb{1}_{f_W(x)\in I_m}}{n_{\mathrm{te}}}. \tag{64}
$$

Then, the first line and second line of Eq. (50) can be rewritten as

$$
\sum_{i=1}^{B} |P(f_w(x) \in I_m)) - \hat{P}(I_i)| \leq \sum_{i=1}^{B} \left| \mathbb{E}_{Z''} l_i(Z'') - \frac{1}{n_{\mathrm{te}}} \sum_{m=1}^{n_{\mathrm{te}}} l_i(Z'_m) \right| \tag{65}
$$

where $\hat{P}(I_i)$ is the empirical estimate of the binning probability $P(I_m)$. The right-hand side can be bounded in the same way as Appendix D.1.2, which requires evaluating the exponential moment. The proof goes the same way, that is, we utilize the results of Gupta & Ramdas [12] and the upper bound of the exponential moment. The procedure is exactly the same. Thus, we have

$$
\sum_{i=1}^{B} |P(f_w(x) \in I_m)) - \hat{P}(I_i)| \leq \sqrt{\frac{2B \log 2}{(n_{\mathrm{te}} - B)}} + \frac{2B}{n_{\mathrm{te}} - B}. \tag{66}
$$

By definition, we put equal mass in any bin, thus,

$$
\hat{P}(I_i) = \frac{u - l + 1}{n_{\mathrm{te}}} \leq \frac{1}{B}.
$$

from the proof of Theorem 3 in Gupta & Ramdas [12].

Thus, by the Jensen inequality, we have for any $m \in [B]$,

$$P(f_w(x) \in I_m)) \leq \frac{1}{B} + \sqrt{\frac{2B \log 2}{(n_{\text{te}} - B)}} + \frac{2B}{n_{\text{te}} - B}. \tag{67}$$

Combining these results, the binning bias is upper bounded by

$$\mathbb{E}|f_w(X) - f_{\mathcal{I}}(X)|$$

$$= \sum_{i=1}^{B} P(f_w(X) \in I_i)\mathbb{E}[|\mathbb{E}[f_w(X)|f_w(X) \in I_i] - f_w(X)||f_w(X) \in I_i]$$

$$\leq \left(\frac{1}{B} + \sqrt{\frac{2B \log 2}{(n_{\text{te}} - B)}} + \frac{2B}{n_{\text{te}} - B}\right) \sum_{i=1}^{B} \left(\mathbb{E}[|\mathbb{E}[f_W(X)|f_W(X) \in I_i] - f_W(X)||f_W(X) \in I_i]\right)$$

$$\tag{68}$$

We use the fact that $\mathbb{E}[|f_w(X) - \mathbb{E}[f_w(X)|I_i]||I_i] \leq f_{(\lfloor ni/B \rfloor)} - f_{(\lfloor n(i-1)/B \rfloor)}$ holds by the definition of the UMB, which is the largest difference of the bins. Then

$$\sum_{i=1}^{B} \left(\mathbb{E}[|\mathbb{E}[f_W(X)|f_W(X) \in I_i] - f_W(X)||f_W(X) \in I_i]\right) \leq \sum_{i=1}^{B} f_{(\lfloor ni/B \rfloor)} - f_{(\lfloor n(i-1)/B \rfloor)} \leq 1. \tag{69}$$

Combining the above, Then we have

$$\mathbb{E}|f_w(X) - f_{\mathcal{I}}(X)| \leq \frac{1}{B} + \sqrt{\frac{2B \log 2}{(n_{\text{te}} - B)}} + \frac{2B}{n_{\text{te}} - B}. \tag{70}$$

As for the $\mathbb{E}||\mathbb{E}[Y|f_w(X)] - \mathbb{E}[Y|f_{\mathcal{I}}(X)]|$, by the assumption of the Lipschitz continuity, we simply multiply $L$ to the above and obtain

$$\mathbb{E}||\mathbb{E}[Y|f_w(X)] - \mathbb{E}[Y|f_{\mathcal{I}}(X)]| \leq L\mathbb{E}|f_w(X) - f_{\mathcal{I}}(X)| \leq L\left(\frac{1}{B} + \sqrt{\frac{2B \log 2}{(n_{\text{te}} - B)}} + \frac{2B}{n_{\text{te}} - B}\right). \tag{71}$$

This concludes the proof. $\qquad\square$

### D.5 Hölder continuity does not improve the total bias

In the nonparametric estimation, imposing the higher order smoothness improves the bias order. According to Tsybakov [36], the lower bound is $\mathcal{O}(n_{\text{te}}^{-\frac{\beta}{\beta+1}})$ if we assume $\beta$-Hölder continuity.

In our total bias analysis, the statistical bias is not improved by this assumption. As for the binning bias, if we assume that $\mathbb{E}[Y|f_w(X)]$ satisfies $\beta$-Hölder continuity with constant $M$ for all the order, then we obtain that for UWB,

$$|\text{TCE}(f_w) - \text{TCE}(f_{\mathcal{I}})| \leq \mathbb{E}|\mathbb{E}[Y|f_w(X)] - \mathbb{E}[Y|f_{\mathcal{I}}(X)]| + \mathbb{E}|f_w(X) - f_{\mathcal{I}}(X)| \leq \frac{M}{B^\beta} + \frac{1}{B} \tag{72}$$

Combined with the statistical bias we have that

$$\text{Bias}_{\text{tot}}(f_w, S_{\text{te}}) \leq \frac{M}{B^\beta} + \frac{1}{B} + \sqrt{\frac{2B \log 2}{n_{\text{te}}}} \tag{73}$$

and the optimal bin size is again $\mathcal{O}(n_{\text{te}}^{1/3})$ and resulting bias is $\text{Bias}_{\text{tot}}(f_w, S_{\text{te}}) = \mathcal{O}(n_{\text{te}}^{-1/3})$, which does not improve the bias. This is because of the error term of $\mathbb{E}|f_w(X) - f_{\mathcal{I}}(X)|$. This term cannot be improved by $1/B$, thus we cannot leverage the underlying smoothness of the data. A similar discussion holds for the UMB setting.

## D.6 Additional comparison with existing work

In Gupta & Ramdas [12], the error bound of ECE is derived through the following three steps: (i) Firstly, showing that the samples assigned to each bin are i.i.d., (ii) using Hoeffding inequality, derieving $|\mathbb{E}[Y|f_{\mathcal{I}}(x)] - f_{\mathcal{I}}(x)| = \mathcal{O}_p(\sqrt{B/n_{\text{te}}})$ for each bin, and (iii) finally, summing up these error bounds for all bins, resulting in $\mathcal{O}_p(\sqrt{B \log B/n_{\text{te}}})$ (in expectation $O(B/\sqrt{n_{\text{te}}})$). This slow convergence is attributed to the separated analysis for each bin, which necessitates multiple applications of concentration inequalities.

In addition to the slow convergence, it is difficult to derive the error bound for UWB using this approach. The difficulty lies in demonstrating convergence for specific bins. For instance, in existing UMB studies Gupta & Ramdas [12], it becomes inevitable to address the allocation of samples to each bin when attempting to discuss the convergence of sample means for each bin. In UMB, an equal number of samples, $n_{\text{te}}/B$, are allocated to each bin to achieve equal mass across all bins. In UWB, however, there is no guarantee about the number of samples entering each bin (in the worst case, all samples might be assigned to a single bin) because the widths of all bins are set equally. This necessitates discussions about the number of samples allocated to intervals under strong assumptions regarding $\mathbb{E}[Y|f_w(x)]$, requiring stronger assumptions compared to both this study and existing research.

# E  Proofs of Section 4

First, we introduce notations, which are used in the IT-based analysis. We express the super-samples as

$$z = (x, y), \tag{74}$$
$$\tilde{z} = (\tilde{x}, \tilde{y}), \tag{75}$$
$$\tilde{z}_m = (\tilde{x}_m, \tilde{y}_m), \tag{76}$$
$$\tilde{z}_U = (\tilde{x}_U, \tilde{y}_U), \tag{77}$$
$$\tilde{z}_{m,U_m} = (\tilde{x}_{m,U_m}, \tilde{y}_{m,U_m}), \tag{78}$$
$$\tilde{z}_{m,\bar{U}_m} = (\tilde{x}_{m,\bar{U}_m}, \tilde{y}_{m,\bar{U}_m}). \tag{79}$$

We also define the total bias when using $S_{\text{tr}}$ as

$$\text{Bias}_{\text{tot}}(f_w, S_{\text{tr}}) := |\text{TCE}(f_w) - \text{ECE}(f_w, S_{\text{tr}})|. \tag{80}$$

We then decompose this bias into two biases as follows:

$$\text{Bias}_{\text{tot}}(f_w, S_{\text{tr}}) \leq \text{Bias}_{\text{bin}}(f_w, f_{\mathcal{I}}) + \text{Bias}_{\text{stat}}(f_w, S_{\text{tr}}), \tag{81}$$

where

$$\text{Bias}_{\text{bin}}(f_w, f_{\mathcal{I}}) := |\text{TCE}(f_w) - \text{TCE}(f_{\mathcal{I}})|, \tag{82}$$
$$\text{Bias}_{\text{stat}}(f_w, S_{\text{tr}}) := |\text{TCE}(f_{\mathcal{I}}) - \text{ECE}(f_w, S_{\text{tr}})|. \tag{83}$$

We remark that the bins used in $f_{\mathcal{I}}$ of UMB are constructed using $S_{\text{tr}}$ and thus, $\text{TCE}(f_{\mathcal{I}})$ also depends on $S_{\text{tr}}$.

## E.1  Proof of Theorem 4 (The statistical bias when reusing the training dataset)

*Proof.* We start with the case of UWB. The proofs goes almost similar way as the standard information-theoretic generalization error bounds.

Using the relation between the loss and ECEs by Eqs. (24) and (27), first we reformulate the ECEs as follows;

$$\text{ECE}(f_{\mathcal{A}(\tilde{Z}_U,R)}, \tilde{Z}_{\bar{U}}) = \sum_{i=1}^{B} \left| \frac{1}{n} \sum_{m=1}^{n} (\tilde{Y}_{m,\bar{U}_m} - f_{\mathcal{A}(\tilde{Z}_U,R)}(\tilde{X}_{m,\bar{U}_m})) \cdot \mathbb{1}_{f_W(\tilde{X}_{m,\bar{U}_m}) \in I_i} \right| \tag{84}$$

$$\text{ECE}(f_{\mathcal{A}(\tilde{Z}_U,R)}, \tilde{Z}_U) = \sum_{i=1}^{B} \left| \frac{1}{n} \sum_{m=1}^{n} (\tilde{Y}_{m,U_m} - f_{\mathcal{A}(\tilde{Z}_U,R)}(\tilde{X}_{m,U_m})) \cdot \mathbb{1}_{f_W(\tilde{X}_{m,U_m}) \in I_i} \right| \tag{85}$$

To simplify the notation, we also introduce the loss as

$$l(\mathcal{A}(\tilde{Z}_U, R), Z, i) := ((Y - f_{\mathcal{A}(\tilde{Z}_U, R)}(X)) \cdot \mathbb{1}_{f_{\mathcal{A}(\tilde{Z}_U, R)}(X) \in I_i}, \tag{86}$$

where $Z = (X, Y)$. Then we obtain

$$\Delta(U, \tilde{Z}, \mathcal{A}(\tilde{Z}_U, R)) := |\mathrm{ECE}(f_{\mathcal{A}(\tilde{Z}_U, R)}, \tilde{Z}_{\bar{U}}) - \mathrm{ECE}(f_{\mathcal{A}(\tilde{Z}_U, R)}, \tilde{Z}_U)|$$

$$= \left| \sum_{i=1}^{B} \left| \frac{1}{n} \sum_{m=1}^{n} l(\mathcal{A}(\tilde{Z}_U, R), \tilde{Z}_{m, \bar{U}_m}, i) \right| - \sum_{i=1}^{B} \left| \frac{1}{n} \sum_{m=1}^{n} l(\mathcal{A}(\tilde{Z}_U, R), \tilde{Z}_{m, U_m}, i) \right| \right|$$

$$\leq \sum_{i=1}^{B} \left| \frac{1}{n} \sum_{m=1}^{n} l(\mathcal{A}(\tilde{Z}_U, R), \tilde{Z}_{m, \bar{U}_m}, i) - \frac{1}{n} \sum_{m=1}^{n} l(\mathcal{A}(\tilde{Z}_U, R), \tilde{Z}_{m, U_m}, i) \right|, \tag{87}$$

where $W$ in the second line should be $W = \mathcal{A}(\tilde{Z}_U, R)$, but to make the presentation simpler, we used $W$. We also used the triangle inequality $||a| - |b|| < |a - b|$.

With this notation, by using the Donsker–Varadhan lemma, we have

$$\mathbb{E}_{R, \tilde{Z}, U} \Delta(U, \tilde{Z}, \mathcal{A}(\tilde{Z}_U, R))$$

$$\leq \inf_{t>0} \frac{I(\Delta(U, \tilde{Z}, \mathcal{A}(\tilde{Z}_U, R)); U|\tilde{Z}) + \mathbb{E}_{\tilde{Z}} \log \mathbb{E}_{R, U', U} e^{t\Delta(U', \tilde{Z}, \mathcal{A}(\tilde{Z}_U, R))}}{t}$$

$$\leq \inf_{t>0} \frac{I(\Delta(U, \tilde{Z}, \mathcal{A}(\tilde{Z}_U, R)); U|\tilde{Z})}{t}$$

$$+ \frac{\mathbb{E}_{\tilde{Z}} \log \mathbb{E}_{R, U', U} e^{t \sum_{i=1}^{B} \left| \frac{1}{n} \sum_{m=1}^{n} l(\mathcal{A}(\tilde{Z}_U, R), \tilde{Z}_{m, \bar{U}'_m}, i) - \frac{1}{n} \sum_{m=1}^{n} l(\mathcal{A}(\tilde{Z}_U, R), \tilde{Z}_{m, U'_m}, i) \right|}}{t}$$

$$= \inf_{t>0} \frac{I(\Delta(U, \tilde{Z}, \mathcal{A}(\tilde{Z}_U, R)); U|\tilde{Z}) + \mathbb{E}_{\tilde{Z}} \log \mathbb{E}_{R, U', U} e^{t \sum_{i=1}^{B} |g(\tilde{Z}, U, R, U', i)|}}{t}, \tag{88}$$

where we introduced

$$g(\tilde{z}, u, r, U', i) := \frac{1}{n} \sum_{m=1}^{n} l(\mathcal{A}(\tilde{z}_u, r), \tilde{z}_{m, \bar{U}'_m}, i) - \frac{1}{n} \sum_{m=1}^{n} l(\mathcal{A}(\tilde{z}_u, r), \tilde{z}_{m, U'_m}, i). \tag{89}$$

Our first observation is that conditioned on $\tilde{Z} = \tilde{z}$, $R = r$, and $U = u$, the expectation of the exponent is

$$\mathbb{E}_{U'} \frac{t}{B} \sum_{i=1}^{B} g(\tilde{z}, u, r, U', i) = 0, \tag{90}$$

by definition. Then similarly to Appendix D.1, we upper bound the exponential moment as follows;

$$\mathbb{E}_{U'} e^{t \sum_{i=1}^{B} |g(\tilde{z}, u, r, U', i)|} = \mathbb{E}_{U'} \prod_{i=1}^{B} e^{t|g(\tilde{z}, u, r, U', i)|}$$

$$\leq \mathbb{E}_{U'} \prod_{i=1}^{B} \left( e^{tg(\tilde{z}, u, r, U', i)} + e^{-tg(\tilde{z}, u, r, U', i)} \right)$$

$$= \mathbb{E}_{U'} \sum_{v_1, \dots, v_B = 0, 1} e^{t \sum_{i=1}^{B} (-1)^{v_i} g(\tilde{z}, u, r, U', i)} \tag{91}$$

$$= \sum_{v_1, \dots, v_B = 0, 1} \mathbb{E}_{U'} e^{t \sum_{i=1}^{B} (-1)^{v_i} g(\tilde{z}, u, r, U', i)}$$

$$= \sum_{v_1, \dots, v_B = 0, 1} \mathbb{E}_{U'} \prod_{m=1}^{n} e^{\frac{t}{n} \sum_{i=1}^{B} (-1)^{v_i} \left[ l(w, \tilde{z}_{m, \bar{U}'_m}, i) - l(w, \tilde{z}_{m, U'_m}, i) \right]}$$

$$= \sum_{v_1, \dots, v_B = 0, 1} \prod_{m=1}^{n} \mathbb{E}_{U'_m} e^{\frac{t}{n} \sum_{i=1}^{B} (-1)^{v_i} \left[ l(w, \tilde{z}_{m, \bar{U}'_m}, i) - l(w, \tilde{z}_{m, U'_m}, i) \right]}, \tag{92}$$

where $\sum_{v_1,\ldots,v_B=0,1}$ is all the combinations that will be generated by expanding $\prod_{i=1}^{B}$ in Eq. (91) and it has $2^B$ combinations.

We would like to upper bound $\mathbb{E}_{U'_m} e^{\frac{t}{n} \sum_{i=1}^{n} (-1)^{v_i} \left[ l(w, \tilde{z}_{m, \bar{U}'_m}, i) - l(w, \tilde{z}_{m, U'_m}, i) \right]}$ using Lemma 3 conditioned on all other random variables. For that purpose, here we evaluate $c_i$ of Lemma 3. To estimate it let us focus on $U'_m$ and it takes value $U'_m = \{0, 1\}$. So let us consider how the exponent changes by changing $U'_m = 1$ to $U'_m = 0$. Then the difference of the exponent is written as

$$\sum_{i=1}^{B} \frac{t(-1)^{v_i}}{n} \cdot [l(w, \tilde{z}_{m,1}, i) - l(w, \tilde{z}_{m,0}, i)] - \frac{t(-1)^{v_i}}{n} \cdot [l(w, \tilde{z}_{m,0}, i) - l(w, \tilde{z}_{m,1}, i)]$$

$$= 2\frac{t(-1)^{v_1}}{n} \cdot \left( \left( [y_{m,1} - f_w(x_{m,1})] \cdot \mathbb{1}_{f_w(x_{m,1}) \in I_1} \right) - \left( [y_{m,0} - f_w(x_{m,0})] \cdot \mathbb{1}_{f_w(x_{m,0}) \in I_1} \right) \right) +$$

$$\vdots$$

$$+ 2\frac{t(-1)^{v_B}}{n} \cdot \left( \left( [y_{m,1} - f_w(x_{m,1})] \cdot \mathbb{1}_{f_w(x_{m,1}) \in I_B} \right) - \left( [y_{m,0} - f_w(x_{m,0})] \cdot \mathbb{1}_{f_w(x_{m,0}) \in I_B} \right) \right). \tag{93}$$

To evaluate the indicator function, we repeat the same discussion in Eqs. 44 and (45). On the basis of that discussion, by the definition of binning, for the input $x_{n,0}$, exactly one of the indicators $\{\mathbb{1}_{f_w(x_{n,0}) \in I_i}\}_{i=1}^{B}$ is non-zero, denoted as $b$. Consequently, all other indicators are zero, i.e., $\mathbb{1}_{f_w(x_{n,0}) \in I_{b' \neq b}} = 0$. Similarly, for input $x_{n,1}$, the corresponding non-zero bin index is denoted as $\tilde{b}$, so $\mathbb{1}_{f_w(x_{n,1}) \in I_{\tilde{b}}}$ is nonzero and others are zero. It should be noted that $b$ and $\tilde{b}$ may be the same or different.

Thus, although there are $2B$ indicator functions in Eq. (93), at most only two indicator functions are not zero. Combined with the fact that $|y_{m,1} - f_w(x_{m,1})| \leq 1$ and $|y_{m,0} - f_w(x_{m,0})| \leq 1$, Eq. (93) is upper bounded by $\frac{4t}{n}$.

Note that by Assumption 1, $\{f_w(x_{m,U_m})\}_{m=1}^{n}$ takes the distinct values almost surely and in the above discussion, we do not consider the case when $b/B = f_w(x_{m,U_m})$ for some $b$ holds, which means that the predicted probability is just the value of the boundary of bins. In conclusion, we obtain the upper bound of Eq. (93) as

$$\left| \sum_{i=1}^{B} \frac{t(-1)^{v_i}}{n} \cdot [l(w, \tilde{z}_{m,1}, i) - l(w, \tilde{z}_{m,0}, i)] - \frac{t(-1)^{v_i}}{n} \cdot [l(w, \tilde{z}_{m,0}, i) - l(w, \tilde{z}_{m,1}, i)] \right| \leq \frac{4t}{n}. \tag{94}$$

Then by Lemma 3, we have that

$$\sum_{v_1,\ldots,v_B=0,1} \prod_{m=1}^{n} \mathbb{E}_{U'_m} e^{\frac{t}{n} \sum_{i=1}^{n} (-1)^{v_i} \left[ l(w, \tilde{z}_{m, \bar{U}'_m}, i) - l(w, \tilde{z}_{m, U'_m}, i) \right]} \leq \sum_{v_1,\ldots,v_B=0,1} \prod_{m=1}^{n} e^{\frac{2t^2}{n^2}} = 2^B e^{\frac{2t^2}{n}}. \tag{95}$$

Thus

$$\mathbb{E}_{U'} e^{t \sum_{i=1}^{B} |g(\tilde{z}, u, r, U', i)|} = \mathbb{E}_{U'} \prod_{i=1}^{B} e^{t|g(\tilde{z}, u, r, U', i)|} \leq 2^B e^{\frac{2t^2}{n}}, \tag{96}$$

and combining with Eq. (88), we have

$$\mathbb{E}_{R, \tilde{Z}, U} \left| \sum_{i=1}^{B} \left| \frac{1}{n} \sum_{m=1}^{n} l(\mathcal{A}(\tilde{Z}_U, R), \tilde{Z}_{m, \bar{U}_m}, i) \right| - \sum_{i=1}^{B} \left| \frac{1}{n} \sum_{m=1}^{n} l(\mathcal{A}(\tilde{Z}_U, R), \tilde{Z}_{m, U_m}, i) \right| \right|$$

$$\leq \inf_{t>0} \frac{I(\Delta(U, \tilde{Z}, \mathcal{A}(\tilde{Z}_U, R)); U|\tilde{Z}) + B \log 2 + \frac{2t^2}{n}}{t}$$

$$\leq \sqrt{\frac{8(I(\Delta(U, \tilde{Z}, \mathcal{A}(\tilde{Z}_U, R)); U|\tilde{Z}) + B \log 2)}{n}}. \tag{97}$$

This concludes the proof of UWB.

We next prove the case of UMB. The key difference lies in the fact that the bins are dependent on the training samples.

$$\Delta(U, \tilde{Z}, \mathcal{A}(\tilde{Z}_U, R)) \coloneqq |\text{ECE}(f_W, \tilde{Z}_{\bar{U}}) - \text{ECE}(f_W, \tilde{Z}_U)|$$

$$= \left| \sum_{i=1}^{B} \left| \frac{1}{n} \sum_{m=1}^{n} (\tilde{y}_{m,\bar{U}_m} - f_{\mathcal{A}(\tilde{Z}_U, R)}(\tilde{x}_{m,\bar{U}_m})) \cdot \mathbb{1}_{f_w(\tilde{x}_{m,\bar{U}_m}) \in I_i(\tilde{Z}_U)} \right| \right.$$

$$\left. - \sum_{i=1}^{B} \left| \frac{1}{n} \sum_{m=1}^{n} (y_{m,\bar{U}_m} - f_{\mathcal{A}(\tilde{Z}_U, R)}(x_{m,\bar{U}_m})) \cdot \mathbb{1}_{f_w(x_{m,\bar{U}_m}) \in I_i(\tilde{Z}_U)} \right| \right|,$$

where we expressed the dependency of bins on the training samples as $I_i(\tilde{Z}_U)$. However, this dependency does not change the proof in the above; we use the Donsker–Varadhan lemma. We upper bound of the exponential moment. When upper bounding the exponential moment, we conditioned on $U$, which means we conditioned on the bins. So we can exactly use the same derivation. So we can proceed with the proof exactly in the same way as UWB.

Finally, we can bound the statistical bias using the Jensen inequality as follows: First following the proof of Theorem 2, using Eqs. (24) and (27), we have

$$\mathbb{E}_{R,S_{\text{tr}}} |\text{TCE}(f_{\mathcal{I}}) - \text{ECE}(f_W, S_{\text{tr}})|$$

$$= \mathbb{E}_{R,S_{\text{tr}}} \left| \sum_{i=1}^{B} \left| \mathbb{E}_{Z''=(X'',Y'')} [(Y'' - f_W(X'')) \cdot \mathbb{1}_{f_W(X'') \in I_i}] \right| - \sum_{i=1}^{B} \left| \frac{1}{n} \sum_{m=1}^{n} (Y_m - f_W(X_m)) \mathbb{1}_{f_W(X_m) \in I_i} \right| \right|$$

$$\leq \mathbb{E}_{R,S_{\text{tr}}} \sum_{i=1}^{B} \left| \mathbb{E}_{Z''=(X'',Y'')} [(Y'' - f_W(X'')) \cdot \mathbb{1}_{f_W(X'') \in I_i}] - \frac{1}{n} \sum_{m=1}^{n} (Y_m - f_W(X_m)) \cdot \mathbb{1}_{f_W(X_m) \in I_i} \right|$$

$$= \mathbb{E}_{R,S_{\text{tr}}} \sum_{i=1}^{B} \left| \mathbb{E}_{\{Z''_m\}_{m=1}^n} \frac{1}{n} \sum_{m=1}^{n} [(Y''_m - f_W(X''_m)) \cdot \mathbb{1}_{f_W(X''_m) \in I_i}] - \frac{1}{n} \sum_{m=1}^{n} (Y_m - f_W(X_m)) \mathbb{1}_{f_W(X_m) \in I_i} \right|$$

$$\leq \mathbb{E}_{R,S_{\text{tr}},\{Z''_m\}_{m=1}^n} \sum_{i=1}^{B} \left| \frac{1}{n} \sum_{m=1}^{n} [(Y''_m - f_W(X''_m)) \cdot \mathbb{1}_{f_W(X''_m) \in I_i}] - \frac{1}{n} \sum_{m=1}^{n} (Y_m - f_W(X_m)) \cdot \mathbb{1}_{f_W(X_m) \in I_i} \right|$$

$$= \mathbb{E}_{R,S_{\text{tr}},S_{\text{te}}} |\text{ECE}(f_W, S_{\text{te}}) - \text{ECE}(f_W, S_{\text{tr}})|, \tag{98}$$

where the first inequality is the triangle inequality and the second inequality is the Jensen inequality. Note that the above reformulation is possible for both UWB and UMB. Although in the case of UMB, bins of the TCE still depend on $S_{\text{tr}}$, it makes no difference in the above inequalities. We then use Theorem 4.

$$\square$$

**Remark 1.** *In the above proof of UWB, instead of Eq. (88), it is possible to consider the following type Donsker–Varadhan inequality*

$$\mathbb{E}_{R,\tilde{Z},U} \left| \sum_{i=1}^{B} \left| \frac{1}{n} \sum_{m=1}^{n} l(\mathcal{A}(\tilde{Z}_U, R), \tilde{Z}_{m,\bar{U}_m}, i) \right| - \sum_{i=1}^{B} \left| \frac{1}{n} \sum_{m=1}^{n} l(\mathcal{A}(\tilde{Z}_U, R), \tilde{Z}_{m,U_m}, i) \right| \right|$$

$$\leq \inf_{t>0} \frac{I(l(\mathcal{A}(\tilde{Z}_U, R), \tilde{Z}, B); U | \tilde{Z})}{t}$$

$$+ \frac{\mathbb{E}_{\tilde{Z}} \log \mathbb{E}_{R,U',U} e^{t \sum_{i=1}^{B} \left| \frac{1}{n} \sum_{m=1}^{n} l(\mathcal{A}(\tilde{Z}_U, R), \tilde{Z}_{m,\bar{U}'_m}, i) - \frac{1}{n} \sum_{m=1}^{n} l(\mathcal{A}(\tilde{Z}_U, R), \tilde{Z}_{m,U'_m}, i) \right|}}{t}$$

$$= \inf_{t>0} \frac{I(l(\mathcal{A}(\tilde{Z}_U, R), \tilde{Z}, B); U | \tilde{Z}) + \mathbb{E}_{\tilde{Z}} \log \mathbb{E}_{R,U',U} e^{t \sum_{i=1}^{B} |g(\tilde{Z}, U, R, U', i)|}}{t}, \tag{99}$$

*which results in a looser bound than the above proof, which can be confirmed by the data processing inequality.*

## E.2 Proof of Theorem 5 (The total bias)

Before presenting the proof of the total bias, we first provide the following binning bias analysis for the UMB.

**Theorem 8.** *For UMB, Under the CMI setting and under Assumptions 1 and 2, we have*

$$\mathbb{E}_{R,S_{\text{tr}}}|\text{TCE}(f_W) - \text{TCE}(f_{\mathcal{I}})| \le (1+L)\left(\frac{1}{B} + \sqrt{\frac{2}{n}\left(\text{fCMI} + B\log 2\right)}\right), \tag{100}$$

*where fCMI is defined in Eq. (17).*

*Proof of Theorem 8.* The proof is similar in Appendix D.4. The difference is that in the current setting, we reuse the training dataset, so we need to evaluate the bias for that. First, in the same way as in Appendix D.4, we have that

$$\mathbb{E}_{R,S_{\text{tr}}}\mathbb{E}|f_W(X) - f_{\mathcal{I}}(X)|$$

$$= \mathbb{E}_{R,S_{\text{tr}}}\sum_{i=1}^{B} P(f_W(X) \in I_i)\mathbb{E}|\mathbb{E}[f_W(X)|f_W(X) \in I_i] - f_W(X)||f_W(X) \in I_i]$$

$$\le \sum_{i=1}^{B}\left(\mathop{\mathbb{E}}_{R,S_{\text{tr}}}|P(f_W(X) \in I_i)|\right)\left(\mathop{\mathbb{E}}_{R,S_{\text{tr}}}|\mathbb{E}[|\mathbb{E}[f_W(X)|f_W(X) \in I_i] - f_W(X)||f_W(X) \in I_i]|_\infty\right), \tag{101}$$

where we used Hölder inequality in the second line and $\mathbb{E}|\cdot|_\infty$ is the maximum of the integrand.

We want to estimate $P(I_i) \coloneqq P(f(X) \in I_i)$. For this purpose, we use Eqs. (87) and (86). We re-define the loss of Eq. (86)

$$l(\mathcal{A}(\tilde{Z}_U, R), z, i) = \mathbb{1}_{f_{\mathcal{A}(\tilde{Z}_U, R)}(x) \in I_i}. \tag{102}$$

and substitute it into Eq. (87), then we have that

$$\mathop{\mathbb{E}}_{R,\tilde{Z},U}\sum_{i=1}^{B}\left|\tilde{P}(I_i) - \hat{P}(I_i)\right| = \mathop{\mathbb{E}}_{R,\tilde{Z},U}\sum_{i=1}^{B}\left|\frac{\sum_{m=1}^{n}\mathbb{1}_{f_{\mathcal{A}(\tilde{Z}_U, R)}(\tilde{X}_{m,\bar{U}_m}) \in I_i}}{n} - \frac{\sum_{m=1}^{n}\mathbb{1}_{f_{\mathcal{A}(\tilde{Z}_U, R)}(\tilde{X}_{m,U_m}) \in I_i}}{n}\right|, \tag{103}$$

where $\hat{P}(I_i)$ is the empirical estimate of the binning probability using supersample $\tilde{X}_{m,U_m}$ and $\tilde{P}(I_i)$ is that of obtained by $\tilde{X}_{m,\bar{U}_m}$. Then, to obtain the upper bound of the right-hand side of the above, we repeat the proof of Theorem 4 in Appendix E.1. Let us define

$$\Delta(U, \tilde{Z}, \mathcal{A}(\tilde{Z}_U, R)) \coloneqq \sum_{i=1}^{B}\left|\frac{1}{n}\sum_{m=1}^{n}(\mathbb{1}_{f_{\mathcal{A}(\tilde{Z}_U, R)}(\tilde{X}_{m,\bar{U}_m}) \in I_i} - \mathbb{1}_{f_{\mathcal{A}(\tilde{Z}_U, R)}(\tilde{X}_{m,U_m}) \in I_i})\right| \tag{104}$$

With this notation, by using the Donsker–Varadhan lemma, we have

$$
\mathbb{E}_{R,\tilde{Z},U}\sum_{i=1}^{B}\left|\tilde{P}(I_i)-\hat{P}(I_i)\right|
$$

$$
=\mathbb{E}_{R,\tilde{Z},U}\sum_{i=1}^{B}\left|\frac{\sum_{m=1}^{n}\mathbb{1}_{f_{\mathcal{A}(\tilde{Z}_U,R)}(\tilde{X}_{m,\bar{U}_m})\in I_i}}{n}-\frac{\sum_{m=1}^{n}\mathbb{1}_{f_{\mathcal{A}(\tilde{Z}_U,R)}(\tilde{X}_{m,U_m})\in I_i}}{n}\right|
$$

$$
\mathbb{E}_{R,\tilde{Z},U}\Delta(U,\tilde{Z},\mathcal{A}(\tilde{Z}_U,R))
$$

$$
\leq\inf_{t>0}\frac{I(\Delta(U,\tilde{Z},\mathcal{A}(\tilde{Z}_U,R));U|\tilde{Z})+\mathbb{E}_{\tilde{Z}}\log\mathbb{E}_{R,U',U}\,e^{t\Delta(U',\tilde{Z},\mathcal{A}(\tilde{Z}_U,R))}}{t}
$$

$$
\leq\inf_{t>0}\frac{I(f_{\mathcal{A}(\tilde{Z}_U,R)}(\tilde{X});U|\tilde{Z})}{t}
$$

$$
+\frac{\mathbb{E}_{\tilde{Z}}\log\mathbb{E}_{R,U',U}\,e^{t\sum_{i=1}^{B}|\frac{\sum_{m=1}^{n}\mathbb{1}_{f_{\mathcal{A}(\tilde{Z}_U,R)}(\tilde{X}_{m,\bar{U}'_m})\in I_i}}{n}-\frac{\sum_{m=1}^{n}\mathbb{1}_{f_{\mathcal{A}(\tilde{Z}_U,R)}(\tilde{X}_{m,U'_m})\in I_i}}{n}|}}{t}
$$

$$
=\inf_{t>0}\frac{I(f_{\mathcal{A}(\tilde{Z}_U,R)}(\tilde{X});U|\tilde{Z})+\mathbb{E}_{\tilde{Z}}\log\mathbb{E}_{R,U',U}e^{t\sum_{i=1}^{B}|g(\tilde{Z},U,R,U',i)|}}{t},\tag{105}
$$

where we introduced

$$
g(\tilde{z},u,r,U',i):=\frac{1}{n}\sum_{m=1}^{n}l(\mathcal{A}(\tilde{z}_u,r),\tilde{z}_{m,\bar{U}'_m},i)-\frac{1}{n}\sum_{m=1}^{n}l(\mathcal{A}(\tilde{z}_u,r),\tilde{z}_{m,U'_m},i).\tag{106}
$$

Here we use the fCMI $:=I(f_{\mathcal{A}(\tilde{Z}_U,R)}(\tilde{X});U|\tilde{Z})$ and $I(\Delta(U,\tilde{Z},\mathcal{A}(\tilde{Z}_U,R));U|\tilde{Z})\leq$ fCMI by the data processing inequality since the indicator functions depend on $f_w$.

Then, we can estimate this using Lemma 3 in a similar way. The difference is the estimation of the upper-bound in Eq. (94), which is used for the evaluation of the exponential moment. Since we use the indicator function as a loss, the coefficient $c_i$s for Lemma 3 is upper-bounded by $2t/n$, not $4t/n$. Then we repeat the proof strategy replacing the exponential moment evaluation by $2t/n$, not $4t/n$.

With this difference,

$$
\mathbb{E}_{U'}e^{t\sum_{i=1}^{B}|g(\tilde{z},u,r,U',i)|}=\mathbb{E}_{U'}\prod_{i=1}^{B}e^{t|g(\tilde{z},u,r,U',i)|}\leq 2^{B}e^{\frac{t^2}{2n}},\tag{107}
$$

and Eq. (97) can be rewritten in the following way

$$
\mathbb{E}_{R,\tilde{Z},U}\sum_{i=1}^{B}\left|\tilde{P}(I_i)-\hat{P}(I_i)\right|\leq\sqrt{\frac{2(\text{fCMI}+B\log 2)}{n}},\tag{108}
$$

and clearly, by fixing some $i$, we have that

$$
\mathbb{E}_{R,\tilde{Z},U}\left|\tilde{P}(I_i)-\hat{P}(I_i)\right|\leq\mathbb{E}_{R,\tilde{Z},U}\sum_{i=1}^{B}\left|\tilde{P}(I_i)-\hat{P}(I_i)\right|\leq\sqrt{\frac{2(\text{fCMI}+B\log 2)}{n}}.\tag{109}
$$

Since we put an equal mass for each bin for the training dataset with UMB, we have

$$
\hat{P}(I_i)=\frac{u-l+1}{n}\leq\frac{1}{B}.
$$

Combined with Jensen inequality, we have

$$
\mathbb{E}_{R,S_{\text{tr}}}P(I_i)\leq\frac{1}{B}+\sqrt{\frac{2(\text{fCMI}+B\log 2)}{n}}.\tag{110}
$$

Then we have

$$\mathbb{E}_{R,S_{\mathrm{tr}}}\mathbb{E}|f_W(X) - f_{\mathcal{I}}(X)|$$

$$= \sum_{i=1}^{B}\left(\frac{1}{B} + \sqrt{\frac{2(\mathrm{fCMI} + B\log 2)}{n}}\right) \underset{R,S_{\mathrm{tr}}}{\mathbb{E}} |\mathbb{E}[\mathbb{E}[f_W(X)|f_W(X) \in I_i] - f_W(X)||f_W(X) \in I_i]|_{\infty}$$

$$= \left(\frac{1}{B} + \sqrt{\frac{2(\mathrm{fCMI} + B\log 2)}{n}}\right) \sum_{i=1}^{B} \underset{R,S_{\mathrm{tr}}}{\mathbb{E}} |\mathbb{E}[|\mathbb{E}[f_W(X)|f_W(X) \in I_i] - f_W(X)||f_W(X) \in I_i]|_{\infty}.$$

$$(111)$$

Finally, we use the fact that $\mathbb{E}[||f(X) - \mathbb{E}[f(X)|I_i]||I_i] \leq f_{(\lfloor ni/B \rfloor)} - f_{(\lfloor n(i-1)/B \rfloor)}$ holds by the definition of the UMB, which is the largest difference of the bins. Then

$$\sum_{i=1}^{B} \underset{R,S_{\mathrm{tr}}}{\mathbb{E}} |\mathbb{E}[|\mathbb{E}[f_W(X)|f_W(X) \in I_i] - f_W(X)||f_W(X) \in I_i]|_{\infty}$$

$$\leq \sum_{i=1}^{B} f_{(\lfloor ni/B \rfloor)} - f_{(\lfloor n(i-1)/B \rfloor)} \leq 1. \tag{112}$$

where $\mathbb{E}|\cdot|_{\infty}$ is the maximum of the integrand.

Combining the above, Then we have

$$\underset{R,S_{\mathrm{tr}}}{\mathbb{E}} \mathbb{E}|f_W(X) - f_{\mathcal{I}}(X)|$$

$$= \left(\frac{1}{B} + \sqrt{\frac{2(\mathrm{fCMI} + B\log 2)}{n}}\right) \sum_{i=1}^{B} \underset{R,S_{\mathrm{tr}}}{\mathbb{E}} |\mathbb{E}[|\mathbb{E}[f_W(X)|f_W(X) \in I_i] - f_W(X)||f_W(X) \in I_i]|_{\infty}$$

$$\leq \frac{1}{B} + \sqrt{\frac{2(\mathrm{fCMI} + B\log 2)}{n}}. \tag{113}$$

This concludes the proof. □

Using this we provide the proof of the total bias as follows;

*Proof of Theorem 5.* We use the triangle inequality,

$$\mathbb{E}_{R,S_{\mathrm{tr}}}|\mathrm{TCE}(f_W) - \mathrm{ECE}(f_W, S_{\mathrm{tr}})|$$
$$= \mathbb{E}_{R,S_{\mathrm{tr}}}|\mathrm{TCE}(f_W) - \mathrm{TCE}(f_{\mathcal{I}}) + \mathrm{TCE}(f_{\mathcal{I}}) - \mathrm{ECE}(f_W, S_{\mathrm{tr}})|$$
$$\leq \mathbb{E}_{R,S_{\mathrm{tr}}}|\mathrm{TCE}(f_W) - \mathrm{TCE}(f_{\mathcal{I}})| + \mathbb{E}_{R,S_{\mathrm{tr}}}|\mathrm{TCE}(f_{\mathcal{I}}) - \mathrm{ECE}(f_W, S_{\mathrm{tr}})|. \tag{114}$$

The first term is the binning bias and the second term is the statistical bias.

We start from the UMB; we can bound the binning bias in the first term by Theorem 8 and the statistical bias in the second term by Theorem 4 of the UMB.

As for the UWB, the binning bias is simply $(1 + L)/B$, which can be derived similarly as in Appendix D.4. As for the second term, we can bound it by Theorem 4 of the UWB.

This concludes the proof of Eq. (16). □

Provided in the proof of Theorem 8 (especially in Eq. (105)), we can obtain the tighter version of the binning bias bound as follows;

**Corollary 3.** *For UMB, Under the CMI setting and under Assumptions 1 and 2, we have*

$$\mathbb{E}_{R,S_{\mathrm{tr}}}|\mathrm{TCE}(f_W) - \mathrm{TCE}(f_{\mathcal{I}})| \leq (1 + L)\left(\frac{1}{B} + \sqrt{\frac{2}{n}\left(I(\Delta(U, \tilde{Z}, \mathcal{A}(\tilde{Z}_U, R)); U|\tilde{Z}) + B\log 2\right)}\right),$$

$$(115)$$

*where*

$$\Delta(U, \tilde{Z}, \mathcal{A}(\tilde{Z}_U, R)) := \sum_{i=1}^{B}\left|\frac{1}{n}\sum_{m=1}^{n}(\mathbb{1}_{f_{\mathcal{A}(\tilde{Z}_U, R)}(\tilde{X}_{m,\bar{U}_m}) \in I_i} - \mathbb{1}_{f_{\mathcal{A}(\tilde{Z}_U, R)}(\tilde{X}_{m,U_m}) \in I_i})\right|. \tag{116}$$

In the proof of Theorem 8, we used the fact that $I(\Delta(U, \tilde{Z}, \mathcal{A}(\tilde{Z}_U, R)); U | \tilde{Z}) \leq$ fCMI by the data processing inequality. Thus, fCMI appearing in Theorem 5 can be replaced with $I(\Delta(U, \tilde{Z}, \mathcal{A}(\tilde{Z}_U, R)); U | \tilde{Z})$, which results in a tighter bound.

## E.3 Proof of Theorem 6 (metric entropy)

*Proof.* Recall the setting, where we assume that $f_w \in \mathcal{F}$ has the metric entropy, $\log \mathcal{N}(\mathcal{F}, \|\cdot\|_\infty, \delta)$, with parameter $\delta$ ($> 0$). That is, there exists a set of functions $\mathcal{F}_\delta := \{f_1, \ldots, f_{\mathcal{N}(\mathcal{F}, \|\cdot\|_\infty, \delta)}\}$ that consists $\delta$-cover of $\mathcal{F}$. We will consider to replace $f_w$ with the functions from the $\delta$-cover.

Using the $\delta$-cover, we want to construct a set of functions $\tilde{\mathcal{F}}$ that satisfies the following property; there exists a function $f \in \tilde{\mathcal{F}}$ such that for any input $x$ and that $x$ is allocated to $i$-th bin, that is, $f_w(x) \in I_i$, then $f$ satisfies $f(x) \in I_i$ and $\|f - f_w\|_\infty < \delta$.

The original $\delta$ cover $\mathcal{F}$ may not satisfy this property as follows; If we simply consider that $h = \arg\min_{f \in \mathcal{F}_\delta} \|f_w - f\|_\infty$, then there is a possibility that $h(x) \notin I_i$ for $x \in \mathcal{X}$ such that $f_w(x) \in \mathcal{I}$. This will cause a problem when approximating the ECE using $h$. If $h(x) \notin I_i$, it significantly changes the estimation of the conditional expectation in each bin, leading to a larger change of the ECE. To avoid this, we consider a set of functions such that for each $i = 1, \ldots, B$, for any $f \in \mathcal{F}_\delta$, we define

$$f(x) := \max\left[\min\left(f(x), \frac{i}{B}\right), \frac{i-1}{B} + \epsilon\right], \tag{117}$$

where $0 < \epsilon < 1/2\delta$. We refer to this set of clipped functions as $\mathcal{F}_i$ for $i = 1, \ldots, B$. The parameter $\epsilon$ is introduced so that the clipped value does not take the boundary value between the $i$-th bin and $(i-1)$-th bin. Since we set $0 < \epsilon < 1/2\delta$, the parameter $\epsilon$ does not affect the bias analysis below. Then, we define the function

$$f(x) := \sum_{i=1}^{B} f_i(x) \mathbb{1}_{f_w(x) \in I_i}, \tag{118}$$

where $f_i := \arg\min_{f \in \mathcal{F}_i} \sup_{x \in \mathcal{X}_i} |f_w(x) - f(x)|$ and $\mathcal{X}_i := \{x \in \mathcal{X} | f_w(x) \in I_i\}$. (if $\mathcal{X}_i = \phi$, we do not need to consider $f_i$ and any function in $\mathcal{F}_i$ can be used.) Note that under this definition, $\sup_{x \in \mathcal{X}_i} |f_w(x) - f_i(x)| < \delta$ holds for each $i = 1, \ldots, B$. This can be confirmed as follows; given any $f_w$, we assume that there exists a point $x^* \in \mathcal{X}$ (the following discussion still holds when there are multiple such $x^*$ and we refer to them as $\mathcal{X}^*$) that achieves $\sup_x |f_w - h|_\infty$. If $f_w(x^*) \in I_i$, by the definition of $\mathcal{F}_i$, $|f_i(x^*) - f_w(x^*)| \leq \max\left[\min\left(h(x^*), \frac{i}{B}\right), \frac{i-1}{B} + \epsilon\right] - f_w(x^*)| \leq |h(x^*) - f_w(x^*)| \leq \delta$. For $x$, which does not achieves $\sup_x |f_w(x) - h(x)|_\infty$ (that is, $x \notin \mathcal{X}^*$) and satisfies $f_w(x) \in I_i$, then we have $|f_i(x) - f_w(x)| \leq \max\left[\min\left(h(x), \frac{i}{B}\right), \frac{i-1}{B} + \epsilon\right] - f_w(x)| \leq |h(x) - f_w(x)| \leq \delta$. So for any $x \in \mathcal{X}_i$, $|f_w(x) - f_i(x)| < \delta$ holds. Thus the function $f$ defined in Eq. (117) satisfies that for any $x$, if $f_w(x) \in I_i$, then $f$ satisfies $f(x) \in I_i$ and $\|f - f_w\|_\infty < \delta$.

With these settings, we consider replacing the functions in the total bias defined by above. Here in after, we set $\delta < 1/B$. Then the error in the TCE by replacing $f_w$ with $f$ is given as

$$|\mathbb{E}|\mathbb{E}[Y|f_w] - f_w| - \mathbb{E}|\mathbb{E}[Y|f] - f|| \leq (1 + L)\delta, \tag{119}$$

which is obtained by the Lipschitz assumption.

Next, we consider the error in the ECE by replacing $f_w$ with $f$ is given as follows;

$$|\mathrm{ECE}(f_w, S_{\mathrm{tr}}) - \mathrm{ECE}(f, S_{\mathrm{tr}})|$$

$$= |\sum_{i=1}^{B} |\mathbb{E}_{(X,Y)\sim\hat{S}_{\mathrm{tr}}}(Y - f_w(X)) \cdot \mathbb{1}_{f_w(X)\in I_i}| - \sum_{i=1}^{B} |\mathbb{E}_{(X,Y)\sim\hat{S}_{\mathrm{tr}}}(Y - f(X)) \cdot \mathbb{1}_{f(X)\in I_i}||$$

$$= |\sum_{i=1}^{B} |\mathbb{E}_{(X,Y)\sim\hat{S}_{\mathrm{tr}}}(Y - f_w(X)) \cdot \mathbb{1}_{f_w(X)\in I_i}| - \sum_{i=1}^{B} |\mathbb{E}_{(X,Y)\sim\hat{S}_{\mathrm{tr}}}(Y - f(X)) \cdot \mathbb{1}_{f_w(X)\in I_i}||$$

$$\leq |\sum_{i=1}^{B} |\mathbb{E}_{(X,Y)\sim\hat{S}_{\mathrm{tr}}}(f(X) - f_w(X)) \cdot \mathbb{1}_{f_w(X)\in I_i}||$$

$$= \sum_{i=1}^{B} \frac{|I_i|}{n_e} \left| \frac{1}{|I_i|} \sum_{m=1}^{n} \mathbb{1}_{f_w(x_m)\in I_i} f_w(x_m) - \frac{1}{|I_i|} \sum_{m=1}^{n} \mathbb{1}_{f_w(x_m)\in I_i} f(x_m) \right|$$

$$\leq \sum_{i=1}^{B} \frac{|I_i|}{n} \frac{\delta|I_i|}{|I_i|}$$

$$\leq \delta. \tag{120}$$

where $|I_i| := \sum_{m=1}^{n} \mathbb{1}_{f_w(x_m)\in I_i}$ and we used the fact that $\sum_i |I_i| = n$ and $\mathbb{1}_{f_w(X)\in I_i} = \mathbb{1}_{f(X)\in I_i}$ for any $x$ by definition.

Then the total bias is upper bounded by using $f$ as follows;

$$\mathbb{E}_{R,S_{\mathrm{tr}}}|\mathrm{TCE}(f_W) - \mathrm{ECE}(f_W, S_{\mathrm{tr}})|$$
$$= \mathbb{E}_{R,S_{\mathrm{tr}}}|\mathrm{TCE}(f_W) - \mathrm{TCE}(f) + \mathrm{TCE}(f) - \mathrm{ECE}(f_W, S_{\mathrm{tr}})|$$
$$\leq (1+L)\delta + \mathbb{E}_{R,S_{\mathrm{tr}}}|\mathrm{TCE}(f) - \mathrm{ECE}(f, S_{\mathrm{tr}})| + |\mathrm{ECE}(f, S_{\mathrm{tr}}) - \mathrm{ECE}(f_W, S_{\mathrm{tr}})|$$
$$\leq (2+L)\delta + \mathbb{E}_{R,S_{\mathrm{tr}}}|\mathrm{TCE}(f) - \mathrm{ECE}(f, S_{\mathrm{tr}})|. \tag{121}$$

Since the second term in the above represents the total bias of $f$, using Theorem 5

$$\mathbb{E}_{R,S_{\mathrm{tr}}}|\mathrm{TCE}(f) - \mathrm{ECE}(f, S_{\mathrm{tr}})| \leq \frac{1+L}{B} + \sqrt{\frac{8\left(\mathrm{eCMI}(\tilde{l}) + B\log 2\right)}{n}}, \tag{122}$$

where we replace $f_w$ appearing in the bound appearing Theorem 5 with $f$ defined above.

From the data processing inequality, we can upper bound the $\mathrm{eCMI}(\tilde{l})$ by the fCMI of $f$. Then such fCMI is bounded by the entropy of $f$ as follows

$$\mathrm{eCMI}(\tilde{l}) \leq \mathrm{fCMI}(f) = I(f(\tilde{X}); U|\tilde{Z}) \leq H[f(\tilde{X})|\tilde{Z}] - H[f(\tilde{X})|U, \tilde{Z}] \leq H[f(\tilde{X})|\tilde{Z}], \tag{123}$$

where we used the definition of mutual information and the conditional entropy of $f$ given $U$ ($H[f(\tilde{X})|U, \tilde{Z}]$) is larger than 0 because $f(\tilde{X})$ is the discrete random variable and the entropy of the discrete random variable is always larger than 0. We can confirm that $f(\tilde{X})$ is a discrete random variable since $f$ belongs to the function which is defined by the $\delta$ covering and the input $\tilde{X}$ is $2n$.

Then we upper bound $H[f(\tilde{X})|\tilde{Z}]$ by the log of the number of distinct values that are represented by Eq. (118) given $2n$ inputs of supersamples $\tilde{Z}$. We refer to it as $N$. Define $d_{2n}(f, g) := \max_{i\in[2n]} |f(x_i) - g(x_i)|$ for $f, g \in \mathcal{F}$. The $\delta$-covering number of $\mathcal{F}$ with respect to $d_{2n}$ is denoted as $\mathcal{N}(\mathcal{F}, d_{2n}, \delta)$, and we define $\mathcal{N}(\mathcal{F}, \delta, 2n) := \sup_{x^n\in\mathcal{X}^n} \mathcal{N}(\mathcal{F}, d_{2n}, \delta)$. It is known that $\mathcal{N}(\mathcal{F}, \delta, 2n) \leq \mathcal{N}(\mathcal{F}, \|\cdot\|_\infty, \delta)$ in general [37]. We focus on the fact that $f$ defined in Eq. (118) is the element of $\mathcal{F}_1 + \mathcal{F}_2 + \ldots, +\mathcal{F}_B$, where $\mathcal{F}_1 + \mathcal{F}_2$ implies the set of functions that is $\{f + g | f \in \mathcal{F}_1, g \in \mathcal{F}_2\}$. Note that the covering number of sum of two functions are upper bounded by the multiplication of each covering number, that is

$$\mathcal{N}(\mathcal{F}_1 + \mathcal{F}_2, \delta, 2n) \leq \mathcal{N}(\mathcal{F}_1, \delta/2, 2n)\mathcal{N}(\mathcal{F}_2, \delta/2, 2n) \tag{124}$$

and thus, we have

$$\mathcal{N}(\mathcal{F}_1 + \mathcal{F}_2 + \ldots, +\mathcal{F}_B, \delta, 2n) \leq \prod_{i=1}^{B} \mathcal{N}(\mathcal{F}_i, \delta/B, 2n) \leq (\mathcal{N}(\mathcal{F}_i, \delta/B, 2n))^B. \tag{125}$$

Thus we have

$$N \leq \prod_{i=1}^{B} (\mathcal{N}(\mathcal{F}, \delta, 2n)) \leq (\mathcal{N}(\mathcal{F}, \| \cdot \|_\infty, \delta/B))^B. \tag{126}$$

In conclusion, we have that

$$\mathrm{eCMI}(\tilde{l}) \leq \log N \leq B \log \mathcal{N}(\mathcal{F}, \| \cdot \|_\infty, \delta/B). \tag{127}$$

Combining these we have

$$
\begin{aligned}
\mathbb{E}_{R, S_{\mathrm{tr}}} \mathrm{Bias}_{\mathrm{tot}}(f_W, S_{\mathrm{tr}}) &\leq \frac{1 + L}{B} + (2 + L)\delta + \sqrt{\frac{8\left(B \log \mathcal{N}(\mathcal{F}, \| \cdot \|_\infty, \delta/B) + B \log 2\right)}{n}} \\
&= \frac{1 + L}{B} + (2 + L)\delta + \sqrt{\frac{8B \log 2\mathcal{N}(\mathcal{F}, \| \cdot \|_\infty, \delta/B)}{n}}.
\end{aligned}
\tag{128}
$$

Finally, as for the order discussion, for example, we can obtain $\mathcal{N}(\mathcal{F}, \| \cdot \|_\infty, \delta) \asymp \left(\frac{L_0}{\delta}\right)^d$ when $f_w$ is a $d$-dimensional parametric function that is $L_0$-Lipschitz continuous ($L_0 > 0$) [37], leading to the following upper bound:

$$\mathbb{E}_{R, S_{\mathrm{tr}}} \mathrm{Bias}_{\mathrm{tot}}(f_W, S_{\mathrm{tr}}) \lesssim \frac{3 + 2L}{B} + \sqrt{8 \frac{dB \log 2L_0 B^2}{n}}, \tag{129}$$

where we set $\delta = \mathcal{O}(1/B)$. This bound is minimized when $B = \mathcal{O}(n^{1/3})$, resulting in a bias of $\mathcal{O}(\log n / n^{1/3})$. $\qquad \square$

Here we provide additional discussion about the metric entropy. The bound based on the metric entropy depends on the model's dimensionality, making them unsuitable for large models such as neural networks. Some existing studies [15, 17] present the upper bound of the eCMI and fCMI by the dimension-independent complexities, such as the VC dimension for binary classification and connecting IT theory to UC theory.

Inspired by these results, here we provide the upper bound of eCMI and fCMI using such dimension-independent complexities. As provided in the lower bound analysis above, since TCE estimation is similar to the nonparametric regression, we use the fat-shattering dimension [1] to upper bound the eCMI. Specifically, from Lemma 3.5 in Alon et al. [1] in if our model class $f_w(\cdot)$ satisfies $\delta/4$-fat dimension with $d_{\delta/4}$ for $\delta \in [0, 1]$, we have

$$\mathrm{eCMI} \leq \mathrm{fCMI} = \mathcal{O}\left(d_{\delta/4} \log \frac{n}{d_{\delta/4}\delta} \log\left(\frac{n}{\delta^2}\right)\right) \tag{130}$$

which results in the dimension-independent upper bound. To evaluate the fat-shattering dimension for specific models, see Bartlett & Maass [2] for the details.

### E.4  Proof of Theorem 7 (recalibration)

*Proof.* Recall the definition of the recalibration. Here we show the expression when we use the training dataset $S_{\mathrm{tr}}$:

$$h_{\mathcal{I}, S_{\mathrm{tr}}}(x) = \sum_{i=1}^{B} \hat{\mu}_{i, S_{\mathrm{tr}}} \cdot \mathbb{1}_{f_w(x) \in I_i}, \tag{131}$$

$$\hat{\mu}_{i, S_{\mathrm{tr}}} := \frac{\sum_{m=1}^{n} y_m \cdot \mathbb{1}_{f_w(x_m) \in I_i}}{\sum_{m=1}^{n} \mathbb{1}_{f_w(x_m) \in I_i}}. \tag{132}$$

The proof is similar to the proof of Theorem 8. We use Eqs. (87) and (86), let $S_{\mathrm{tr}} = \{Z_m\}_{m=1}^{n}$, then we have

$$\mathbb{E}_{R,S_{\mathrm{tr}}}\mathbb{E}[|\mathbb{E}[Y|h_{\mathcal{I},S_{\mathrm{tr}}}(x)] - h_{\mathcal{I},S_{\mathrm{tr}}}(x)]$$

$$= \mathbb{E}_{R,S_{\mathrm{tr}}}\left|\sum_{i=1}^{B}\left|\mathbb{E}_{Z''=(X'',Y'')}\left[(Y'' - \hat{\mu}_{i,S_{\mathrm{tr}}}) \cdot \mathbb{1}_{f_W(X'')\in I_i}\right]\right|\right|$$

$$= \mathbb{E}_{R,S_{\mathrm{tr}}}\sum_{i=1}^{B}\left|\mathbb{E}_{\{Z'_m\}_{m=1}^n}\frac{1}{n}\sum_{m=1}^{n}\left[(Y'_m - \hat{\mu}_{i,S_{\mathrm{tr}}}) \cdot \mathbb{1}_{f_W(X'_m)\in I_i}\right]\right|$$

$$\leq \underset{R,\{Z_m\}_{m=1}^n,\{Z'_m\}_{m=1}^n}{\mathbb{E}}\sum_{i=1}^{B}$$

$$\left|\frac{1}{n}\sum_{m=1}^{n}\left[(Y'_m\cdot\mathbb{1}_{f_W(X'_m)\in I_i} - \hat{\mu}_{i,S_{\mathrm{tr}}}\cdot\mathbb{1}_{f_W(X_m)\in I_i} + \hat{\mu}_{i,S_{\mathrm{tr}}}\cdot\mathbb{1}_{f_W(X_m)\in I_i} - \hat{\mu}_{i,S_{\mathrm{tr}}}\cdot\mathbb{1}_{f_W(X'_m)\in I_i})\right]\right|$$

$$= \underset{R,\{Z_m\}_{m=1}^n,\{Z'_m\}_{m=1}^n}{\mathbb{E}}\sum_{i=1}^{B}$$

$$\left|\frac{1}{n}\sum_{m=1}^{n}\left[(Y'_m\cdot\mathbb{1}_{f_W(X'_m)\in I_i} - Y_m\cdot\mathbb{1}_{f_W(X_m)\in I_i}) + \hat{\mu}_{i,S_{\mathrm{tr}}}(\mathbb{1}_{f_W(X_m)\in I_i} - \mathbb{1}_{f_W(X'_m)\in I_i})\right]\right|$$

$$\leq \underset{R,\{Z_m\}_{m=1}^n,\{Z'_m\}_{m=1}^n}{\mathbb{E}}\sum_{i=1}^{B}\left|\frac{1}{n}\sum_{m=1}^{n}\left[(Y'_m\cdot\mathbb{1}_{f_W(X'_m)\in I_i} - Y_m\cdot\mathbb{1}_{f_W(X_m)\in I_i})\right]\right|$$

$$+ \underset{R,\{Z_m\}_{m=1}^n,\{Z'_m\}_{m=1}^n}{\mathbb{E}}\sum_{i=1}^{B}\left|\frac{1}{n}\sum_{m=1}^{n}\left[(\mathbb{1}_{f_W(X_m)\in I_i} - \mathbb{1}_{f_W(X'_m)\in I_i})\right]\right| \tag{133}$$

where we used the Jensen inequality first and the used the triangle inequality and used the fact that $|\hat{\mu}_{i,S_{\mathrm{tr}}}| \leq 1$ by the definition of UMB in the next inequality.

We then rewrote the above in the CMI setting, we have

$$\mathbb{E}_{R,S_{\mathrm{tr}}}\mathbb{E}[|\mathbb{E}[Y|h_{\mathcal{I},S_{\mathrm{tr}}}(x)] - h_{\mathcal{I},S_{\mathrm{tr}}}(x)]$$

$$\leq \mathbb{E}_{R,\tilde{Z},U}\sum_{i=1}^{B}\left|\frac{1}{n}\sum_{m=1}^{n}(\tilde{Y}_{m,\bar{U}_m}\cdot\mathbb{1}_{f_{\mathcal{A}(\tilde{Z}_U,R)}(\tilde{X}_{m,\bar{U}_m})\in I_i} - \tilde{Y}_{m,U_m}\cdot\mathbb{1}_{f_{\mathcal{A}(\tilde{Z}_U,R)}(\tilde{X}_{m,U_m})\in I_i}\right|$$

$$+ \mathbb{E}_{R,\tilde{Z},U}\sum_{i=1}^{B}\left|\frac{1}{n}\sum_{m=1}^{n}\mathbb{1}_{f_{\mathcal{A}(\tilde{Z}_U,R)}(\tilde{X}_{m,\bar{U}_m})\in I_i} - \mathbb{1}_{f_{\mathcal{A}(\tilde{Z}_U,R)}(\tilde{X}_{m,U_m})\in I_i}\right|$$

$$\leq \sqrt{\frac{2(I(\Delta_1(U,\tilde{Z},\mathcal{A}(\tilde{Z}_U,R));U|\tilde{Z}) + B\log 2)}{n}} + \sqrt{\frac{2(I(\Delta_2(U,\tilde{Z},\mathcal{A}(\tilde{Z}_U,R));U|\tilde{Z}) + B\log 2)}{n}}.$$
$$\tag{134}$$

where

$$\Delta_1(U,\tilde{Z},\mathcal{A}(\tilde{Z}_U,R)) := \sum_{i=1}^{B}\left|\frac{1}{n}\sum_{m=1}^{n}(\tilde{Y}_{m,\bar{U}_m}\cdot\mathbb{1}_{f_{\mathcal{A}(\tilde{Z}_U,R)}(\tilde{X}_{m,\bar{U}_m})\in I_i} - \tilde{Y}_{m,U_m}\cdot\mathbb{1}_{f_{\mathcal{A}(\tilde{Z}_U,R)}(\tilde{X}_{m,U_m})\in I_i})\right|$$
$$\tag{135}$$

$$\Delta_2(U,\tilde{Z},\mathcal{A}(\tilde{Z}_U,R)) := \sum_{i=1}^{B}\left|\frac{1}{n}\sum_{m=1}^{n}(\mathbb{1}_{f_{\mathcal{A}(\tilde{Z}_U,R)}(\tilde{X}_{m,\bar{U}_m})\in I_i} - \mathbb{1}_{f_{\mathcal{A}(\tilde{Z}_U,R)}(\tilde{X}_{m,U_m})\in I_i})\right| \tag{136}$$

where the last line is we repeat the proof of Theorem 8. The proof of Theorem 8 has discussed the loss composed of the indicator function, we can use exactly the same proof procedure. The only difference is the CMI; here we consider the eCMI which uses above $\Delta_1$ and $\Delta_2$ as the random variables, and their CMIs are the conditional mutual information between $\Delta_1$ and $\Delta_2$.

From Eq. (134), we can further simplify the upper bound using the data processing inequality,

$$I(\Delta_1(U, R, \tilde{Z}); U|\tilde{Z}), I(\Delta_2(U, R, \tilde{Z}); U|\tilde{Z}) \leq \text{eCMI} \tag{137}$$

$$\text{eCMI} := I(l(\mathcal{A}(\tilde{Z}_U, R), \tilde{Z}, B); U|\tilde{Z}), \tag{138}$$

$$l(\mathcal{A}(\tilde{Z}_U, R), z, B) := (\mathbb{1}_{f_{\mathcal{A}(\tilde{Z}_U, R)}(x) \in I_1}, \ldots, \mathbb{1}_{f_{\mathcal{A}(\tilde{Z}_U, R)}(x) \in I_B}). \tag{139}$$

We then have

$$\mathbb{E}_{R, S_{\text{tr}}}\mathbb{E}[|\mathbb{E}[Y|h_{\mathcal{I}, S_{\text{tr}}}(x)] - h_{\mathcal{I}, S_{\text{tr}}}(x)] \leq 2\sqrt{\frac{2(\text{eCMI} + B\log 2)}{n}} \leq 2\sqrt{\frac{2(\text{fCMI} + B\log 2)}{n}}. \tag{140}$$

$$\square$$

In the numerical experiments, we use the tighter version, which appears in the proof above;

**Corollary 4.** *Under the same setting and assumptions as Theorem 7, we have*

$$\mathbb{E}_{R, S_{\text{tr}}}\mathbb{E}[|\mathbb{E}[Y|h_{\mathcal{I}, S_{\text{tr}}}(x)] - h_{\mathcal{I}, S_{\text{tr}}}(x)]$$

$$\leq \sqrt{\frac{2(I(\Delta_1(U, \tilde{Z}, \mathcal{A}(\tilde{Z}_U, R)); U|\tilde{Z}) + B\log 2)}{n}} + \sqrt{\frac{2(I(\Delta_2(U, \tilde{Z}, \mathcal{A}(\tilde{Z}_U, R)); U|\tilde{Z}) + B\log 2)}{n}}, \tag{141}$$

*where*

$$\Delta_1(U, \tilde{Z}, \mathcal{A}(\tilde{Z}_U, R)) := \sum_{i=1}^{B}\left|\frac{1}{n}\sum_{m=1}^{n}(\tilde{Y}_{m, \bar{U}_m} \cdot \mathbb{1}_{f_{\mathcal{A}(\tilde{Z}_U, R)}(\tilde{X}_{m, \bar{U}_m}) \in I_i} - \tilde{Y}_{m, U_m} \cdot \mathbb{1}_{f_{\mathcal{A}(\tilde{Z}_U, R)}(\tilde{X}_{m, U_m}) \in I_i})\right| \tag{142}$$

$$\Delta_2(U, \tilde{Z}, \mathcal{A}(\tilde{Z}_U, R)) := \sum_{i=1}^{B}\left|\frac{1}{n}\sum_{m=1}^{n}\mathbb{1}_{f_{\mathcal{A}(\tilde{Z}_U, R)}(\tilde{X}_{m, \bar{U}_m}) \in I_i} - \mathbb{1}_{f_{\mathcal{A}(\tilde{Z}_U, R)}(\tilde{X}_{m, U_m}) \in I_i})\right|. \tag{143}$$

In the main paper, we further upper bound eCMIs by fCMI, which is followed by the data processing inequality.

### E.5 Recalibration when using test data

Here we show the bias of the recalibration when the test (recalibration data) is used.

**Corollary 5.** *Conditioned on $W = w$, under the same assumption as Theorem 2, we have*

$$\mathbb{E}_{S_{\text{re}}}\mathbb{E}[|\mathbb{E}[Y|h_{\mathcal{I}, S_{\text{re}}}(X)] - h_{\mathcal{I}, S_{\text{re}}}(X)] \leq \sqrt{2B\log 2/(n_{\text{re}} - B)} + 2B/(n_{\text{re}} - B) \tag{144}$$

*Proof.* The recalibration bias corresponds to the statistical bias because from Theorem 2, we have

$$\mathbb{E}_{S_{\text{re}}}\text{Bias}_{\text{stat}}(h, S_{\text{re}}) = \mathbb{E}_{S_{\text{re}}}|\text{TCE}(h_{\mathcal{I}}) - \text{ECE}(h_w, S_{\text{re}})|$$

$$\leq \sqrt{2B\log 2/(n_{\text{re}} - B)} + 2B/(n_{\text{re}} - B). \tag{145}$$

where $h_{\mathcal{I}}$ is the conditional expectation of $h_{\mathcal{I}, S_{\text{re}}}$ given bins $\mathcal{I}$. Note that by the definition of $h_{\mathcal{I}, S_{\text{re}}}$, from the tower property $h_{\mathcal{I}}(x) = h_{\mathcal{I}, S_{\text{re}}}(x)$ holds since we take the expectation in each bin. Thus, by definition $\mathbb{E}_{S_{\text{re}}}\text{TCE}(h_{\mathcal{I}}) = \mathbb{E}_{S_{\text{re}}}\mathbb{E}[|\mathbb{E}[Y|h_{\mathcal{I}, S_{\text{re}}}(X)] - h_{\mathcal{I}, S_{\text{re}}}(X)]$ holds. Moreover $\mathbb{E}_{S_{\text{re}}}\text{ECE}(h_w, S_{\text{re}}) = 0$ by definition, since this is the definition of the recalibrated function. Thus, we obtain the result. $\square$

## F  Further discussion

We have presented the results of our analyses of (i) the total bias in estimating the TCE and (ii) the generalization error analysis for the ECE and the TCE thus far. In this section, we explain the difference between our study and the existing work in the calibration context.

## F.1 Discussion about the assumption

Here we discuss the necessity of Assumption 1. The reasons of using this assumption are two-fold: **i) :** The first purpose is that we want to use the results in Gupta & Ramdas [12], which analyzes the statistical bias of the UMB. Their proofs use the existence of the density of $f_w(x)$, so Assumption 1 cannot be eliminated. **ii) :** The other purpose is that by Assumption 1, we want to use the fact that $\{f_w(x_m)\}_{m=1}^{n_{te}}$ in $x_m \in S_{te}$ takes the distinct values almost surely (same for $\{f_w(x_m)\}_{m=1}^n$ in $x_m \in S$).

Regarding **i)**, we used results in Gupta & Ramdas [12] to prove the result of UMB in Eq. (9) of Theorem 2 and the result of UMB of Theorem 3. Thus, the results using these theorems require Assumption 1. They correspond to the results related to the bias of UMB. So the results of UWB essentially do not require this assumption.

The situation becomes complicated when considering the generalization error analysis. Our analysis uses the IT-based approach and does not use the results in Gupta & Ramdas [12], so we do not need Assumption 1 regarding **i)**. However, if all $\{f_w(x_m)\}_{m=1}^n$ in $x_m \in S$ takes the same value, we cannot construct the bins in UMB. So when considering the training data reuse, we need Assumption 1 to construct the bins of UMB. However, we remark that we can replace Assumption 1 with the assumption that "we assume that $\{f_w(x_m)\}_{m=1}^n$ in $x_m \in S$ takes distinct values almost surely".

When considering the UWB, we do not suffer from such troubles since we simply split the interval $[0, 1]$ with equal width as the $b$-th interval is given as $((b-1)/B, b/B]$. However, there might be a chance that all the $\{f_w(x_m)\}_{m=1}^{n_{te}}$ in $x_m \in S_{te}$ takes $b/B$, then the coefficients of the bound changes. Recall that our proof uses the bounded difference property when upper bounding the exponential moment, for example, in Eq. (93) and Eq. (94). That estimation is based on the fact that $\{f_w(x_m)\}_{m=1}^n$ in $x_m \in S_{tr}$ takes different values. So if all the $f$ takes the same value, the upper bound of the bounded difference will change, which results in the different coefficients in our bound, although we can proceed with the proof in the same way.

For these reasons, we decided to impose Assumption 1 for all the statements. As we discussed above, if we focused on the specific setting, such as UWB and UMB, then there is room to eliminate or replace the assumption.

Next, we discuss the necessity of Assumption 2. Estimating TCE involves nonparametric regression of $\mathbb{E}[Y|f(X) = v]$. For finite samples, smoothness assumptions like Lipschitz continuity are required; without them, small changes in $v$ could cause large variations in label outcomes, making estimation impossible [26]. Such smoothness assumptions are standard in nonparametric regression, including kernel-based ECE. Therefore, without these assumptions, increasing the sample size would not ensure that training (or test) ECE converges to TCE. As noted in our minimax lower bound discussion, the absence of smoothness ($\beta \to 0$) leads to increasing bias.

In practice, Assumption 2 is reasonably mild. For example, if label distributions follow a Bernoulli distribution with the mean depending on the input $x$, Assumption 2 is satisfied [47]. This is a relatively weak assumption in binary classification. Moreover, existing studies have shown that many common benchmark datasets are consistent with this assumption (e.g., [49]). Additionally, our numerical experiments with the data used in this study confirmed that the conditions for smoothness, such as non-diverging first derivatives, are indeed satisfied (see Figure 6 in Appendix H.4). This supports the robustness of our assumptions and the applicability of our methods in real-world scenarios.

Finally, regarding the assumption $n_e \geq 2B$, it is crucial for UMB, as it guarantees the proper construction of bins. In UMB, we first use $B$ samples to build the bins. We then partition the remaining $n_e - B$ samples evenly across these bins. If $n_e \geq 2B$, we have $n_e - B \geq B$, preventing equal distribution of samples, thereby rendering UMB inapplicable. Conversely, UWB does not require this assumption since it divides the $[0, 1]$ interval into equal-width bins.

## F.2 Discussion about our proof techniques

Here we discuss our proof techniques. In our proof for UWB, our proof technique does not heavily depend on the binning construction method, so we can apply our technique to other than UWB and UMB. The important ingredients are the boundedness of $y$ and $f(x)$ and the property of the indicator function. However, our proof builds on the reformulation of Eqs. (24) and (27) this can be

a restriction for some settings. For example, when we consider higher-order ECEs defined as

$$\mathrm{ECE}(f_w, S_e) \coloneqq \sum_{i=1}^{B} p_i |\bar{f}_{i,S_e} - \bar{y}_{i,S_e}|^p, \tag{146}$$

with $p > 1$, which can not be reformulated like Eqs. (24) and (27), and thus our proof technique cannot be applicable.

On the other hand, as we introduce in the above, the technique of Gupta & Ramdas [12] can apply to ECEs with $p > 1$. However, the drawback is that their technique can only apply to UWB without training data reuse.

Next, we discuss our results with Wang et al. [39]. The eCMI appearing in Theorem 4 closely aligns with the $\Delta L$ bound (loss difference) of Theorem 1 of reference Wang et al. [39]. Specifically, the eCMI term in Theorem 4 is not based on the value of ECE itself. Instead, it is derived from the difference between the test data ECE and the training data ECE. This approach aligns with the $\Delta L$ (loss difference) as shown in Theorem 1 of reference Wang et al. [39].

However, extending our bounds using the techniques of Wang et al. [39] presents significant challenges. The $\Delta L$ bound in Wang et al. [39] defines the loss gap for a single data index $i$ as $\Delta L_i$ and utilizes the symmetry of each individual index to derive fast rate bounds, as demonstrated in Theorem 4.3. In contrast, our bound requires treating all $n$ indices simultaneously. This necessity arises because ECE is a nonparametric estimator that uses all $n$ indices, unlike usual losses such as the 0-1 loss, where an estimator can be constructed using a single index. Consequently, the techniques from Wang et al. [39] that utilize the symmetry of a single index are not applicable to our context, making it difficult to employ the methods from Wang et al. [39].

## F.3 Comparison of our bound with existing and trivial bounds

Here we discuss the order of our generalization error bias in more depth. Recall that our Theorem 4 is

$$\mathbb{E}_{R, S_{\mathrm{tr}}, S_{\mathrm{te}}} |\mathrm{ECE}(f_W, S_{\mathrm{te}}) - \mathrm{ECE}(f_W, S_{\mathrm{tr}})| \leq \sqrt{\frac{8(\mathrm{eCMI} + B \log 2)}{n}}, \tag{147}$$

and the important property is that the bound is of order $\mathcal{O}(\sqrt{B/n})$ if we neglect the order of eCMI.

Here we see that when we use existing Theorem 1 directly, the resulting bound is $\mathcal{O}(B/\sqrt{n})$. Recall that

$$\mathrm{ECE}(f, S_{\mathrm{tr}}) = \sum_{i=1}^{B} |\mathbb{E}_{(X,Y) \sim \hat{S}_{\mathrm{tr}}} (Y - f_w(X)) \cdot \mathbb{1}_{f_w(X) \in I_i}|,$$

$$\mathrm{TCE}(f_{\mathcal{I}}) = \sum_{i=1}^{B} |\mathbb{E}_{(X,Y) \sim \mathcal{D}} (Y - f_w(X)) \cdot \mathbb{1}_{f_w(X) \in I_i}|.$$

where $\hat{S}_{\mathrm{tr}}$ is the empirical distribution of the training dataset. Thus

$$\mathbb{E}_{S_{\mathrm{tr}}, S_{\mathrm{te}}, R} |\sum_{i=1}^{B} |\mathbb{E}_{(X,Y) \sim \mathcal{D}} (Y - f_W(X)) \cdot \mathbb{1}_{f_W(X) \in I_i}| - \sum_{i=1}^{B} |\mathbb{E}_{(X,Y) \sim \hat{S}_{\mathrm{tr}}} (Y - f_W(X)) \cdot \mathbb{1}_{f_W(X) \in I_i}||$$

$$\leq \mathbb{E}_{S_{\mathrm{tr}}, S_{\mathrm{te}}, R} |\sum_{i=1}^{B} |\mathbb{E}_{(X,Y) \sim \mathcal{D}} (Y - f_W(X)) \cdot \mathbb{1}_{f_W(X) \in I_i} - \mathbb{E}_{(X,Y) \sim \hat{S}_{\mathrm{tr}}} (Y - f_W(X)) \cdot \mathbb{1}_{f_W(X) \in I_i}||$$

$$= \mathbb{E}_{S_{\mathrm{tr}}, S_{\mathrm{te}}, R} \sum_{i=1}^{B} |\mathbb{E}_{(X,Y) \sim \mathcal{D}} (Y - f_W(X)) \cdot \mathbb{1}_{f_W(X) \in I_i} - \mathbb{E}_{(X,Y) \sim \hat{S}_{\mathrm{tr}}} (Y - f_W(X)) \cdot \mathbb{1}_{f_W(X) \in I_i}|$$

$$\leq \sum_{i=1}^{B} \sqrt{\frac{2}{n} (\mathrm{eCMI}(\tilde{l}_i) \log 2)}$$

$$\leq B \sqrt{\frac{2}{n} (\mathrm{fCMI} + \log 2)}, \tag{148}$$

where we used the triangle inequality in the second line and from the third line to the fourth line, we applied Eq. (5) in Theorem 1 by fixing the binning index $i$ and eCMI is where $\text{eCMI}(\tilde{l}_i) = I(\tilde{l}_i; U|\tilde{Z})$

$$
\begin{aligned}
&\tilde{l}_i(U, R, \tilde{Z}) \\
&:= \big| \frac{1}{n} \sum_{m=1}^{n} (\tilde{Y}_{m,\bar{U}_m} - f_{\mathcal{A}(\tilde{Z}_U, R)}(\tilde{X}_{m,\bar{U}_m})) \cdot \mathbb{1}_{f_{\mathcal{A}(\tilde{Z}_U, R)}(\tilde{X}_{m,U_m}) \in I_i} \\
&\quad - \frac{1}{n} \sum_{m=1}^{n} (\tilde{Y}_{m,U_m} - f_{\mathcal{A}(\tilde{Z}_U, R)}(\tilde{X}_{m,U_m})) \cdot \mathbb{1}_{f_{\mathcal{A}(\tilde{Z}_U, R)}(\tilde{X}_{m,U_m}) \in I_i} \big| \quad (149)
\end{aligned}
$$

Thus, this proof is simple compared to our proof of Theorem 4, but results in a worse dependency on $B$. In our proof, we used the property of the binning and indicator function of the loss function explicitly, which results in better dependency. On the other hand, when deriving Eq. (148), we do not use such properties, and thus results in worse dependency on $B$.

### F.4 Discussion about the order of eCMI and fCMI

Here we discuss when eCMI and fCMI can be controlled theoretically.

As discussed in Section 4, from data processing inequality [6], we have that $\text{eCMI} \leq \text{fCMI} \leq I(W; S)$. Since fCMI does not depend on $B$, and the dependency on $B$ of fCMI, eCMI is not a problem.

Here, we cite the classical result about $I(W; S)$. Clarke & Barron [5] (see also Rissanen [30], Haussler & Opper [16]) clarified that the growth rate of MI can be controlled as follows: if $w$ takes a value in a $d$-dimensional compact subset of $\mathbb{R}^d$ and $p(y|x; w)$ is smooth in $w$, then as $n \to \infty$, we have

$$
I(W; S) = \frac{d}{2} \log \frac{n}{2\pi e} + h(W) + \mathbb{E} \log \det J + o(1),
$$

where $h(W)$ is the differential entropy of $W$, and $J$ is the Fisher information matrix of $p(Y|X; W)$.

Moreover, Steinke & Zakynthinou [34] introduced the CMI that satisfies $\text{fCMI} \leq \text{CMI}$ discussed the CMI is upper bounded by the various notions of stability. For example, if the training algorithm satisfies $\sqrt{2\epsilon}$-differentially private (DP) algorithm, then CMI is upper-bounded by $\epsilon n$. So this $\epsilon$ is controlled by the DP algorithm, then our eCMI can also be controlled appropriately. For example, Xu & Raginsky [43] discussed that the Gibbs algorithm satisfies $\mathcal{O}(1/n)$-DP when the loss takes value $[0, 1]$. Thus, such Gibbs algorithms can control our eCMI moderately.

Steinke & Zakynthinou [34] also clarified that if the algorithm is $\delta$ stable in total variation distance, then CMI is upper bounded by $\delta n$. From the stability perspective, Mou et al. [28] showed that SGLD satisfies $\mathcal{O}(\frac{T}{n^2})$ stability in the Hellinger distance and $T$ is the iteration number of the SGLD algorithm. Thus, this implies that when $T$ is small, the eCMI of SGLD can be very small. Recently, Farghly & Rebeschini [8] and Futami & Fujisawa [9] showed that under the certain non-convexity assumption, SGLD satisfies the Wasserstein and KL stability of the order of $\mathcal{O}(1/n)$, which also result in the eCMI of SGLD.

### F.5 Relation to the existing study of calibration

Here we discuss other existing work, which is not shown in the main paper mainly due to the space limitation. First, we compare our result and existing analysis by Gupta & Ramdas [12] in Appendix F.2 in detail.

We discuss in Appendix D.2 how our proof technique improves the trivial dependency of $\mathcal{O}(B)$ to our $\mathcal{O}(\sqrt{B})$ for the ECE with the test dataset. We show a similar discussion for the ECE with training dataset reuse.

Kulynych et al. [22] also discussed the relation between generalization and calibration. However, there are two distinct differences from theirs; One is that they only discuss the statistical bias not consider the binning bias. The other is that their statistical bias is of $\mathcal{O}(B)$ while ours is $\mathcal{O}(\sqrt{B})$, which is a significant improvement.

Carrell et al. [4] numerically evaluated the generalization gap of the calibration, which is close to the statistical bias in our training reuse setting. They focused on the numerical aspects and statistical bias, while ours focused on the theoretical analysis and focuses on both the binning and statistical biases.

Gruber & Buettner [10] studied the various statistics related to calibration error. While our study rigorously analyzes both the binning and statistical bias, their work focuses on the asymptotic settings and has not derived the dependency of $B$ and $n$.

There is an additional comparison of our analysis with Gupta & Ramdas [12] in Appendix D.6

### F.6   Discussion about the lower bound

Here, we discuss the lower bound of the bias when estimating the TCE from the following two viewpoints; the TCE estimation can be seen as (i) *estimating a parameter in each bin*, and (ii) *estimating a one-dimensional function in a pointwise*. To understand this, we start deriving the following lower bound by using Jensen inequality:

$$\text{TCE}(f_w) \geq |\mathbb{E}[Y - f_w(X)]|. \tag{150}$$

This bound suggests that estimating $\mathbb{E}[Y] = f_w(x)$ achieves the small TCE. From the classical theory, for any distribution $\mathcal{D}$, the lower bound of a parameter estimation bias is $1/\sqrt{n}$ [37]. Eq. (150) corresponds to the setting of $B = 1$ bin to estimate the conditional expectation in the ECE. With these observations, we discuss the statistical bias of UMB. In UMB, we estimate the conditional expectation using $m = n/B$ samples in each bin and this is a parameter estimation problem. Thus this leads to $\mathcal{O}(1/\sqrt{m})$ bias from the classical theory. On the other hand, we derived the statistical bias $\mathcal{O}(\sqrt{B/n})$ for UMB, thus it is optimal when viewed as the parameter estimation.

However, using a constant function $f_w(x) = \mathbb{E}[Y]$, which achieves the small TCE, is useless in practice. Our original motivation is to measure the perfect calibration, which requires estimating the conditional expectation $\mathbb{E}[Y|f_w(x) = p]$ for the interval $p \in [0, 1]$ in a pointwise, and this is a function estimation problem. Thus, the bins used in the ECE adjust that whether estimating ECE is close to the parameter estimation or the function estimation. Then this trade-off is captured by the binning bias in our analysis. So the total bias represents such a trade-off whether the problem is the parameter or function estimation.

From the classical theory of Fano's method [37], when we estimate the Lipschitz function with a closed interval, it achieves a lower bound $\mathcal{O}(1/n^{d+2})$, with $d$ as the input dimension of the function. In the calibration, the input is the probability $p$ and thus $d = 1$. Since we derived that the bias of the ECE is $\mathcal{O}(1/n^{1/3})$, it achieves the minimax rate if the underground conditional expectation $\mathbb{E}[Y|v = p]$ satisfies Lipsthitz continuity.

Moreover, when the $d$-dimensional target function satisfies stronger smoothness assumption, $\beta$-Hölder continuity, we suffer the bias of $\mathcal{O}(1/n^{\frac{\beta}{2\beta+d}})$ [36]. So if $\mathbb{E}[Y|v = p]$ satisfies such conditions, the lower bound of the bias should be $\mathcal{O}(1/n^{\frac{\beta}{2\beta+1}})$, however as discussed in Appendix D.5, the binning method cannot achieve this rate and thus cannot utilize the smoothness of the data distribution.

### F.7   Additional discussion about the lower bound

Here we discuss additional points regarding the TCE bias estimation. Conditioned on $W = w$, let $v = f_w(x)$ and the distribution induced by $p(x)$ by $f_w(x)$ over $v$ as $p(v)$. We express the support of $p(v)$ as $\mathcal{V} \subset [0, 1]$. Let $g(v) = \mathbb{E}[Y = 1|f_w(x) = v] = \Pr[Y = 1|v]$. Let $\mathcal{G}$ be a class of candidate conditional probability functions over $\mathcal{V}$ and every candidate $g \in \mathcal{G}$ satisfies $0 \leq g(v) \leq 1$ for all $v \in \mathcal{V}$. Let us write the L2 minimax error as

$$R(\mathcal{G}; n) := \inf_{\hat{g}} \sup_{g \in \mathcal{G}} \mathbb{E}|\hat{g}(V) - g(V)|^2, \tag{151}$$

where $\hat{g}$ is over all valid estimators for $g$ using $n$ samples $(X_m, Y_m)_{m=1}^n$ and the expectation is taken with respect to true $g$. The L2 and higher order minimax rate has been shown in [45] by using the Yang-Barron method [46], which is the mutual information-based approach stemming from Fano's method. To cite this result, we need additional assumptions that control the mutual information [45];

**Assumption 3.** *Assume that $\mathcal{G}$ has at least one member $g^*$ that is bounded away from 0 and 1, i.e., there exist constants $0 < c_1 \leq c_2 < 1$ such that $c_1 \leq g^* \leq c_2$.*

Here we assume that $\mathcal{G}$ is a set of functions that satisfies the $\beta$-Lipschitzness; for any $g \in \mathcal{G}$, there exists positive constant $C$ that satisfies $|g(v) - g(v')| \leq C|v - v'|^\beta$ for any $v, v' \in \mathcal{V}$.

According to Lemma 1 in [45], if $\beta$-Lipschitzness and we have

$$R(\mathcal{G}; n) \succeq n^{-2\beta/(2\beta+1)}. \tag{152}$$

Thus, if we consider the histogram-based estimator for $\hat{g}$

$$\hat{g}_{\mathrm{bin}}(v) := \sum_{i=1}^{B} \bar{y}_{i,S_{\mathrm{te}}} \cdot \mathbb{1}_{v \in I_i}, \tag{153}$$

which is used in the definition of the binning ECE, then

$$\sup_{g \in \mathcal{G}} \mathbb{E}|\hat{g}_{\mathrm{bin}}(V) - g(V)|^2 \geq \inf_{\hat{g}} \sup_{g \in \mathcal{G}} \mathbb{E}|\hat{g}(V) - g(V)|^2 \succeq n^{-2\beta/(2\beta+1)}. \tag{154}$$

Thus we can obtain the lower bound of the recalibrated function from true conditional expectation.

Here we discuss how this lower bound is related to the total bias of the ECE. Let us define the semi-metric

$$\rho(g, g') := |\mathbb{E}[|g(V) - V|] - \mathbb{E}[|g'(V) - V|]|. \tag{155}$$

We can easily confirm that $\rho$ satisfies the positivity and triangle inequality, so this is the semi-metric. We can show the following relation for the total bias and this $\rho$ as follows; recall that we can express the TCE as

$$\mathrm{TCE}(f_w) = \mathbb{E}[|g(V) - V|] \tag{156}$$

and

$$\mathbb{E}_{S_{\mathrm{te}}} \mathrm{ECE}(f_w, S_{\mathrm{te}}) = \mathbb{E}[|(\hat{g}_{\mathrm{bin}}(V) - \bar{V}(V) + V) - V|] \tag{157}$$

where

$$\bar{V}(v) = \sum_{i=1}^{B} \bar{v}_{i,S_{\mathrm{te}}} \cdot \mathbb{1}_{v \in I_i}, \quad \bar{v}_{i,S_{\mathrm{te}}} := \frac{1}{|I_i|} \sum_{m=1}^{n_{\mathrm{te}}} \mathbb{1}_{v_m \in I_i} v_m$$

where $v_m = f_w(x_m)$. Thus, the total bias is given as the total bias

$$\begin{aligned} \mathbb{E}_{S_{\mathrm{te}}} \mathrm{Bias}_{\mathrm{tot}}(f_w, S_{\mathrm{te}}) &= \mathbb{E}_{S_{\mathrm{te}}} |\mathrm{TCE}(f_w) - \mathrm{ECE}(f_w, S_{\mathrm{te}})| \\ &\geq |\mathrm{TCE}(f_w) - \mathbb{E}_{S_{\mathrm{te}}} \mathrm{ECE}(f_w, S_{\mathrm{te}})| \\ &= |\mathbb{E}_V[|g(V) - V|] - \mathbb{E}_V[|(\hat{g}_{\mathrm{bin}}(V) - \bar{V}(V) + V) - V|]| = \rho(g, \tilde{g}) \end{aligned} \tag{158}$$

where $\tilde{g} := \hat{g}_{\mathrm{bin}}(V) - \bar{V}(V) + V$. Thus, by studying the risk under $\rho$, we can study the total bias.

On the other hand, this $\rho$ is smaller than the L1 distance

$$\mathbb{E}|\hat{g}(V) - g(V)| \geq |\mathbb{E}[|\hat{g}(V) - V|] - \mathbb{E}[|V - g(V)|]| = \rho(g, \hat{g}) \tag{159}$$

by the triangle inequality. Note that L1 distance is smaller than L2 distance, the above minimax result for $\hat{g}$ and $g$ in Eq. (154) is insufficient to understand the bias of TCE and ECE from the lower bound. Instead, we introduce different lower bound given as follows. Conditioned on $W = w$, under the $\beta$-Lipschitzness for $\mathbb{E}[Y|f_w(x) = v]$ and Assumption 3, we have

$$\sup_{g \in \mathcal{G}} \mathbb{E}[||g(V) - V| - |\hat{g}_{\mathrm{bin}}(V) - \bar{V}(V)||^2] \succeq n^{-2\beta/(2\beta+1)}. \tag{160}$$

And We can further obtain

$$\sup_{g \in \mathcal{G}} \mathbb{E}[||g(V) - V| - |\hat{g}_{\mathrm{bin}}(V) - \bar{V}(V)||_\infty] \succeq n^{-\beta/(2\beta+1)}. \tag{161}$$

where $\mathbb{E}|\cdot|_\infty$ is the maximum of the integrand. These lower bounds imply the pointwise lower bound of the difference of the conditional expectation and $V$. Using these lower bounds, we study the difficulty of estimating the TCE at each $V = v = f_w(x)$.

Table 2: Model architecture of CNN on the MNIST experiments.

| Model architecture of CNN (same as Harutyunyan et al. [15]) | |
|---|---|
| (1st layer) Convolutional | 32 filters, $4 \times 4$ kernels, stride 2, padding 1, batch normalization, ReLU |
| (2nd layer) Convolutional | 32 filters, $4 \times 4$ kernels, stride 2, padding 1, batch normalization, ReLU |
| (3rd layer) Convolutional | 64 filters, $3 \times 3$ kernels, stride 2, padding 0, batch normalization, ReLU |
| (4th layer) Convolutional | 256 filters, $3 \times 3$ kernels, stride 1, padding 0, batch normalization, ReLU |
| Fully connected | 128 units, ReLU |
| Fully connected | 2 units, Linear activation |

*Proof.* First, we focus on the relation

$$\sup_{g \in \mathcal{G}} \mathbb{E}[\big||g(V) - V| - |\hat{g}_{\mathrm{bin}}(V) - \bar{V}(V)|\big|^2] \geq \inf_{\hat{g}} \sup_{g \in \mathcal{G}} \mathbb{E}[\big||g(V) - V| - |\hat{g}(V) - \bar{V}(V)|\big|^2]. \quad (162)$$

We then estimate the minimax L2 estimation error under the semimetric $\tilde{\rho}(g, g') := \mathbb{E}||g(V) - V| - |g'(V) - V||^2$. The proof is the same as that of Lemma 1 in [45], which uses Yang Barron method. The difference is to derive the packing number under the semimetric $\tilde{\rho}$ not the L2 distance. However $\tilde{\rho}$ is nothing but the shifted version of the L2 distance. Thus the order of the packing number is the same as that of L2 distance. Note that since $\mathcal{G}$ is the set of positive functions thus, taking the absolute of $g(V) - V$ does not change the order. Then by using Fano's method, we obtain the result for L2 lower bound. The version of $\| \cdot \|_\infty$ can be proved in the same way. $\qquad \square$

We finally remark that the above pointwise gap is larger than the total bias

$$\sqrt{\mathbb{E}[\big||g(V) - V| - |\hat{g}_{\mathrm{bin}}(V) - \bar{V}(V)|\big|^2]} \quad (163)$$

$$\geq \mathbb{E}_{S_{\mathrm{te}}}[\big|\mathbb{E}|g(V) - V| - \mathbb{E}|\hat{g}_{\mathrm{bin}}(V) - \bar{V}(V)|\big|] = \mathbb{E}_{S_{\mathrm{te}}} \mathrm{Bias}_{\mathrm{tot}}(f_w, S_{\mathrm{te}}) \quad (164)$$

where we used the relation of that L2 is larger than L1 and used the Jensen inequality.

### F.8 Relation to the multiclass settings

Although our study focuses on binary classification, we can extend it to multi-class settings. In existing work [24, 10], the top-label calibration error (top ECE) has been proposed as a measure for multi-class calibration. For instance, in a $K$-class classification problem, we obtain predictions of each label by the final softmax layer in neural networks. We assume that $f_w(x) \in \mathbb{R}^K$ predicts the label by $C := \mathrm{argmax}_k f_w(X)_k$, where $f_w(X)_k$ represents the model's confidence of the label $k \in [K]$. Then top ECE is defined using the highest prediction probability output by $f_w$: $\mathbb{E}|P(Y = C|f_w(X)_C) - f_w(X)_C|$. By considering binning only for the top score, we can compute the ECE in a similar way as binary classification. In this case, since we focus only on the top label, we can treat top-binning ECE in the same way as binary classification, leading to the same generalization and total bias bounds. Our results, therefore, offer flexibility to analyze the widely used top-label calibration error in multi-class settings.

## G   Experimental settings

In this section, we summarize the details of our experiments conduced in Sections 3 and 6.

### G.1   Experiments on the synthetic dataset

For the experiments on the synthetic dataset, we follow Zhang et al. [50] and assume that the distributions of the label $Y$ and the input data $X$ are as follows:

$$P(Y = 1) = P(Y = 0) = \frac{1}{2}, \quad (165)$$

and

$$P(X = x|Y = 1) = \mathcal{N}(x; -1, 1), \quad P(X = x|Y = 0) = \mathcal{N}(x; 1, 1), \quad (166)$$

Table 3: Experimental settings on MNIST [25].

| Experimental setup for MNIST experiments | |
|---|---|
| Task | 4 vs 9 classification |
| Model | CNN with four layers |
| Optimizer | Adam with 0.001 learning rate and $\beta_1 = 0.9$ 
 SGLD with 0.004 learning rate (decaying by a factor 0.9 after each 100 iterations) |
| Batch size | 128 (for Adam) or 100 (for SGLD) |
| Num. of training samples | $[75, 250, 1000, 4000]$ |
| Num. of epochs | 200 |
| Num. of samples for CMI estimation | 5 |
| Num. of samplings for $U$ | 10 |
| Num. of recalibration dataset (existing methods) | 100 |

where $\mathcal{N}(x; m, \sigma)$ is the Gaussian distribution with mean $m$ and standard deviation $\sigma$. Then, the probability of $Y = 1$ given $x$ can be expressed as

$$P(Y = 1 | X = x) = \frac{1}{1 + \exp(2x)}.$$

We also define the prediction models as follows:

$$z = f_w(x) = (z_1, z_2) = \left( \frac{1}{1 + \exp(-\beta_0 - \beta_1 x)}, \frac{\exp(-\beta_0 - \beta_1 x)}{1 + \exp(-\beta_0 - \beta_1 x)} \right),$$

where $w = \{\beta_0, \beta_1\}$ are parameters.

Under these settings, we can calculate the closed-form of the canonical calibration function $\pi(z) = (\pi_1(z), \pi_2(z))$, where

$$\pi_1(z) = \left( 1 + \exp\left[ -2 \frac{\beta_0 + \log(1/z_1 - 1)}{\beta_1} \right] \right)^{-1}, \quad \pi_2(z) = 1 - \pi_1(z).$$

Due to this closed-form calibration function, we can estimate the TCE based on Monte Carlo integration. In this paper, we use $10^6$ random samples generated from Eqs. (165) and (166) and evaluate the sample average value of $|z_1 - \pi_1(z)|$ as the estimator of TCE. Furthermore, we estimated the Lipschitz constant $L$ by taking the maximum values of the gradient of $\pi_1(z)$.

## G.2 MNIST and CIFAR experiments

**Model architectures, datasets, model training process, and implementation:** We summarize the details of model architectures for CNN, datasets, and model training process in Tables 2-4. Our experiments were conducted by adapting the code from Harutyunyan et al. [15] [4] to suit our experimental configurations. Consequently, the datasets utilized in this study were normalized in accordance with the implementation provided in the referenced repository. We used NVIDIA GPUs with 32GB memory (NVIDIA DGX-1 with Tesla V100 and DGX-2) for MNIST (SGLD) and CIFAR-10 experiments. We also used CPU (Apple M1) with 16GB memory for the other experiments.

Table 4: Experimental settings on CIFAR-10 [21].

| Experimental setup for CIFAR experiments | |
|---|---|
| Task | dog-or-not classification |
| Model | ResNet-50 pretrained on ImageNet |
| Optimizer | SGD with 0.01 learning rate and 0.9 momentum 
 SGLD with 0.01 learning rate (decaying by a factor 0.9 after each 300 iterations) |
| Batch size | 64 |
| Num. of training samples | $[500, 1000, 5000, 20000]$ |
| Num. of epochs | 40 |
| Num. of samples for CMI estimation | 2 |
| Num. of samplings for $U$ | 5 |
| Num. of recalibration dataset (existing methods) | 100 |

**Mutual information estimation:** We cannot estimate the mutual information $I(l(\mathcal{A}(\tilde{Z}_S, R), \tilde{Z}, B); S | \tilde{Z})$ in our bounds using the approach of Harutyunyan et al. [15] and

---

[4] https://github.com/hrayrhar/f-CMI

Hellström & Durisi [17]. This is because our loss function $l(\mathcal{A}(\tilde{Z}_S, R), \tilde{Z}, B)$ assumes continuous values, while these works specifically focus on discrete random variables, such as the output values of 0-1 loss or the predicted labels of classifiers. Hence, we developed a plug-in estimator for $I(l(\mathcal{A}(\tilde{Z}_S, R), \tilde{Z}, B); S|\tilde{Z})$ [19, 20, 32], which is computed using estimators for the probability density of $l(\mathcal{A}(\tilde{Z}_S, R))$ and $S_{\mathrm{tr}}$, as well as their joint probability density, employing $k$-nearest-neighbor-based density estimation [27]. The estimation strategy is incorporated into the `sklearn.feature_selection.mutual_info_classif` function (we refer to the following link: [https://scikit-learn.org/stable/modules/generated/sklearn.feature_selection.mutual_info_classif.html](https://scikit-learn.org/stable/modules/generated/sklearn.feature_selection.mutual_info_classif.html)). We set $k = 3$ following the default setting of this function and Kraskov et al. [20], Ross [32].

**Standard deviation evaluation of our bounds:** The standard deviation in Table 1 and Figures 2-5, which is almost unrecognizable due to its small value, are attributed to the randomness inherent in various experimental settings during model training, i.e., randomness of the training dataset and the initial model parameters. For example, in the MNIST experiments in Table 3, the standard deviation of our bound was evaluated under the $5 \times 10$ models.

# H   Additional experimental results

In this section, we show the additional results obtained from our experiments.

## H.1   Bound plot on UWB

We show the results of our bound in Eq.(14) using UWB, as shown in Figure3, which was omitted from the main paper due to page limitations. As we discussed in Section 6, we can see the importance of the choice of $B$ to obtain nonvacuous bound values. We also observed that our optimal choice, $B = \lfloor n^{1/3} \rfloor$, is effective in obtaining nonvacuous bounds.

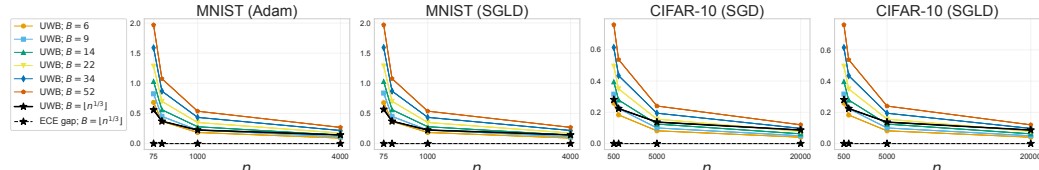

Figure 3: Behavior of the upper bound in Eq. (14) for various $B$ as $n$ increases (mean $\pm$ std.). For clarity, only the results using UWB are shown. The ECE gap is evaluated by estimating $\mathbb{E}_{R, S_{\mathrm{tr}}, S_{\mathrm{te}}}[|\mathrm{ECE}(f_W, S_{\mathrm{te}}) - \mathrm{ECE}(f_W, S_{\mathrm{tr}})|]$. The ECE gap is shown for $B = \lfloor n^{1/3} \rfloor$ since the change in $B$ did not result in significant differences.

## H.2   Bound plot on recalibration reusing training dataset

We further show the plots of our bound for the recalibration scenario in Figure 4. The relationship between $n$, $B$, and bound values is similar to that observed in the non-recalibration case. Interestingly, the choice of optimal $B$ is crucial for obtaining a small bound value when we conduct recalibration.

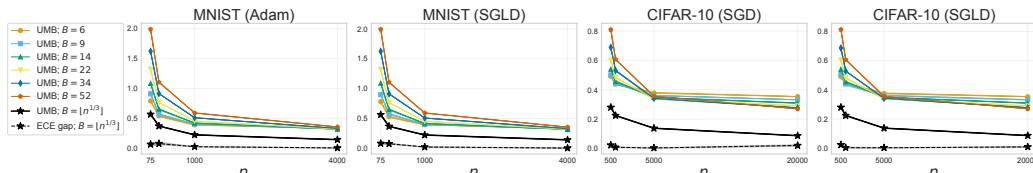

Figure 4: Behavior of the upper bound in Eq. (141) as $n$ increases for different number of bins (mean $\pm$ std.) when using UMB after recalibration.

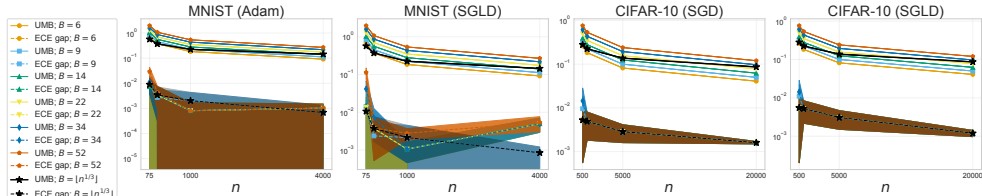

Figure 5: Behavior of the upper bound in Eq. (14) for various $B$ as $n$ increases (mean $\pm$ std.; log-scale) when UMB is used. The ECE gap is evaluated by estimating $\mathbb{E}_{R,S_{\text{tr}},S_{\text{te}}}[|\text{ECE}(f_W, S_{\text{te}}) - \text{ECE}(f_W, S_{\text{tr}})|]$. These results show that the variance of the ECE gap obtained in non-optimal $B$ settings is large, while the ECE gap in settings based on the optimal $B$ is stable.

## H.3 Bound plot for the various number of bins

We examined the ECE gap for various bin sizes using the same setup as Figure 2, and these results are presented in Figure 5. We plotted them on a log scale to illustrate how the ECE gap and upper bound behave with different bin sizes. We found that sometimes bins other than the optimal can yield a better generalization gap. However, the optimal bin size minimizes the total bias as stated in Theorem 5, not necessarily the generalization gap as in Theorem 4. On the other hand, the optimal was found to be numerically stable, although, in certain models, high variance was observed for certain bin sizes, with the ECE gap occasionally not decreasing as increases.

## H.4 Empirical verification of Lipschitz continuous for $\mathbb{E}[Y|f(X)]$.

As discussed in Appendix F, Assumption 2 is generally mild in practice. To assess the empirical plausibility of the Lipschitz continuity assumption in Assumption 2, we performed a numerical evaluation using ResNet experiments. Specifically, we checked whether the value of $\mathbb{E}[Y|f(X)]$, estimated via binning, exhibits relatively smooth variations. These results are presented in Figure 6. The findings indicate that the estimated values fluctuate smoothly to a significant degree, providing empirical support for the Lipschitz continuity assumption. This strengthens the validity of our assumptions and confirms the applicability of our methods in practical, real-world scenarios.

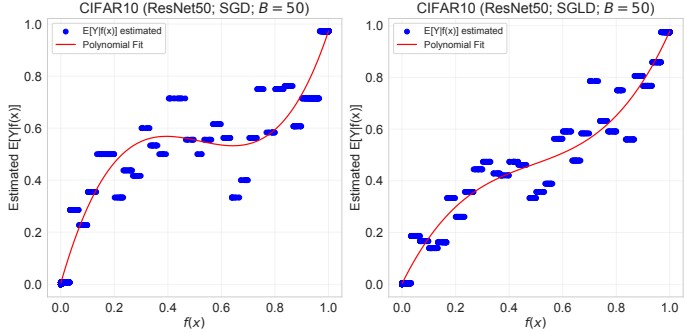

Figure 6: Behavior of the estimator of $\mathbb{E}[Y|f(X)]$ in the ResNet experiments. The red line is the (third-order) Polynomial function fitted to the estimated values of $\mathbb{E}[Y|f(x)]$.

