# OpenReview forum: "Information-theoretic Generalization Analysis for Expected Calibration Error"
_NeurIPS.cc/2024/Conference — NeurIPS 2024 poster_

### Official Review · Reviewer_kktU · 2024-07-10

**Soundness:** 3
**Presentation:** 2
**Contribution:** 2
**Rating:** 5
**Confidence:** 4

**Summary:**

This paper analyzes the estimation bias and generalization error of the expected calibration error (ECE). Specifically, in a binary classification setting, the authors provide an upper bound for the total bias with an improved convergence rate, applicable to both uniform mass and uniform width binning strategies. They also determine the optimal number of bins to minimize the total bias. Furthermore, the authors utilize the information-theoretic generalization framework, particularly the Conditional Mutual Information (CMI) framework, to characterize the generalization of ECE.

**Strengths:**

1. This paper achieves a tighter bound for total bias compared to previous works.

2. The optimal number of bins is determined using the upper bound of the total bias.

**Weaknesses:**

1. As the authors themselves note, a significant limitation is that the analysis in this work is only applicable to binary classification.

2. Some assumptions (e.g., Assumption 2) are not well justified.

3. The writing has significant room for improvement; several arguments are unclear or misleading.

Please find more details in the questions below.

**Questions:**

1. Does Assumption 2 hold true in practice? Is there a way to verify it? Additionally, what is the motivation behind assuming $n_e\geq 2B$? How is this assumption utilized? If $n_{te}\leq 2B$, will Theorems 2 and 3 still be valid?

2. In the proof sketch of Theorem 2, you mention that $\mathrm{ECE}$ and $\mathrm{TCE}$ could be re-written. While they seem correct to me, could you elaborate on how $\mathrm{TCE}(f_\mathcal{I})$ is obtained in the form shown in Line 152? I did not find the details in the complete proof.

3. According to Theorem 5, do the upper bounds indicate that UWB is a better binning strategy than UMB, given that UMB has an additional $\mathrm{fCMI}$ bias term? It seems that only UMB's expected binning bias is sensitive to the training data, which might be seen as a disadvantage in terms of the upper bound.

4. The writing can be significantly improved. For example, in Line 244, you mention "Our theory might guarantee the ECE under test dataset for them." Do you mean your theory might guarantee low ECE under the test dataset? Additionally, in Lines 251-252, "This implies that if the model generalizes well, evaluating the ECE using the training dataset may better reduce the total bias than that using test dataset." Why does evaluating ECE reduce total bias? What we really care about is ECE on unseen/test data. How does evaluating ECE on training data affect this purpose?

5. Why is the metric entropy method only used for UWB? It seems that you upper bound $\mathrm{eCMI}$ by the $\mathrm{fCMI}$ term first in your proof. What prevents you from giving a similar result for UMB?

6. In Line 339-340, you mention that "a notable trend towards acquiring relatively stable nonvacuous bounds can be observed when adopting $B =\lfloor n^{1/3} \rfloor$", but according to Figure 1, it seems $B=52$ is tighter than $B=B =\lfloor n^{1/3} \rfloor$ in most cases. Could you clarify this?

7. Since $\mathrm{eCMI}$ and $\mathrm{fCMI}$ terms are key components in both standard generalization error and expected total bias of calibration error, do you have any new insights into the relationship between calibration and generalization from this perspective?

**Limitations:**

The authors have thoroughly discussed some limitations of this work.

---

> ### Author Rebuttal · Authors · 2024-08-06
>
> We would like to express our deepest appreciation for your insightful reviews and suggestions. We sincerely summarize our responses to you as follows.
>
> ### Q.1: Regarding the strictness of Assumption 2 and the assumption of $n_e\geq 2B$
> **A.**
> First, please refer to the global response for the discussion on the necessity of Assumption 2. In practice, Assumption 2 is reasonably mild. For example, if label distributions follow a Bernoulli distribution with the mean depending on the input $x$, Assumption 2 is met [E]. This is a relatively weak assumption in binary classification. Moreover, existing studies have shown that many common benchmark datasets are consistent with this assumption (e.g., [F]).
> To ascertain whether the assumption of Lipschitz continuity is empirically plausible, we conducted a numerical evaluation based on ResNet experiments. This involved checking if the value of $E[Y|f(X)=v]$, estimated through binning, exhibits relatively smooth variations. These findings are depicted in Fig. 6 of the attached PDF. The results demonstrate that the estimated quantity $E[Y|f(X)=v]$ fluctuates smoothly to a considerable extent, thereby supporting the validity of the Lipschitz continuity assumption. This supports the robustness of our assumptions and the applicability of our methods in real-world scenarios. We will incorporate these results into our paper.
>
> Regarding the assumption $n_e\geq 2B$, it is crucial for UMB, as it guarantees the proper construction of bins. In UMB, we first use $B$ samples to build the bins. We then partition the remaining $n_e−B$ samples evenly across these bins. If $n_e < 2B$, we have $n_e−B< B$, preventing equal distribution of samples, thereby rendering UMB inapplicable. Conversely, UWB does not require this assumption since it divides the $[0,1]$ interval into equal-width bins. We will clearly outline this distinction in the revised paper.
>
> [E] Y. G. Yatracos. A Lower Bound on the Error in Nonparametric Regression Type Problems. Ann. Stat., 1988.
> [F] B. Zadrozny et al. Transforming Classier Scores into Accurate Multiclass Probability Estimates. KDD '02.
>
> ###  Q.2: How to obtain TCE and ECE reformulations.
> **A.**
> The proofs of those reformulations are provided in Appendix C, specifically in Lemma 1. In the revised version, we will explicitly reference this lemma to avoid any confusion.
>
> ###  Q.3: Reason why UMB has an additional term compared to UWB in Theorem 5,
> **A.**
> The additional term (fCMI) arises because UMB uses training data to construct bins. This means we must consider overfitting not only during learning for $f_w$ but also in the bin construction process. This leads to extra fCMI terms in UMB compared to UWB, which is a potential disadvantage. Practically, UMB is designed to prevent issues with bins having no samples when $n_e$ is small. When sufficient data is available, we can partition the entire dataset into training data for learning $f_w$ and another dataset for bin construction. This approach avoids the extra fCMI term and has been explored in previous work [22].
>
> ### Q.4: The meaning of Line 244 and Lines 251-252
> **A.**
> Intent of Line 244: Our goal is to show that, similar to how ERM bounds test loss by training loss plus a generalization gap, our Theorem 4 bounds test ECE by training ECE plus a generalization gap. Some studies [21, 26, 36] use regularization to minimize training ECE in their algorithms, so if a small eCMI or fCMI ensures a small generalization gap, the upper bound of test ECE will also be minimized, thus providing theoretical assurance akin to ERM’s generalization theory.
>
> Lines 251-252: In ERM, the expected loss can be estimated with a validation dataset, with an error rate of $O(1/n_{te}^{1/2})$. However, ECE converges to TCE at a much slower rate of $O(1/n_{te}^{1/3})$. Increasing validation data does not significantly improve this rate. Since training dataset size is often much larger than the test data size, if the generalization error (eCMI or fCMI) is small, using training data to estimate ECE could result in a smaller bias of $O(1/n_{tr}^{1/3})$. Thus, using training data for estimating TCE could be a promising approach. We will clarify these points in the revised paper.
>
> ### Q.5: The difficulty of deriving the metric entropy bound for for UMB
> **A.**
> Deriving the metric entropy bound for UMB is challenging due to the difficulty of constructing the appropriate $\delta$-cover of functions. As highlighted in Eq. 117 and Eq. 120, we need to set $\delta$ smaller than the width of the bins. In the case of UWB, the bin width is $1/B$; however, in UMB, the bin width is not fixed and varies according to the data. This dependency adds complexity to preparing the appropriate $\delta$-cover.
>
> ### Q.６: Clarification in Line 339-340
> **A.**
> Based on the additional experiments, we will revise the explanation as follows:
> - The optimal $B$ is determined based on the upper bound of the total bias (Thoerem 5) and does not necessarily minimize the generalization error bound (Theorem 4). However, setting $B$ in this way ensures that we avoid vacuous values, numerical instabilities and even increase of the ECE gap as we increase $n$.
>
> For the details, please see Fig. 5 in the PDF and response to Reviewers 9r8F (Q.4) (and to Reviewer 8W41; Q.4 for TCE and the total bias evaluation on toy data experiments.).
>
> ### Q.7: New insights into the relationship between calibration and generalization
> **A.** As stated in the global response, the total bias of the TCE and ECE matches the rate achieved in nonparametric regression, conclusively showing that estimating TCE is as challenging as estimating conditional probabilities using nonparametric regression. Thus, compared to usual generalization problems such as classification error, evaluating TCE requires more samples, and achieving a fast learning rate of $O(1/n)$ under parametric learning settings seems to be not possible, highlighting a distinct difference.

---

> > ### Comment · Reviewer_kktU · 2024-08-12
> > **Thank you for the responses**
> >
> > I would like to thank the authors for their responses and apologize for not engaging in the discussion earlier. I have read all the reviews and the corresponding author responses.
> >
> > My concern regarding Assumption 2 has been adequately addressed, and further clarifications have been provided for other questions. I will increase my score to 5 accordingly. However, I am not assigning a higher score because the analysis in this paper is limited to binary classification, with no clear path to extending it to a multi-classification setting. While I also appreciate the minimax results, I am concerned that the analysis (and potentially the assumptions) might require significant refinement in more general or practical settings.

---

> > > ### Author Response · Authors · 2024-08-12
> > > **Acknowledgement and clarification of the limitation of the binary classification**
> > >
> > > We sincerely appreciate your thorough review of our responses to the reviewers’ comments.
> > > And, we are delighted that you have decided to increase your score.
> > >
> > > Concerning the limitations inherent in binary classification, we addressed similar points in our response to Reviewer 8W41 (Q.1). Please refer to that section for details. In summary, our analysis can extend to multiclass scenarios involving top-calibration error, such as when using softmax for classification. In these cases, evaluating the calibration of the highest softmax score is standard practice, and these metrics can be reduced to a formulation similar to binary classification, enabling our analysis.
> > >
> > > Concerning the minimax theorem, additional assumptions are necessary because the theorem deals with worst-case distributions. Worst-case scenarios may occasionally involve settings that are unrealistic or unlikely. However, for establishing upper bounds, as in Theorem 5, Assumption 2 is generally sufficient, making it more practical and likely to be met in real-world scenarios.
> > > We hope these answers will further clarify your understanding.

---

### Official Review · Reviewer_8W41 · 2024-07-11

**Soundness:** 3
**Presentation:** 3
**Contribution:** 3
**Rating:** 6
**Confidence:** 4

**Summary:**

This paper investigates the estimation bias in expected calibration error (ECE) for binary classification models, focusing on uniform mass binning (UMB) and uniform width binning (UWB). The authors present a comprehensive theoretical analysis, establishing upper bounds for the bias and the generalization error. Based on the convergence rates of binning and statistical bias, they identify the optimal number of bins to minimize the total estimation bias.

**Strengths:**

* The paper provides a comprehensive analysis of the estimation bias in ECE, providing upper bounds and optimal bin size choices for both UWB and UMB.
* The authors further derive upper bounds for the generalization error between ECE and TCE using an information-theoretic approach.
* Numerical experiments on deep learning tasks confirm that the derived bounds are non-vacuous.

**Weaknesses:**

* The provided results only apply to binary classification, and require Lipschitz continuity which may not be necessarily satisfied in deep learning models. Also, these bounds are analyzing the ECE using test data but not training data, making them less applicable since test data are not always available in practice.
* The convergence rates of the information-theoretic generalization bounds heavily depend on the actual rate of eCMI and fCMI measures, which are not directly clear in analysis. In theorem 6, the authors show that eCMI scales as O(log n) based on metric entropy, but this bound involves the dimensionality d, and is thus hardly applicable to deep learning models.
* For experimental results, only the statistical bias is evaluated but not the total generalization error. It is also hard to see to what extent these bounds are tight in the current results. These bounds are also hard to estimate due to the existence of eCMI or fCMI measures. I would suggest the authors additionally consider some synthetic settings where TCE, eCMI, and fCMI are analytically tractable to show the tightness of the bounds. (maybe Gaussian data points?)

**Questions:**

Recent information-theoretic bounds have shown improved rates of O(1/n) under the interpolating regime, and also direct computational tractability with loss CMI or entropy metrics. It may be worth some discussions on whether these techniques can be adopted to acquire tighter bounds.

Tighter Information-Theoretic Generalization Bounds from Supersamples. ICML 2023.

Rethinking Information-theoretic Generalization: Loss Entropy Induced PAC Bounds. ICLR 2024.

**Limitations:**

Yes

---

> ### Author Rebuttal · Authors · 2024-08-07
>
> We would like to express our deepest appreciation for your insightful reviews and suggestions. We sincerely summarize our responses to you as follows.
>
> ### Q.1: Regarding the setting of binary classification and the Lipschitz continuity.
> **A.**
> Although our study focuses on binary classification, we can readily extend it to multi-class settings. In existing studies [22, 8], the top-label calibration error (top CE, TCE) has been proposed as a measure for multi-class calibration. For instance, in a $K$-class classification problem, we obtain predictions for each label by the final softmax layer in neural networks. We assume that $f_w(x)\in \mathbb{R}^K$ predicts the label by $C:=\mathrm{argmax}_k f_w(X)_k$, where $f_w(X)_k$ represents the model’s confidence of the label $k \in K$. The top CE (TCE) is then defined using the highest prediction probability output by $f_w: TCE:=\mathbb{E}|P(Y=C| f_w(X)_C)-f_w(X)_C|$. By considering binning only for the top score, we can compute the ECE (top-binning ECE) in a manner similar to binary classification. In this case, since we focus only on the top label, we can treat top-binning ECE in the same way as binary classification, which results in the same generalization and total bias bounds. Our results, therefore, offer flexibility to analyze the widely used top-label calibration error in multi-class settings. We will add this discussion to the paper.
>
> Regarding Assumption 2, this does not imply the Lipschitz continuity of the function $f_w(\cdot)$ itself, but the Lipschitz continuity of the conditional expectation $\mathbb{E}[Y|f_w(x)=v]$, which is a kind of characteristic of the data itself. For a more comprehensive discussion on the Lipschitz continuity assumption, please refer to the global response provided above and the response to Reviewer kktU (Q1).
>
> ### Q.2: Regarding the necessity of test dataset in the ECE evaluation.
> **A.**
> If your question concerns the difference between IT-based bounds and generalization error upper bounds like PAC-Bayes, which can be evaluated using only training data, model distribution, and prior distribution, note that the IT-based upper bound can still be evaluated even if only training data is available in practice. As discussed in Line 257, $\mathrm{eCMI} \leq \mathrm{fCMI}\leq  I(W;S_\mathrm{tr})$ [13,15] holds, and $I(W;S_\mathrm{tr})$ only depends on the training dataset. Thus, by replacing eCMI and fCMI in Theorem 4 and 5 with this mutual information, we get the bound that is independent of the test dataset. On the other hand, the inclusion of test data dependence in eCMI and fCMI through the index \tilde{U} provides the benefit of tighter bounds.
>
> If our explanation does not align with your concern, please provide further questions during the discussion phase.
>
> ### Q.3: Regarding the convergence behavior of our bounds and the vacuousness of the metric-entropy-based bound
> **A.** Your points are accurate. The primary distinction between IT bounds and UC bounds lies in their objectives:
> - IT Bounds: These are algorithm-specific and aim to provide detailed insights into how a particular algorithm performs in terms of generalization. They focus on analyzing these bounds to understand the performance of specific algorithms. By further assuming specific algorithm classes, such as stable algorithms or stochastic convex optimization, we can derive the theoretical behaviors of the mutual information.
> - UC Bounds: These are derived to determine the necessary sample size for achieving generalization performance based on convergence rates, independent of the algorithm used. Typically, UC bounds employ metric entropy to derive results. In response to your suggestion, we provide UC bounds that use the fat-shattering dimension, which is independent of the model’s dimension as detailed in the global response above.
>
> ### Q.4: Regarding the statistical bias evaluation, and the TCE evaluation experiment on the synthetic experimental settings.
> **A.**
> Evaluating the upper bound for TCE proved challenging with benchmark datasets and experiments using CNN and ResNet. However, we were able to conduct TCE evaluation experiments using a synthesized dataset based on the settings in [Z]. The results are shown in Fig. 4 of our additional PDF.
>
> We numerically evaluated both sides of Corollary 1. Since Corollary 1 does not involve eCMI, this approach allows for a more accurate assessment of the bound’s tightness and optimality. The first two rightmost figures display the total bias and upper bound of Corollary 1 under the optimal bin size, demonstrating that the bound effectively captures the behavior of the total bias. The last two figures show the TCE gap plotted against different bin sizes compared to the theoretically optimal number of bins. Despite some fluctuations due to a small sample size $n$, the actual behavior closely aligns with theoretical predictions, validating the optimal bin size.
>
> [Z]: J. Zhang et al. Mix-n-Match: Ensemble and Compositional Methods for Uncertainty
> Calibration in Deep Learning. ICML2020.
>
> ### Q.5: Regarding the improvement of the convergence rate
> **A.**
> As discussed in the reply to Reviewer Ckaf (Q.2), we cannot use the technique proposed in [37] directly. Moreover, we have derived a minimax lower bound for total bias, which is $\mathcal{o}(1/n^{1/3})$, implying that the problem of estimating TCE is as inherently challenging as estimating conditional probabilities in the context of nonparametric regression. Therefore, deriving a fast rate of $\mathcal{O}(1/n)$ under a parametric learning setting is not feasible for the ECE estimation problem. However, discussing whether the techniques adopted in the papers you suggested can be used to improve the constants of the bounds constitutes important future work. We would like to specify this in the conclusion section.

---

> > ### Comment · Reviewer_8W41 · 2024-08-09
> >
> > Thank you for the response and additional experiments. I'll maintain my positive rating.

---

> ### Author Response · Authors · 2024-08-10
> **Thank your for the reply**
>
> Thank you for confirming our response. We believe that the additional experiments address your concerns. If there are any further issues or concerns that need to be addressed to improve the score, please let us know. We are happy to discuss them further.

---

### Official Review · Reviewer_Ckaf · 2024-07-13

**Soundness:** 3
**Presentation:** 3
**Contribution:** 3
**Rating:** 6
**Confidence:** 4

**Summary:**

The paper studies  the expected calibration error  using information-theoretical tools. They derive different tight fCMI and eCMI bounds in this setting. Empirical results show that the results are nonvacuous.

**Strengths:**

1/ The paper is in general well written. Adequate discussions are given in the main body of the paper and the appendices.

2/ The paper provides the first information-theoretic comprehensive analysis of the bias associated with the ECE when using the test and training datasets.

3/ The theoretical results seem sound. I skimmed through most of the proofs (I did not go through all of them in detail)  but the proofs are well-structured and easy to follow.

3/ Empirical results show that the bound is tight for deep learning models.

**Weaknesses:**

The only weakness, if any, is perhaps that the paper uses conventional machinery for deriving information-theoretic generalization bounds and that it has not developed novel proof techniques.

**Questions:**

Besides fCMI and eCMI based bounds, is it possible to extend the analysis and derive $\Delta$-L based bounds [37]? These bounds are typically tighter compared to fCMI and eCMI.

**Limitations:**

The limitations are adequately addressed in the conclusion.

---

> ### Author Rebuttal · Authors · 2024-08-06
>
> We would like to express our deepest appreciation for your insightful reviews and suggestions. We sincerely summarize our responses to you as follows.
>
> ### Q.1: Regarding the novelty of proof techniques
> ### A.
> First, it is important to clarify that our techniques are fundamentally different from those used in previous studies, as referenced in [11, 10, 22]. In these studies, the error bound for the ECE is derived through the following three steps: (i) Initially, it is shown that the samples assigned to each bin are i.i.d.; (ii) the Hoeffding's inequality is then applied to derive that $|\mathbb{E}[Y|f_{\mathcal{I}}(x)] - f_{\mathcal{I}}(x)| = O(\sqrt{n_{te}/B})$ for each bin; and (iii) these error bounds are summed up across all bins to yield $O(B/\sqrt{n_{te}})$. This approach leads to a slow convergence, which can be attributed to the separate analysis conducted for each bin, requiring multiple applications of concentration inequalities.
>
> In our approach, we leverage a reformulation of the ECE and TCE as outlined in lines 151 and 152. These equations frame both the ECE and TCE in terms of the relationship between the empirical expectation and its population expectation. This framework allows us to apply concentration inequalities to the errors in all bins combined, specifically to the ECE and TCE themselves. Unlike traditional methods, we employ McDiarmid's inequality—notably, leveraging the technique within its proof—to derive an upper bound without the need to decompose the error into individual bins. This strategy eliminates the necessity to compute the cumulative sum of statistical biases across bins, achieving a more efficient convergence rate of $O(\sqrt{B/n})$ in expectation.
>
> While McDiarmid’s inequality is widely used, our application of it within the binning context is technically novel because it requires meticulous handling of data replacement, as detailed in Appendix D.1.1. This approach in treating the data and applying the inequality offers a marked improvement in the convergence order compared to existing studies.
>
> We would also like to highlight the novelty of Theorem 6. Our work extends beyond the standard construction of a $\delta$-cover for Lipschitz functions. As outlined in Appendix E.3, our approach involves a careful design of the $\delta$-cover to ensure that both the output of the original function $f_w$ and that of the $\delta$-cover function are categorized into the same bin. This careful alignment is significant to control the discretization error associated with the binning ECE when employing $\delta$-cover functions. This strategy, which we introduce for the first time, is crucial for enhancing the accuracy and effectiveness of our theoretical framework, providing a novel contribution to the field.
>
> ### Q.2: Relation to the result in [37]
> ### A.
> The eCMI appearing in Theorem 4 (Eq. (15)) closely aligns with the $\Delta L$ bound (loss difference) of Theorem 1 of reference [37].
> Specifically, the eCMI term in Eq. (15) is not based on the value of ECE itself. Instead, it is derived from the difference between the test data ECE and the training data ECE. This approach aligns with the $\Delta L$ (loss difference) as shown in Theorem 1 of reference [37].
>
> However, extending our bounds using the techniques of [37] presents significant challenges. The $\Delta L$ bound in [37] defines the loss gap for a single data index $i$ as $\Delta L_i$ and utilizes the symmetry of each individual index to derive fast rate bounds, as demonstrated in Theorem 4.3. In contrast, our bound requires treating all $n$ indices simultaneously. This necessity arises because ECE is a nonparametric estimator that uses all $n$ indices, unlike usual losses such as the 0-1 loss, where an estimator can be constructed using a single index. Consequently, the techniques from [37] that utilize the symmetry of a single index are not applicable to our context, making it difficult to employ the methods from [37].
>
> Furthermore, as previously discussed, we have derived a minimax lower bound for total bias, amounting to $\mathcal{o}(1/n^{1/3})$ under Lipschitz conditions. This implies that the problem of estimating ECE is as fundamentally challenging as estimating conditional probabilities in the context of nonparametric regression. Therefore, deriving a fast rate of $\mathcal{O}(1/n)$ under a parametric learning setting, as seen in [37], is not feasible for the ECE estimation problem.
>
> We will add these discussions in the revised version of our paper.

---

> > ### Comment · Reviewer_Ckaf · 2024-08-10
> >
> > Thank you for the response. I'll maintain my positive score.

---

> > > ### Author Response · Authors · 2024-08-10
> > > **Thank you for the comment.**
> > >
> > > Thank you for confirming our response. If there are any further issues or concerns that need to be addressed to improve the score, please let us know. We are happy to discuss them further.

---

### Official Review · Reviewer_9r8F · 2024-07-15

**Soundness:** 3
**Presentation:** 3
**Contribution:** 3
**Rating:** 7
**Confidence:** 4

**Summary:**

This paper presents a comprehensive analysis of the estimation bias for expected calibration error (ECE), focusing on two common binning strategies: uniform mass and uniform width binning. The analysis establishes upper bounds on the bias, resulting in an improved convergence rate. Furthermore, these bounds reveal the optimal number of bins needed to minimize the estimation bias. The study also extends the bias analysis to generalization error analysis using an information-theoretic approach, deriving upper bounds that facilitate numerical evaluation for recalibration methods based on training data. Experiments with deep learning models demonstrate that the bounds are nonvacuous, due to the information-theoretic generalization analysis approach.

**Strengths:**

As the author pointed out, the existing literature lacks a theoretical analysis of the estimated ECE and a more principled approach to estimation. This paper addresses and closes this gap.

**Weaknesses:**

1.	Tightness issue of the upper bound in Corollary 1. It is commendable that the authors included a discussion on the tightness of Equation 12. However, it would be more rigorous to formally establish a minimax lower bound for the estimation bias that applies to all types of estimators. The authors could either use existing results from Tsybakov [33] or construct a worst-case analysis using Le Cam’s method to establish the lower bound. While it is acceptable if the constant does not match the upper bound, it is crucial to demonstrate the rate.

2.	A drawback of information-theoretic (IT) bounds is the implicit dependency on the algorithm. For example, Theorem 7 appears very similar to Theorem 4, as the recalibration-induced dependence is encapsulated in the CMI term. The authors should provide more commentary on this aspect and clarify the connection between Theorems 6 and 5, as well as which bound is more practical for use.

3.	In the caption of Figure 1, It is said that the ECE gap does not change significantly in B. How can we justify that the selection of $B = n^{1/3}$ is better? Figure 1 primarily plots the bound in (14), but as I mentioned earlier, such a bound can be very loose, and more empirical justification should be provided for the selection of optimal B.

**Questions:**

1.	Clarification: what is the ECE gap plotted in Figure 1 and Table 1? Estimated ECE? To my understanding, all the bounds in Figure 1 are quite loose. Shouldn't we plot the left-hand side of Equation 14 for a more accurate comparison?

2.	How should the error bars for the bound values in Table 1 be interpreted? What is the source of the randomness?

3.	It is not accurate to say that I(S;W)=O(log n) in Line 258, as such Barron’s result assume that samples Z are conditional i.i.d. given model parameter w. However, in the learning context, we always assume that training samples are i.i.d.

**Limitations:**

The Limitations are well addressed in section 7.

---

> ### Author Rebuttal · Authors · 2024-08-07
>
> We would like to express our deepest appreciation for your insightful reviews and suggestions. We sincerely summarize our responses to you as follows.
>
> ### Q.1: Derive a minimax lower bound for the total bias.
> **A.** Please see our global response for a minimax lower bound.
>
> ### Q.2: Regarding the drawback of IT bounds.
> **A.**
> The objectives of IT-based generalization bounds and UC bounds differ significantly. The former aims to derive upper bounds that are algorithm-dependent to gain a detailed understanding of how a training algorithm behaves to achieve generalization performance. This also involves analyzing these bounds to quantitatively examine the specific algorithms. In contrast, UC bounds are primarily derived to determine the sample size needed to achieve sufficient generalization performance from a convergence rate perspective, regardless of the algorithms.
>
> Next, we explain the explicit relationship between Theorems 5 (IT bound) and 6 (UC bound).
> In the proof of Theorem 6, we derive the upper bound of eCMI (fCMI) using the metric entropy of the specific $\delta$-cover (Eq. 124). Therefore, the metric entropy-based UC bound serves as a further upper bound to the IT bounds. In addition to this, as shown in the global response, we show the new connection between IT and UC bounds using the fat-shattering dimension, which is independent of the model’s dimension. These insights will be included in the revised manuscript.
>
> ### Q.3: Regarding the ECE gap in Figure 1 and Table 1.
> **A.**
> First, the ECE gap shown in Fig. 1 corresponds to the empirical estimate of the middle expression in Eq. (14). We added the formal definition in Fig. 5 of our additional pdf.
>
> The ECE gap reported in Tab. 1 is evaluated through the empirical estimator of the statistical bias: $E_{R,S_{re}} E [|E[Y|h_{\mathcal{I},S_{{re}}}(X)] - h_{\mathcal{I},S_{{re}}}(X)|] =  E_{R, S_{{re}},S_{te}}ECE(h_{\mathcal{I}, S_{\mathrm{re}}},S_{te})$ for existing recalibration methods due to the definition of recalibration.
> For our proposed method, which uses all training data as recalibration data, it is evaluated as $E_{R,S_{tr}} E [|E[Y|h_{\mathcal{I},S_{tr}}(X)] - h_{\mathcal{I},S_{tr}}(X)|] = E_{R,S_{tr},S_{te}}ECE(h_{\mathcal{I}, S_{tr}},S_{te})$.
>
> Thanks to your suggestion, we realized that this explanation is too brief and would like to add a detailed explanation of how we numerically measured the ECE gap in Appendix G.
>
> Also, we believe that visualizing the estimate of the left-hand term in our theoretical framework is crucial for verifying whether the right-hand side is effectively upper bounded numerically and for analyzing the relationship between these two behaviors (refer to our answer to Reviewer 8W41, Q.4).
>
> ### Q.4: Regarding empirical justification of our optimal $B$.
> **A.**
> Regarding the optimal $B$ and the ECE gap, we initially showed only the ECE gap with this optimal $B$ in Fig. 1 for clarity. Motivated by your review, we examined the ECE gap for various bin sizes using the same setup as Fig. 1, and these results are presented in Fig. 5 of the additional PDF. We plotted them on a log scale to illustrate how the ECE gap and upper bound behave with different bin sizes. We found that sometimes bins other than the optimal $B$ can yield a better generalization gap. However, the optimal bin size minimizes the total bias as stated in Theorem 5, not necessarily the generalization gap (Theorem 4). On the other hand, the optimal $B$ was found to be numerically stable, although, in certain models, high variance was observed for certain bin sizes, with the ECE gap occasionally not decreasing as $n$ increases. Finally, to our knowledge, no existing work has evaluated model performance based on ECE from a generalization perspective. Our contribution is the first bound that allows for this numerical evaluation.
>
> To further assess the optimality of $B$, we need to evaluate the total bias. However, evaluating the upper bound for TCE with benchmark datasets and experiments using CNN and ResNet proved challenging. Therefore, we performed additional experiments using Toy data to more easily evaluate the tightness of the bound and the optimality of $B$. Specifically, we created Toy data following the settings in existing studies and numerically evaluated both sides of Corollary 1. Since Corollary 1 does not involve eCMI, this approach allows for a more accurate assessment of the bound's tightness and optimality. The results are shown in Fig. 4 of the PDF. The first two figures from the right display the total bias and upper bound of Corollary 1 under the optimal bin size, showing that the bound effectively captures the behavior of the total bias. The last two figures plot the TCE gap with different bin sizes compared to the theoretically optimal number of bins. Despite some fluctuations due to a small sample size $n$, the actual behavior closely aligns with theoretical predictions, validating the optimal bin size.
> These facts will be added in Appendix H.
>
> ### Q.5: Regarding the interpretation of the error bars for the bound values in Table 1
> **A.** The error bars in Tab. 1 (and the standard deviation in Fig. 1, which is almost unrecognizable due to its small value) are attributed to the randomness inherent in various experimental settings during model training, i.e., randomness of the training dataset and the initial model parameters. We acknowledge that we have not explained this point clearly enough, so we will include this explanation in Appendix G.
>
> ### Q.6: The issue with the statement on Line 258 about $I(S;W) = O(\log n)$
> **A.**
> Your point is correct, and the cited results consider a setting similar to Bayesian inference, involving conditionally i.i.d. samples. We cited the result as an example illustrating a scenario where mutual information is theoretically controlled. To make the meaning of the citation clearer, we plan to explicitly state that the cited paper addresses conditionally i.i.d. settings.

---

> > ### Comment · Reviewer_9r8F · 2024-08-12
> >
> > I appreciate the authors' effort in preparing the response. It thoroughly addresses all my comments, and I am pleased to see the matched lower-bound results and additional empirical evidence supporting the claim. I will raise my score to 7—this is a solid paper.
> >
> > Please ensure that the new results are incorporated into the final revision, and make sure the figures in the main body are clear and easy to follow.

---

> > > ### Author Response · Authors · 2024-08-12
> > > **Acknowledgement**
> > >
> > > We greatly appreciate your acknowledgment of our responses. We will incorporate the theoretical facts and insights, along with their proofs, discussed during the rebuttal period into the revised version. Additionally, we will improve the clarity of the existing experiments and include the additional experiments in the revised version as well.
> > >
> > > We are delighted that you have decided to increase your score.
> > > Thank you again for your insightful feedback.

---

### Author Rebuttal · Authors · 2024-08-07

We would like to express our sincere appreciation for your insightful reviews and suggestions. First, we will address the common concerns raised by the reviewers. Following that, we will address each individual question.

## Discussion about the lower bound of the total bias
As pointed out by Reviewer 9r8F, here we show the lower bound of the total bias by using the result of the nonparametric binary classification in [A]. The method in [A] is the extension of Fano's method focusing on evaluating mutual information (MI) with the Yang-Barron method [B].

Conditioned on $W=w$ we express $v=f_w(x)$. Let $p(v)$ be the probability density of $v$ induced by $p(x)$ using $f_w(x)$ and $\mathcal{V}\subset [0,1]$ be its support. Define $g(v)=\mathbb{E}[Y|f_w(x)=v]=\mathrm{Pr}[Y=1|v]$. Let $\mathcal{G}$ be a class of candidate conditional probability functions over $\mathcal{V}$ and every candidate $g\in\mathcal{G}$ satisfies $0\leq g(v)\leq 1$ for all $v\in\mathcal{V}$.

We need the following assumption to control MI [A]; Assume that $\mathcal{G}$ has at least one member $g^*$ that is bounded away from 0 and 1, i.e., there exist constants $0<c_1\leq c_2<1$ such that $c_1\leq g^*\leq c_2$. Instead of Assumption 2 in our paper, assume that for any $g\in\mathcal{G}$ and for any $v,v'\in\mathcal{V}$, $|g(v)-g(v')|\leq C|v-v'|^\beta$ with $C>0$. When $\beta=1$, this becomes Assumption 2.

Under the above two assumptions, the following relation holds;
$$
    \sup_{g\in\mathcal{G}}\mathbb{E}|\mathrm{TCE}(f_w)- \mathrm{ECE}(f_w,S_{te})|\geq \inf_{\hat{g}}\sup_{g\in\mathcal{G}}|\mathbb{E}[|g(V)-V|]-\mathbb{E}[|\hat{g}(V)-V|]|\succeq  n^{-\beta/(2\beta+1)},
$$ where $\hat{g}$ is over all valid estimators for $g$ using $n$ samples $(X_m,Y_m)_{m=1}^n$ and the expectation is taken with respect to true $g$. Note that the left-hand side of the above corresponds to the total bias in our paper. Note that the term $\mathbb{E}[|g(V)-V|]$ is the TCE. If we consider $\hat{g}$ as the kernel density estimation, the above result can be used as the lower bound of the kernel ECE, which is also a popular method to estimate the TCE.

From this, when $\beta=1$, the upper bound of our total bias (e.g., Eq.(13)) achieves the same rate as the lower bound thus the binning achieves the optimal rate. However, when $\beta>1$, the upper bounds do not reach the lower bound. As discussed in lines 191-204, this limitation arises because binning cannot exploit the underlying smoothness of the distribution.

In [A], the L1 minimax rate of conditional probability is given as
$$
    \inf_{\hat{g}}\sup_{g\in\mathcal{G}}\mathbb{E}|\hat{g}(V)-g(V)|\succeq  n^{-\beta/(2\beta+1)}.
$$
Combining this with our results, we can see that the rate of the total bias matches the rate of nonparametric regression, conclusively showing that **estimating TCE is as challenging as estimating conditional probabilities.**  We will incorporate these results and proof into our revised paper.

[A] Minimax nonparametric classification rates of convergence, Y. Yang. IEEE Transactions on Information Theory, 1999

[B] Information-theoretic determination of minimax rates of convergence, Y. Yang and A. Barron. The Annals of Statistics, 1999.

## Connection between IT-bound and UC theory (fat-shattering dimension)
s pointed out by Reviewer 8W41, the bound by metric entropy depends on the model's dimensionality, making them unsuitable for large models such as neural networks. Some existing studies [13,15] present the upper bound of the eCMI and fCMI by the dimension-independent complexities, such as the VC dimension for binary classification and connecting IT theory to UC theory.

Inspired by these results, here we provide the upper bound of eCMI and fCMI using such dimension-independent complexities. As provided in the lower bound analysis above, since TCE estimation is similar to the nonparametric regression, we use **fat-shattering dimension** [C] to upper bound the eCMI. Specifically, if our model class $f_w(\cdot)$ satisfies $\delta/4$-fat dimension with $d_{\delta/4}$ for $\delta\in[0,1]$, we have
$$
eCMI(Eq.15)\leq fCMI(Eq.(17))= O\Big(d_{\delta/4}\log \frac{n}{d_{\delta/4}\delta} \log \Big(\frac{n}{\delta^2}\Big)\Big)
$$
which results in the dimension-independent upper bound. To evaluate the fat-shattering dimension for specific models, see [D] for the details. We will incorporate these results into our paper.

[C] Scale-sensitive dimensions, uniform convergence, and learnability. N. Alon, et.al., Journal of the ACM, 1997

[D] Vapnik-Chervonenkis Dimension of Neural Nets. Peter L. Bartlett

## Discussion about Assumption 2
Here, we discuss the necessity of Assumption 2. Estimating TCE involves nonparametric regression of $E[Y∣f(X)=v]$. For finite samples, smoothness assumptions like Lipschitz continuity are required; without them, small changes in $v$ could cause large variations in label outcomes, making estimation impossible [E]. Such smoothness assumptions are standard in nonparametric regression, including kernel-based ECE. Without these assumptions, increasing the sample size would not ensure that training (or test) ECE converges to TCE. As noted in our minimax lower bound discussion, the absence of smoothness ($\beta\to 0$) leads to increasing bias.

[E] Minimax optimal conditional density estimation under total variation smoothness, M. Li, et. al., Electron. J. Statist, 2022.

## Additional numerical validation
We add numerical experiments addressing the concerns raised by the reviewers in PDF:
- Toy data experiments to evaluate the total bias and optimality of $B$ (Fig. 4). For the detailed explanation, see the reply to Q.4 of Reviewer 8W41.
- A logarithmic plot of the ECE gap in Figure 1 to clarify the behavior of the bounds (Fig. 5). For the detailed explanation, see the reply to Q.4 of Reviewer 9r8F.
- Experiments validating the Lipschitz continuity in Assumption 2 (Fig. 6). For the detailed explanation, see the reply to Q.1 of Reviewer kktU.

---

### Decision · Program_Chairs · 2024-09-25

**Decision:**

Accept (poster)

**Comment:**

This work provides a theoretical analysis into the estimation bias for expected calibration error in the binary classification setting, including upper bounds on the bias and discussion of optimal binning strategies. Most reviewers responded positively, highlighting the quality of the writing, the empirical study on the tightness of the obtained bounds, and how the work closes an important gap in the literature. While there was some concern about the restriction to the binary classification setting, the authors claim that the study can be extended to multi-class settings. Including this discussion will greatly improve the applicability of this work. Provided these inclusions are incorporated into the camera-ready submission, I recommend acceptance.